

Atmospheric
Measurement
Techniques



# Intercomparison of NO$_2$, O$_4$, O$_3$ and HCHO slant column measurements by MAX-DOAS and zenith-sky UV–visible spectrometers during CINDI-2

Karin Kreher[1], Michel Van Roozendael[2], Francois Hendrick[2], Arnoud Apituley[3], Ermioni Dimitropoulou[2], Udo Frieß[4], Andreas Richter[5], Thomas Wagner[6], Johannes Lampel[4,7], Nader Abuhassan[8], Li Ang[9], Monica Anguas[10], Alkis Bais[11], Nuria Benavent[10], Tim Bösch[5], Kristof Bognar[12], Alexander Borovski[13], Ilya Bruchkouski[14], Alexander Cede[8,15], Ka Lok Chan[16,26], Sebastian Donner[6], Theano Drosoglou[11], Caroline Fayt[2], Henning Finkenzeller[17], David Garcia-Nieto[10], Clio Gielen[2], Laura Gómez-Martín[18], Nan Hao[19], Bas Henzing[20], Jay R. Herman[8], Christian Hermans[2], Syedul Hoque[21], Hitoshi Irie[21], Junli Jin[22], Paul Johnston[23], Junaid Khayyam Butt[24], Fahim Khokhar[24], Theodore K. Koenig[17], Jonas Kuhn[4,6], Vinod Kumar[6,25], Cheng Liu[26], Jianzhong Ma[22], Alexis Merlaud[2], Abhishek K. Mishra[25], Moritz Müller[15,27], Monica Navarro-Comas[18], Mareike Ostendorf[5], Andrea Pazmino[28], Enno Peters[5,a], Gaia Pinardi[2], Manuel Pinharanda[28], Ankie Piters[3], Ulrich Platt[4], Oleg Postylyakov[13], Cristina Prados-Roman[18], Olga Puentedura[18], Richard Querel[23], Alfonso Saiz-Lopez[10], Anja Schönhardt[5], Stefan F. Schreier[29], André Seyler[5], Vinayak Sinha[25], Elena Spinei[8,30], Kimberly Strong[12], Frederik Tack[2], Xin Tian[9], Martin Tiefengraber[15,27], Jan-Lukas Tirpitz[4], Jeroen van Gent[2], Rainer Volkamer[17], Mihalis Vrekoussis[5,31,32], Shanshan Wang[10,33], Zhuoru Wang[34], Mark Wenig[16], Folkard Wittrock[5], Pinhua H. Xie[9], Jin Xu[9], Margarita Yela[18], Chengxin Zhang[26], and Xiaoyi Zhao[12,b]

[1]BK Scientific, Mainz, Germany
[2]Royal Belgian Institute for Space Aeronomy, Brussels, Belgium
[3]Royal Netherlands Meteorological Institute, De Bilt, the Netherlands
[4]Institute of Environmental Physics, University of Heidelberg, Heidelberg, Germany
[5]Institute of Environmental Physics, University of Bremen, Bremen, Germany
[6]Max Planck Institute for Chemistry, Mainz, Germany
[7]Airyx, Eppelheim, Germany
[8]NASA Goddard Space Flight Center, Greenbelt, Maryland, USA
[9]Anhui Institute of Optics and Fine Mechanics, Chinese Academy of Sciences, Hefei, China
[10]Department of Atmospheric Chemistry and Climate, Institute of Physical Chemistry Rocasolano, Madrid, Spain
[11]Laboratory of Atmospheric Physics, Aristotle University of Thessaloniki, Thessaloniki, Greece
[12]Department of Physics, University of Toronto, Toronto, Canada
[13]A. M. Obukhov Institute of Atmospheric Physics, Russian Academy of Sciences, Moscow, Russia
[14]Belarusian State University, Minsk, Belarus
[15]LuftBlick Earth Observation Technologies, Mutters, Austria
[16]Meteorologisches Institut, Ludwig-Maximilians-Universität München, Munich, Germany
[17]Department of Chemistry & Cooperative Institute for Research on Environmental Sciences (CIRES), University of Colorado, Boulder, USA
[18]National Institute for Aerospace Technology (INTA), Madrid, Spain
[19]European Organisation for the Exploitation of Meteorological Satellites (EUMETSAT), Darmstadt, Germany
[20]Netherlands Organisation for Applied Scientific Research (TNO), The Hague, the Netherlands
[21]Center for Environmental Remote Sensing, Chiba University, Chiba, Japan
[22]Meteorological Observation Center and Chinese Academy of Meteorological Science, China Meteorological Administration, Beijing, China
[23]National Institute of Water and Atmospheric Research, Lauder, New Zealand
[24]National University of Sciences and Technology, Islamabad, Pakistan

**Published by Copernicus Publications on behalf of the European Geosciences Union.**

[25]Department of Earth and Environmental Sciences, Indian Institute of Science Education and Research, Mohali, Punjab, India

[26]School of Earth and Space Sciences, University of Science and Technology of China, Hefei, Anhui, China

[27]Department of Atmospheric and Cryospheric Sciences, University of Innsbruck, Innsbruck, Austria

[28]Laboratoire Atmosphères, Milieux, Observations Spatiales, Université de Versailles Saint-Quentin-en-Yvelines, Centre National de la Recherche Scientifique, Guyancourt, France

[29]Institute of Meteorology, University of Natural Resources and Life Sciences, Vienna, Austria

[30]Virginia Polytechnic Institute and State University, Blacksburg, VA, USA

[31]Center of Marine Environmental Sciences (MARUM), University of Bremen, Bremen, Germany

[32] Energy, Environment and Water Research Center (EEWRC), The Cyprus Institute, Nicosia, Cyprus

[33]Shanghai Key Laboratory of Atmospheric Particle Pollution and Prevention, Department of Environmental Science & Engineering, Fudan University, Shanghai, China

[34]Remote Sensing Technology Institute, German Aerospace Center (DLR), Oberpfaffenhofen, Germany

[a]now at: Institute for the Protection of Maritime Infrastructures, German Aerospace Center (DLR), Bremerhaven, Germany

[b]now at: Measurement and Analysis Research Section, Environment and Climate Change Canada, Toronto, M3H 5T4, Canada

**Correspondence:** Karin Kreher (karin.kreher@bkscientific.eu)

Received: 15 April 2019 – Discussion started: 27 May 2019
Revised: 2 December 2019 – Accepted: 16 December 2019 – Published:

**Abstract.** In September 2016, 36 spectrometers from 24 institutes measured a number of key atmospheric pollutants for a period of 17 d during the Second Cabauw Intercomparison campaign for Nitrogen Dioxide measuring Instruments (CINDI-2) that took place at Cabauw, the Netherlands (51.97° N, 4.93° E). We report on the outcome of the formal semi-blind intercomparison exercise, which was held under the umbrella of the Network for the Detection of Atmospheric Composition Change (NDACC) and the European Space Agency (ESA). The three major goals of CINDI-2 were (1) to characterise and better understand the differences between a large number of multi-axis differential optical absorption spectroscopy (MAX-DOAS) and zenith-sky DOAS instruments and analysis methods, (2) to define a robust methodology for performance assessment of all participating instruments, and (3) to contribute to a harmonisation of the measurement settings and retrieval methods. This, in turn, creates the capability to produce consistent high-quality ground-based data sets, which are an essential requirement to generate reliable long-term measurement time series suitable for trend analysis and satellite data validation.

The data products investigated during the semi-blind intercomparison are slant columns of nitrogen dioxide (NO$_2$), the oxygen collision complex (O$_4$) and ozone (O$_3$) measured in the UV and visible wavelength region, formaldehyde (HCHO) in the UV spectral region, and NO$_2$ in an additional (smaller) wavelength range in the visible range. The campaign design and implementation processes are discussed in detail including the measurement protocol, calibration procedures and slant column retrieval settings. Strong emphasis was put on the careful alignment and synchronisation of the measurement systems, resulting in a unique set of measurements made under highly comparable air mass conditions.

The CINDI-2 data sets were investigated using a regression analysis of the slant columns measured by each instrument and for each of the target data products. The slope and intercept of the regression analysis respectively quantify the mean systematic bias and offset of the individual data sets against the selected reference (which is obtained from the median of either all data sets or a subset), and the rms error provides an estimate of the measurement noise or dispersion. These three criteria are examined and for each of the parameters and each of the data products, performance thresholds are set and applied to all the measurements. The approach presented here has been developed based on heritage from previous intercomparison exercises. It introduces a quantitative assessment of the consistency between all the participating instruments for the MAX-DOAS and zenith-sky DOAS techniques.

## 1 Introduction

Passive UV–visible spectroscopy using scattered sunlight as a light source provides one of the most effective methods for routine remote sensing of atmospheric trace gases from the ground. While zenith-sky observations have been used for several decades to monitor stratospheric gases such as NO$_2$, O$_3$, BrO and OClO (e.g. Noxon, 1975; Platt et al., 1979; Solomon et al., 1987; Pommereau and Goutail, 1988; Richter et al., 1999; Liley et al., 2000; Hendrick et al., 2011; Yela et al., 2017), measurements scanning the sky vertically at several elevation angles between horizon and zenith have been

established more recently. In addition to total columns, the MAX-DOAS (multi-axis differential optical absorption spectroscopy; Hönninger et al., 2004) technique also allows the derivation of vertically resolved information on a number of tropospheric species such as NO$_2$, formaldehyde (HCHO), BrO, glyoxal, IO, HONO and SO$_2$ (see, e.g., Hönninger and Platt, 2002; Wittrock et al., 2004; Heckel et al., 2005; Lee et al., 2008, 2009; Sinreich et al., 2010; Frieß et al., 2011; Hendrick et al., 2014; Prados-Roman et al., 2018) as well as aerosols (see, e.g., Wagner et al., 2004; Frieß et al., 2006; Clémer et al., 2010; Ortega et al., 2016). The number of MAX-DOAS instruments used worldwide has grown considerably in recent years notably in support of satellite validation (e.g. Wang et al., 2017a; Herman et al., 2018) and for urban pollution studies (e.g. Gratsea et al., 2016; Wang et al., 2017b), and this increase in the deployment of MAX-DOAS instrumentation for tropospheric observations, together with the diversity of the designs and operation protocols, has created the need for regular formal intercomparisons which should include as many different instruments as possible.

In 2005 and 2006, two field campaigns were held at Cabauw, the Netherlands, involving MAX-DOAS instruments as part of DANDELIONS (Dutch Aerosol and Nitrogen Dioxide Experiments for vaLIdation of OMI and SCIAMACHY). This project was dedicated to the validation of satellite NO$_2$ measurements by the Ozone Monitoring Instrument (OMI) and SCIAMACHY (Scanning Imaging Absorption SpectroMeter for Atmospheric CartographY) and aerosol measurements by OMI and the Advanced Along-Track Scanning Radiometer (AATSR) (Brinksma et al., 2008). This was followed by the first Cabauw Intercomparison campaign for Nitrogen Dioxide measuring Instruments (CINDI), which was organised in 2009 under the auspices of the European Space Agency (ESA), the Network for the Detection of Atmospheric Composition Change (NDACC), and the European Union (EU) FP6 Global Earth Observation and MONitoring (GEOMON) project. This effort resulted in the first successful large-scale intercomparison of both MAX-DOAS and zenith-sky ground-based remote sensors of NO$_2$ and O$_4$ slant columns (Roscoe et al., 2010). Data sets of NO$_2$, aerosols and other air pollution components observed during CINDI were documented in a number of peer-reviewed articles (Piters et al., 2012; Roscoe et al., 2010; Pinardi et al., 2013; Zieger et al., 2011; Irie et al., 2011; Frieß et al., 2016), providing an assessment of the performance of ground-based remote-sensing instruments for the observation of NO$_2$, HCHO and aerosol. Recommendations were issued regarding the operation and calibration of the instruments, the retrieval settings and the observation strategies for use in ground-based networks for air quality monitoring and satellite data validation. Several important findings were highlighted in view of preparing future campaigns, in particular (1) the need for accurate calibration and monitoring of the elevation angle of MAX-DOAS scanners and (2), for

intercomparison purposes, the importance of synchronising measurements in time and space very accurately. The lack of such a synchronisation was indeed regarded as being responsible for a large part of the scatter observed during CINDI (Roscoe et al., 2010), which limited the interpretation of the results.

Seven years after CINDI, a second campaign (CINDI-2) was undertaken at the same site (Cabauw Experimental Site for Atmospheric Research – CESAR) from 25 August until 7 October 2016. Its goal was to intercompare the new and extended generation of ground-based remote-sensing and in situ air quality instruments. The interest of ESA in such intercalibration activities is motivated by the ongoing development of several UV–visible space missions targeting air quality monitoring such as the Copernicus Sentinel 5 Precursor (S5P) satellite launched in October 2017 and the future Copernicus Sentinel 4 and 5 satellites. The validation and ongoing support of measurements from such space missions is essential and requires dedicated ground-truth measurement systems. Because tropospheric measurements from space-borne nadir UV–visible sensors show little or no vertical discrimination and inherently provide measurements of the total tropospheric amount, surface in situ measurements are generally unsuitable for such a validation effort. Instead, validation requires a technique that can deliver column-integrated and vertically resolved information on the key tropospheric species measured by satellite instruments such as NO$_2$, HCHO and SO$_2$ with a horizontal representativeness compatible with the resolution of space measurements (e.g. $3.5\,\mathrm{km} \times 7\,\mathrm{km}$ for S5P).

Hence, the specific goals of CINDI-2 were to support the creation of high-quality ground-based data sets as needed for long-term measurements, trend analysis and satellite data validation. To achieve this, it is essential to characterise the differences between a large number of MAX-DOAS and zenith-sky DOAS instruments and analysis methods and to assess the participating instruments in their ability to retrieve the same geophysical quantities (i.e. slant columns of NO$_2$, O$_4$, HCHO and O$_3$) when measured and processed in a controlled way (i.e. using a prescribed measurement protocol and retrieval settings). The design of CINDI-2 and the development of the measurement protocol, adhered to specifically during the official intercomparison phase, was based on the experience gained during the first CINDI in 2009 as well as more recent projects and campaigns such as the Multi-Axis Doas – Comparison campaign for Aerosols and Trace gases (MAD-CAT) in Mainz, Germany, in 2013 (e.g. Peters et al., 2017).

This paper is organised as follows. In Sect. 2, the campaign design is discussed including an overview of the participating groups and their instruments, and a discussion of the measurement protocol details. In Sect. 3, the results of the semi-blind slant column intercomparison are presented, and in Sect. 4, a systematic approach is proposed to quantitatively assess the performance of the participating instruments

for the different target trace gas data products. Section 5 provides recommendations for observation networks and future intercomparison campaigns and Sect. 6 summarises the campaign outcomes.

## 2 Intercomparison campaign design and measurement protocol

The CESAR site was accessible for the installation of the instruments from 25 August 2016 onwards, with the formal semi-blind intercomparison being held for 17 d from 12–28 September 2016. Here, we concentrate on this official intercomparison phase of CINDI-2, and measurements and results are discussed for this time period only. A general description of the overall campaign including a more detailed discussion of the CESAR site and all ancillary measurements can be found in Apituley et al. (2020). In short, the CESAR site at Cabauw is overall a rural site, with only a few pollution sources nearby, but the wider vicinity of Cabauw is densely populated, with the cities of Utrecht, Amsterdam, The Hague and Rotterdam less than 60 km away and a dense highway grid within 25 km, so that the site experiences recurring pollution events, e.g. such as from the daily morning and afternoon rush hours.

The MAX-DOAS instruments were also complemented with a suite of in situ, profiling and mobile observations, which are described in detail by Apituley et al. (2020). In particular, a long-path DOAS measuring near surface mixing ratios of $NO_2$ and HCHO but also a range of other species such as HONO and $SO_2$ (see, e.g., Merten et al., 2011, for a description of the technique) was operated at the CESAR site for the period of the campaign. Several mobile MAX-DOAS measurements were also made around Cabauw and between Rotterdam and Utrecht (e.g. Merlaud, 2013) in addition to the static observations. $NO_2$ profiles were measured with $NO_2$ sondes (Sluis et al., 2010) and lidar (e.g. Volten et al., 2009), as well as through in situ observations using the Cabauw meteorological tower. Extensive aerosol information was also gathered using Raman aerosol lidar and in situ samplers.

### 2.1 Instruments

Table 1 lists the groups and instruments that were included in the CINDI-2 semi-blind intercomparison, and an overview of the relevant instrumental details is given in Table 2. Among the 36 participating instruments, 17 were two-dimensional (2-D) MAX-DOAS systems allowing for scans in both elevation and azimuth, 16 were one-dimensional (1-D) MAX-DOAS systems performing elevation scans in one fixed azimuthal direction, 1 was an imaging DOAS instrument (Imaging MaPper for Atmospheric observaTions – IMPACT; Peters et al., 2019) for which only measurements in the common viewing direction were submitted, and the last 2 instruments were zenith-sky DOAS systems of the SAOZ (Système d'Analyse par Observation Zénithale) (Pommereau and Goutail, 1988) and most recent Mini-SAOZ version. The complete technical specifications for each instrument can be found in Sect. S3 of the Supplement.

Instruments have been sorted into different categories. Custom-built systems refer to instruments developed by scientific organisations for their own research activities. Other categories denote commercial systems of various types. Pandora instruments (Herman et al., 2009) are being developed at NASA/LuftBlick, commercialised by the SciGlob company and deployed as part of the Pandonia Global Network (PGN) (http://pandonia.net/, last access: 18 March 2020). EnviMes (now: SkySpec from Airyx GmbH (http://www.airyx.de, last access: 18 March 2020)) MAX-DOAS instruments (Lampel et al., 2015) have been recently commercialised based on expertise developed at the University of Heidelberg. Mini-DOAS instruments (e.g. Hönninger et al., 2004; Bobrowski, 2005) are produced in Germany by Hoffmann GmbH (http://www.hmm.de/, last access: 18 March 2020).

No particular guidelines were given concerning the spectral calibration of instruments, which means that participating groups were free to apply calibration steps of various levels of complexity. In addition to standard calibration procedures involving dark-current and electronic offset corrections, wavelength registration, and slit function determination, some groups performed more advanced pre-processing steps such as radiometric calibration, stray light and inter-pixel variability correction, or an explicit correction for detector response non-linearity, the latter being a known feature of Avantes spectrometers.

### 2.2 Campaign design

To allow for optimal synchronisation of the measurements, all the spectrometers participating in the semi-blind intercomparison exercise were installed in close proximity to each other on the remote-sensing site (RSS) of the CESAR station (see Fig. 1 and Apituley et al., 2020). To achieve this, mobile units (similar to shipping containers) were temporarily installed for the campaign period.

The rationale behind this setup was to arrange the instruments in such a way as to minimise ambiguity in air masses observed simultaneously by all spectrometers. This is essential for tropospheric $NO_2$ but also for aerosol and HCHO, since all these species can feature rapidly changing concentrations in both space and time. Considering the large number of systems that needed to be accommodated, two rows of containers were deployed with the bottom row being similar to the one deployed during the previous CINDI. This bottom row of containers was predominantly used to host the 1-D MAX-DOAS instruments and the two zenith-sky systems. A second row of containers was deployed on top of the first one, with the stacked double containers providing additional

**Table 1.** List of participating groups and corresponding instrument IDs in alphabetical order according to their acronym.

| Institute | Country | Acronym | Instrument ID |
|---|---|---|---|
| Anhui Institute of Optics and Fine Mechanics | China | AIOFM | aiofm-1 |
| A. M. Obukhov Institute of Atmospheric Physics, Russian Academy of Sciences, Moscow | Russia | AMOIAP | amoiap-2 |
| Aristotle University of Thessaloniki | Greece | AUTH | auth-3 |
| Royal Belgian Institute for Space Aeronomy | Belgium | BIRA-IASB | bira-4 |
| University of Natural Resources and Life Sciences, Vienna | Austria | BOKU | boku-6 |
| Belarusian State University | Belarus | BSU | bsu-5 |
| Chiba University | Japan | CHIBA | chiba-9 |
| China Meteorological Administration | China | CMA | cma-7, cma-8 |
| Spanish National Research Council | Spain | CSIC | csic-10 |
| University of Colorado | USA | CU-Boulder | cu-boulder-11, cu-boulder-12 |
| Deutsches Zentrum für Luft- und Raumfahrt/University of Science and Technology of China | Germany/China | DLR-USTC | dlrustc-13, dlrustc-14 |
| Indian Institute of Science Education and Research Mohali | India | IISERM | iiserm-16 |
| National Institute for Aerospace Technology | Spain | INTA | inta-17 |
| University of Bremen | Germany | IUP-Bremen | iupb-18, iupb-37 |
| University of Heidelberg | Germany | IUP-Heidelberg | iuph-19 |
| Royal Netherlands Meteorological Institute | The Netherlands | KNMI | knmi-21, knmi-22, knmi-23 |
| Laboratoire Atmosphères, Milieux, Observations Spatiales | France | LATMOS | latmos-24, latmos-25 |
| Ludwig-Maximilians-Universität München | Germany | LMU-MIM | lmumim-35 |
| LuftBlick Earth Observation Technologies | Austria | Luftblick | luftblick-26, luftblick-27, luftblick-260, luftblick-270 |
| Max Planck Institute for Chemistry, Mainz | Germany | MPIC | mpic-28 |
| NASA Goddard Space Flight Center | USA | NASA | nasa-31, nasa-32 |
| National Institute of Water and Atmospheric Research | New Zealand | NIWA | niwa-29, niwa-30 |
| National University of Sciences and Technology | Pakistan | NUST | nust-33 |
| University of Toronto | Canada | UTO | uto-36 |

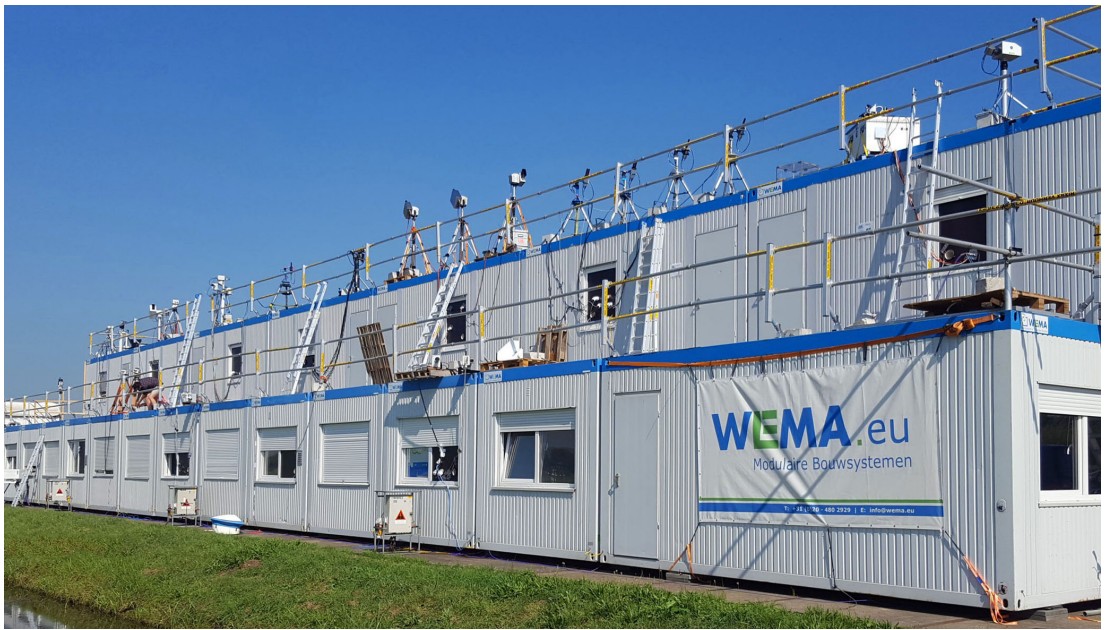

**Figure 1.** Picture of the CINDI-2 container layout at the main campaign site showing the organisation of the MAX-DOAS instruments on two superposed rows of mobile units (similar to shipping containers).

**Table 2.** Overview of the main characteristics of the instruments taking part in the semi-blind intercomparison campaign. The table lists the type, specific ID and model name for each participating instrument (columns 1–3). In columns 4–5, it also specifies whether instruments could take azimuthal scans (ASc) and/or be operated in direct-sun mode (DS). The spectral range, spectral resolution and field of view (FOV) are summarised in columns 6–8. Note that the FOV given in column 8 is the value provided as part of the instrument specification which may differ from the effective FOV shown in Fig. 6. Light coupling (column 9) denotes whether spectrometers were fed by means of optical fibres (F) or using a telescope or lens directly coupled to the entrance slit (D). The detector type is specified in column 10 as either a charge-coupled device (CCD) or a linear array (LinArr), and the detector temperature is listed in column 11.

| Instrument type | Instrument ID | Instrument name | ASc | DS | Spectral range (nm) | Spectral res. (nm) | FOV (°) | Light coupl. | Detector type | Detector $T$ (°C) |
|---|---|---|---|---|---|---|---|---|---|---|
| Custom-built MAX-DOAS | bira-4 | 2-D MAX-DOAS | y | y | 300–390/400–560 | 0.37/0.58 | 1.0/0.5 | F | CCD | −50/−50 |
| | iupb-18 | 2-D MAX-DOAS | y | n | 305–390/406–579 | 0.5/0.85 | 1.0 | F | CCD | −35/−30 |
| | boku-6 | 2-D MAX-DOAS | y | n | 419–553 | 0.8 | 0.8 | F | CCD | −60 |
| | cu-boulder-11 | 2-D MAX-DOAS | y | y | 325–470/430–680 | 0.7/1.2 | 0.7 | F | CCD | −30 |
| | cu-boulder-12 | 1-D MAX-DOAS | y | y | 300–465/380–490 | 0.8/0.5 | 0.7 | F | CCD | −30/0 |
| | inta-17 | RASAS-III | y | n | 420–540 | 0.55 | 1.0 | F | CCD | −30 |
| | mpic-28 | Tube MAX-DOAS | y | n | 305–464 | 0.6 | 0.7 | F | CCD | 20 |
| | niwa-30 | ACTON275 MAX-DOAS | n | n | 290–363/400–460 | 0.54 | 0.5 | F | CCD | −20 |
| | uto-36 | 2-D MAX-DOAS | y | y | 340–560 | 0.75 | 0.62 | F | CCD | −70 |
| | auth-3 | Phaethon | y | n | 300–450 | 0.4 | 1.0 | F | CCD | 5 |
| | aiofm-1 | 2-D MAX-DOAS | y | y | 290–380 | 0.4 | 0.2 | F | CCD | −30 |
| | chiba-9 | CHIBA-U MAX-DOAS | y | n | 310–515 | 0.4 | < 1 | F | CCD | ambient $T$ |
| | csic-10 | 1-D MAX-DOAS | n | n | 300–500 | 0.5 | 0.7 | F | CCD | −70 |
| | amoiap-2 | 2-port DOAS | n | n | 315–385/395–465/420–490 | 0.4 | 0.3 | F | CCD | −40 |
| | bsu-5 | MARSB | n | n | 300–500 | 0.4 | 1.0 | D | CCD | −40 |
| | iupb-37 | Imaging-DOAS | y | n | 420–500 | 0.8 | 1.2 | F | CCD | −30 |
| Pandora | knmi-23 | Pandora-1S | y | n | 290–530 | 0.6 | 1.5 | F | CCD | 20 |
| | luftblick-26 | Pandora-2S | y | n | 280–540 | 0.6 | 1.5 | F | CCD | 20 |
| | luftblick-260 | Pandora-2S | y | n | 380–900 | 1.1 | 1.5 | F | CCD | 20 |
| | luftblick-27 | Pandora-2S | y | n | 280–540 | 0.6 | 1.5 | F | CCD | 20 |
| | luftblick-270 | Pandora-2S | y | n | 380–900 | 1.1 | 1.5 | F | CCD | 20 |
| | nasa-31 | Pandora-1S | y | y | 280–540 | 0.6 | 1.6 | F | CCD | 20 |
| | nasa-32 | Pandora-1S | y | y | 280–540 | 0.6 | 1.6 | D | CCD | 20 |
| EnviMes | iupb-19 | 2-D EnviMes | y | y | 300–460/440–580 | 0.6/0.5 | < 0.5 | F | CCD | 20 |
| | dlrustc-13 | 1-D EnviMes | n | n | 300–460/450–600 | 0.6/0.6 | 0.4 | F | CCD | 20 |
| | dlrustc-14 | 1-D EnviMes | n | n | 300–460/450–600 | 0.6/0.6 | 0.4 | F | CCD | 20 |
| | niwa-29 | 1-D EnviMes | n | n | 305–460/410–550 | 0.6 | < 0.5 | F | CCD | 20 |
| | lmumim-35 | 2-D EnviMes | y | n | 300–460/450–600 | 0.6/0.9 | 0.4 | F | CCD | 20 |
| Mini-DOAS Hoffmann GmbH | cma-7 | Mini-DOAS-UV | n | n | 300–450 | 0.7 | 0.8 | F | LinArr | ambient $T$ |
| | cma-8 | Mini-DOAS-Vis | n | n | 400–710 | 1.6 | 0.8 | F | LinArr | ambient $T$ |
| | iiserm-16 | Mini-DOAS-UV | n | n | 316–466 | 0.7 | 0.7 | F | CCD | −10.4 |
| | knmi-21 | Mini-DOAS-UV | n | n | 290–443 | 0.6 | 0.45 | F | LinArr | −5 |
| | knmi-22 | Mini-DOAS-Vis | n | n | 400–600 | 0.5 | 0.4 | F | LinArr | −5 |
| | nust-33 | Mini-DOAS-UV | n | n | 320–465 | 0.7 | 1.2 | F | CCD | ambient $T$ |
| SAOZ | latmos-24 | SAOZ | n | n | 270–640 | 1.3 | 20 | D | LinArr | ambient $T$ |
| | latmos-25 | Mini-SAOZ | n | n | 270–820 | 0.7 | 8 | F | CCD | ambient $T$ |

height. All 2-D MAX-DOAS systems were installed on the roof of the top-level containers allowing for more flexibility on the azimuth scan settings and avoiding any risk of interference with the 1-D systems. All the 1-D MAX-DOAS instruments used the same azimuth viewing direction of 287° (i.e. approximately WNW, with north (N) being 0° and east (E) 90°, etc.) which was already used during the first CINDI since it provided an unobstructed view to the horizon. This direction was also one of the azimuth directions used by the 2-D MAX-DOAS systems (see also discussion of the measurement protocol in Sect. 2.4).

In Sect. 2.4–2.6, further procedures aiding the comparability of the MAX-DOAS measurements such as the overall measurement protocol, elevation angle calibrations and slant column retrieval settings are discussed in more detail. Prescribing these procedures as strictly as possible was highlighted as important during previous campaigns (see in particular Roscoe et al., 2010) and the campaign design of CINDI-2 focused on implementing such recommendations.

## 2.3 Semi-blind intercomparison

As in previous intercomparison campaigns of the same type (see, e.g., Vandaele et al., 2005; and Roscoe et al., 1999, 2010), a semi-blind intercomparison protocol was adopted. The CINDI-2 exercise had three key objectives: (1) to characterise the differences between a large number of measurement systems and approaches, (2) to discuss the performance of the various types of instruments and define a robust methodology for performance assessment, and (3) to provide guidelines to further harmonise the measurement settings and analysis methods. The adopted semi-blind intercomparison protocol was based on the following approach.

a. The data acquisition schedule applied by the participants was strictly prescribed to coordinate the timing and geometry of each individual measurement as exactly as possible, so that the same air mass could be measured by all instruments with good synchronisation.

b. For each data product, a set of retrieval settings and parameters was prescribed (see Appendix A). These were mandatory for participation in the semi-blind exercise. The data analysis software, however, was not prescribed and the different software types used by each institute are listed in Table 3.

c. All slant column data sets measured during the previous day were submitted to an independent campaign referee (Karin Kreher) and her assistant (Ermioni Dimitropoulou) every morning by 10:00 local time. At daily meetings in the afternoon (usually at 16:00), the results of the slant column comparison for measurements from the previous day were displayed anonymously, i.e. without any assignment to the different instruments. Basic analysis plots exploring the differences in the data sets measured during the previous days were shown and discussed.

d. The referee notified instrument representatives if there was an obvious problem with their submitted data set so that this issue could be addressed and, if possible, corrected for the remainder of the campaign.

e. After the formal campaign had finished, all participants had about 3 weeks to undertake the analysis according to the prescribed measurement and analysis protocol (see Sect. 2.4), and the final slant column data sets had to be submitted by 18 October 2016. After this date, any resubmissions were only accepted if the group could clearly state the reasons why the data set needed to be updated, e.g. if an error was found in the analysis and needed to be remedied. Further details on this process are given in Sect. 3.3 and Appendix B.

The semi-blind intercomparison exercise focused on a limited number of key data products of direct relevance for satellite validation and NDACC operational continuity. These data products are listed in Table 4. Depending on the specific characteristics of their instrumentation, participants were free to submit all or only a subset of the data products.

## 2.4 Measurement protocol

As discussed above, it was recognised in previous intercomparison campaigns (see in particular Roscoe et al., 2010) that the achievable level of agreement between MAX-DOAS sensors is often limited by imperfect co-location and a lack of synchronisation. This problem is especially critical for tropospheric NO$_2$ comparisons because of the large variability of this pollutant on very small scales. However, it is also relevant for other gases such as HCHO, O$_4$, SO$_2$ and glyoxal. For this reason, it was decided to co-locate all the MAX-DOAS instruments on the same observation platform (see Sect. 2.2) and additionally to impose a strict protocol on the timing of the spectral acquisition.

The baseline for all MAX-DOAS instruments was to point towards a fixed azimuth direction (287°) throughout the day. This direction was chosen because of the very close to obstruction-free line of sight towards the horizon. In addition, the 2-D MAX-DOAS instruments performed azimuthal scans simultaneously according to a strict measurement schedule. The scheme described below was designed to ensure the maximum of synchronisation between the same type of instruments (e.g. azimuthal scans by 2-D MAX-DOAS) but also between the different types of instruments (1-D and 2-D MAX-DOAS and zenith-sky DOAS). A distinction was made between twilight (morning and evening) and daytime conditions, for which separate data acquisition protocols were prescribed. According to the geometry of the solar position during the campaign, the daytime period (excluding twilight) was defined to be from 06:00 to 16:45 UTC

**Table 3.** Overview of analysis software used by each of the participating institutes.

| Data analysis software | Institute acronym |
|---|---|
| QDOAS | AUTH, BIRA-IASB, CSIC, CU-Boulder, LMU-MIM |
| QDOAS and WinDOAS | AIOFM, NUST |
| QDOAS and software developed in-house | UTO |
| DOASIS | DLR-USTC, IUP-Heidelberg |
| DOASIS and WinDOAS | IISERM, |
| DOASIS and software developed in-house (STRATO) | NIWA |
| WinDOAS | CMA, MPIC |
| WinDOAS and software developed in-house | BSU |
| Blick Software Suite | LuftBlick, NASA |
| Blick Software Suite and software developed in-house | KNMI |
| NLIN | BOKU, IUP-Bremen |
| LANA | INTA |
| SAOZ SAM v5.9 and Mini-SAOZ software developed in-house | LATMOS |
| JM2 (Japanese MAX-DOAS profile retrieval algorithm, version 2) | CHIBA |
| Andor Solis and software developed in-house | AMOIAP |

**Table 4.** Data products included in the semi-blind intercomparison exercise and wavelength intervals selected for the analysis. Performance limits on bias (deviation from unity slope), offset and rms of dSCD linear regressions are also listed for each of the eight data products.

| Data product | Spectral interval (nm) | Bias (%) | Offset (molec. cm$^{-2}$) | rms (molec. cm$^{-2}$) |
|---|---|---|---|---|
| NO2vis | 425–490 | 5 | $1.5 \times 10^{15}$ | $8.0 \times 10^{15}$ |
| NO2visSmall | 411–445 | 5 | $1.5 \times 10^{15}$ | $8.0 \times 10^{15}$ |
| NO2uv | 338–370 | 6 | $2.0 \times 10^{15}$ | $1.0 \times 10^{16}$ |
| O4vis* | 425–490 | 5 | $0.7 \times 10^{42}$ | $3.0 \times 10^{42}$ |
| O4uv* | 338–370 | 6 | $0.8 \times 10^{42}$ | $3.0 \times 10^{42}$ |
| HCHO | 336.5–359 | 10 | $5.0 \times 10^{15}$ | $1.0 \times 10^{16}$ |
| O3vis | 450–520 | 4 | $0.2 \times 10^{18}$ | $1.0 \times 10^{18}$ |
| O3uv | 320–340 | 4 | $1.0 \times 10^{18}$ | $4.0 \times 10^{18}$ |

* Note: the units for O$_4$ are molec.$^2$ cm$^{-5}$.

with 06:00 UTC corresponding to a solar zenith angle (SZA) of approximately 83–87° and 16:45 UTC to an SZA of approximately 76–82°, depending on the exact date during the campaign.

To allow for an NDACC-type intercomparison of stratospheric measurements (e.g. Vandaele et al., 2005), zenith-sky twilight observations were also performed. The acquisition scheme for the dawn observations prescribed 39 measurements with a duration of 3 min each (integration time: 170 s; overhead time: 10 s), starting at 04:00:00 UTC and ending at 05:57:00 UTC. This sequence was followed by a 180 s (3 min) interval allowing for a transition to the MAX-DOAS mode of which the first scans started at 06:00:00 UTC. For measurements at dusk, 40 acquisitions were recorded with a duration of 180 s each starting at 16:45:00 UTC and ending at 18:45:00 UTC.

During daytime, the acquisition scheme for MAX-DOAS and zenith-sky systems included four sequences of 15 min per hourly slot starting at 06:00:00 UTC. Individual acquisitions (at one given angle) were set to 1 min long in all cases. For 1-D systems, the pointing azimuth direction was set to 287° with elevation angles of 1, 2, 3, 4, 5, 6, 8, 15, 30 and 90°. For 2-D systems, the azimuth angles 45, 95, 135, 195, 245 and 355° were successively sampled in addition to the reference angle of 287°. In each azimuthal direction, four elevation angles (1, 3, 5, 15°) were scanned except for the reference azimuth of 287°, where the same elevations as prescribed for the 1-D MAX-DOAS systems were used. One zenith reference spectrum was recorded every 15 min, and for 2-D systems or instruments equipped with a sun tracker, almucantar scans and/or direct-sun measurements were performed between the 10th and 15th minute of the sequence. For zenith-sky instruments, 1 min long acquisitions were performed during the whole day from 06:00:00 to 16:44:00 UTC.

Figure 2a provides an overview of the number of days each instrument was on duty during the intercomparison period. It also illustrates (Fig. 2b) the accuracy with which the different groups were able to match the imposed measurement protocol. As can be seen, the instruments were in operation most of the time during the 17 d of the semi-blind period and most of them were able to follow the schedule to better than 1 min. In comparison to past campaigns, the level of synchronisation was clearly improved, which significantly reduced the need for smoothing or interpolating data in time. As a result, the impact of the atmospheric variability on the data comparisons could be reduced considerably but not completely eliminated (see Sect. 3.7).

As discussed above, the measurement procedure was strict, but in spite of this comprehensive protocol, there was still some freedom left on how to implement details of the ac-

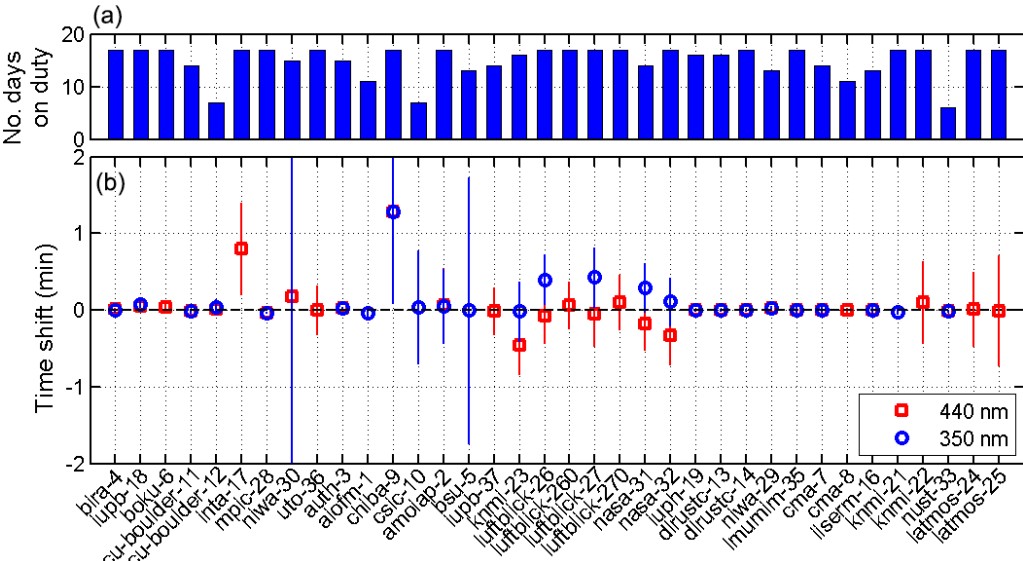

**Figure 2. (a)** The number of days when instruments were on duty during the 17 d intercomparison period. **(b)** The mean and standard deviation of the time deviations (in decimal minutes) observed in the MAX-DOAS measurements as reported by each participating group with respect to the measurement schedule defined for the campaign. Note that the instruments are listed in order of how they are categorised, and this is further explained in Sect. 4.

quisitions. For example, for managing the acquisition time, most groups decided to move the telescope and gather the spectra within the prescribed 1 min time period, while the National Institute for Aerospace Technology (INTA) (inta-17) gathered spectra for 1 min and then moved the telescope. As a result, a time shift was accumulated when compared to other groups (see Fig. 2). Chiba-9 also shows a noticeable time shift due to constraints in the acquisition software that prevented the strict implementation of the protocol. In the case of niwa-30, the large time shift in the UV was due to instrument-imposed alternating between measurements in the visible and UV wavelength regions (hence only one spectral range could be synchronised with the protocol).

Likewise, it must be noted that Pandora instruments also take separate measurements for the visible and the UV range, where a blocking filter is inserted into the optical path for the UV measurements in order to reduce spectral stray light. Therefore, a compromise had to be found in the time synchronisation bracketing the requested measurement time. This is the reason for the systematic offsets for Pandoras in Fig. 2b. Another consequence of this was that the total measurement time of Pandora instruments was about half the time of the other participating instruments, which affects the noise levels for Pandoras described throughout this paper to some extent.

### 2.5 Calibration of the MAX-DOAS elevation scans

Because of the importance of the elevation pointing accuracy for MAX-DOAS measurements at low elevation and as recommended after the first CINDI (Roscoe et al., 2010), different calibration tests involving all the participating instruments were undertaken during both the warm-up and semi-blind intercomparison phases. Three different approaches were used.

– On several evenings, MPIC (Max Planck Institute for Chemistry) installed an Opel car 1999 xenon lamp with a 17 cm diameter lens at a distance of 1280 m from the measurement site (angular lamp extension ~ 0.008°) in the main viewing azimuth direction (287°) of the MAX-DOAS instruments. It served as a common light source at long distance, and MAX-DOAS instruments recorded downward and upward scan spectra pointing towards the lamp.

– A white stripe on a black target at known elevation close to the instruments was scanned.

– Intensities were measured regularly during horizon scans (see Sect. 3.2 for details).

Additional calibration measurements using a near-distance lamp placed a few metres away from instruments were also performed by IUP-Heidelberg and several other groups. Overall, these calibration procedures allowed the pointing accuracy of the different instruments and their stability during the campaign to be fully characterised (see Donner et al., 2020). As such they played an important role for the interpretation of the semi-blind intercomparison results (see Sect. 3.7).

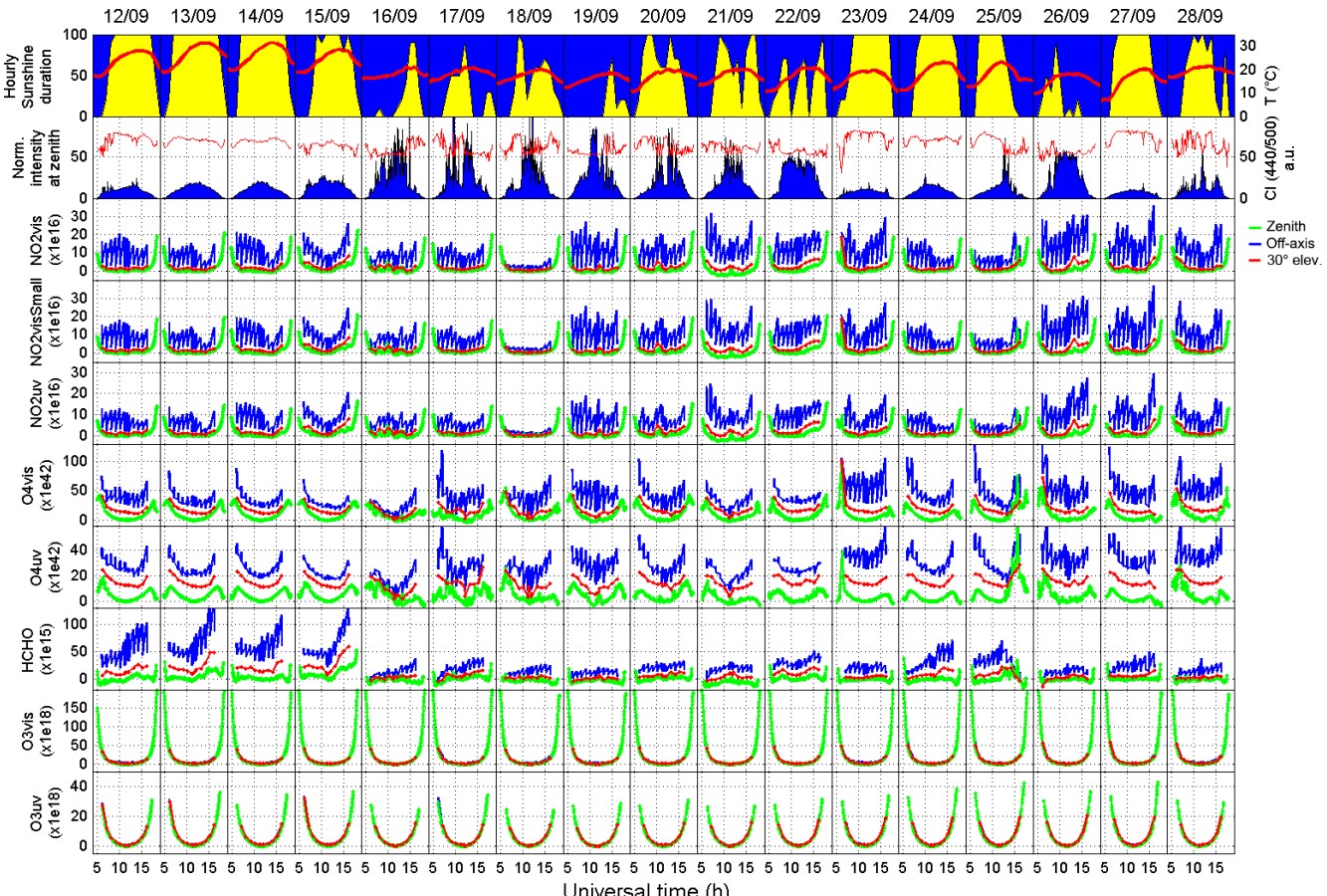

**Figure 3.** Hourly sunshine duration (yellow area) and temperature at the surface (red line) during the intensive campaign (topmost row), the intensity measured in the zenith and the colour index (second row from top), and the variability of the various trace gas slant column measurements performed during the semi-blind intercomparison exercise (all other rows). Slant column data measured at the main azimuth viewing direction (287°) with the IUP-Bremen instrument (iupb-18) are shown. Green lines and symbols represent zenith-sky measurements, red lines and symbols off-axis data at 30° elevation, and blue lines off-axis measurements up to 15° elevation.

## 2.6  Slant column retrieval settings

To minimise the sources of difference between measurements, a set of common retrieval settings and parameters was prescribed ahead of the campaign. The use of these settings was mandatory for participation in the semi-blind exercise. The detailed spectral retrieval settings imposed for each data product referenced in Table 1 are given in Appendix A. These settings were based on the NDACC protocol for UV–vis measurements (http://www.ndaccdemo.org/data/protocols, last access: 19 March 2020) as well as results from the first CINDI (e.g. Pinardi et al., 2013), MAD-CAT (http://joseba.mpch-mainz.mpg.de/mad_analysis.htm, last access: 19 March 2020) and the QA4ECV project (http://www.qa4ecv.eu/, last access: 19 March 2020). Although not necessarily optimal, they represent a common baseline applicable to all data sets in a consistent way. Concerning the choice of the Fraunhofer reference spectrum, daily reference spectra obtained from the mean of all zenith-sky spectra acquired between 11:30:00 and 11:41:00 UTC were used. Slant columns retrieved against this reference spectrum are hereinafter referred to as differential slant column densities (dSCDs).

Note that additional retrievals were also performed using sequential reference spectra (zenith-sky observations taken close to the time of the respective horizon measurements). These data were, however, not included in the formal semi-blind intercomparison since they essentially lead to similar comparison results as the analyses using daily reference spectra. They were also not available from all groups. Moreover, the use of daily reference spectra presents the advantage of being directly applicable to twilight measurements and provides a better test of the instrumental stability over several hours of operation. As already noted in Sect. 2.1, the determination of the instrumental slit function and its eventual wavelength dependence was the responsibility of the participating groups.

## 3 Semi-blind intercomparison results

### 3.1 Overview of slant column measurements and meteorological conditions

The meteorological conditions during CINDI-2 were exceptionally favourable for the location and season. The uppermost row of Fig. 3 shows the hourly sunshine duration and surface temperature records for the whole semi-blind intercomparison period (for more details, see Apituley et al., 2020). The first 4 d of the semi-blind phase were characterised by a clear sky with some haze in the morning and very high air temperatures for the season ($> 30\,°C$), allowing for efficient formaldehyde production. The next 7 d were cloudier with lower temperatures. The last 6 d of the semi-blind intercomparison exercise were also characterised by mostly clear sky or occasionally broken cloud conditions.

All other panels of Fig. 3 display the time variation in each of the dSCD data products included in the intercomparison, as measured by the IUP-Bremen instrument, which had excellent data coverage throughout the campaign duration. Green lines represent zenith-sky measurements, red lines off-axis data at 30° elevation, and blue lines off-axis measurements up to 15° elevation. Results show a large variability of the NO₂, O₄ and HCHO tropospheric columns while ozone data display the expected regular diurnal pattern mainly due to the variation in the stratospheric light path during the ascent and descent of the sun. Due to the unusually favourable weather conditions, higher than expected values were observed for tropospheric HCHO while tropospheric NO₂ was at its lowest during the first Sunday (18 September) of the intercomparison campaign. The variability of the tropospheric trace gas content and the exceptionally large number of clear-sky sunny conditions were ideal for comparison purposes.

### 3.2 Horizon scans

Horizon scans, which consist of measuring the change in intensity when scanning the sky radiance across the horizon line, were systematically performed every day at noon during the semi-blind intercomparison period. Although difficult to calibrate absolutely because the horizon is generally not free of obstacles (e.g. trees, buildings or terrain height fluctuations), they provide a simple and valuable technique for monitoring the elevation pointing stability of MAX-DOAS instruments. Figure 4 shows an example of the variation in the intensity at 440 nm, as reported by the IUP-Bremen instrument (blue circles). Considering that the intensity measured as a function of the elevation angle yields the integral over the telescope's point spread function, measurements were fitted using an error function (Gaussian integral) according to Eq. (1):

$$S = A\left[\mathrm{erf}\left(\frac{x - x_0}{B}\right) + 1\right] + C\,(x - x_0) + D, \qquad (1)$$

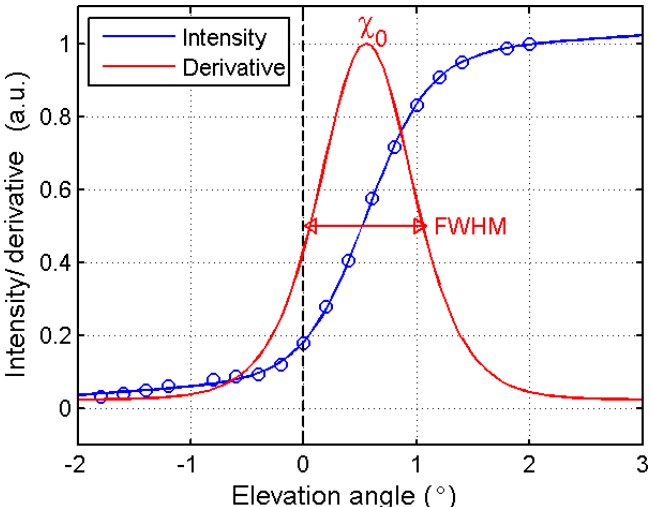

**Figure 4.** Horizon scans measured by IUP-Bremen on 14 September 2016 in the visible wavelength range. The blue circles display the intensity at 440 nm plotted as a function of the elevation angle reported by the instrument. Measured points are fitted by least-squares minimisation using an error function (blue line) allowing to estimate the horizon elevation ($\chi_0$) and effective field of view (FWHM) (see Sect. 3.2). The corresponding Gaussian curve (analytical derivative of the fitted blue curve) is represented in red.

where $x$ is the elevation angle and $A$, $B$, $C$ and $D$ are fitting parameters. The centre ($x_0$), also fitted, provides a measure of the horizon elevation.

The analytic derivative of Eq. (1) is a Gaussian curve of which the full width at half maximum (FWHM) is given by

$$\mathrm{FWHM} = 2\sqrt{\ln(2)}\,B. \qquad (2)$$

We used this quantity to estimate the effective field of view (FOV) of the instrument (see Fig. 4, red line).

Applying this fitting methodology, horizon scans delivered daily by each group were systematically analysed. Figure 5 presents an overview of the time evolution of the horizon elevation derived from each instrument (and their median values represented by red lines), all of them being measured in the visible wavelength range except for knmi-21. The same analysis was also performed at UV wavelengths. A summary of the resulting median and $1\sigma$ standard deviation FOV derived from each instrument is presented in Fig. 6.

The time series of horizon scans provide a useful assessment of the stability and precision of the elevation pointing devices used by the different instruments. In some cases, horizon scans allowed the identification of calibration biases, which could then be addressed by the instrument teams and corrected straight away. This is in particular the case for the dlrustc-13 and dlrustc-14 instruments. Considering the effective field of view (FOV), a large variability between the instruments was identified. This generally reflects differences in the optical design of the different systems. However, horizon scans can also be influenced by atmospheric conditions

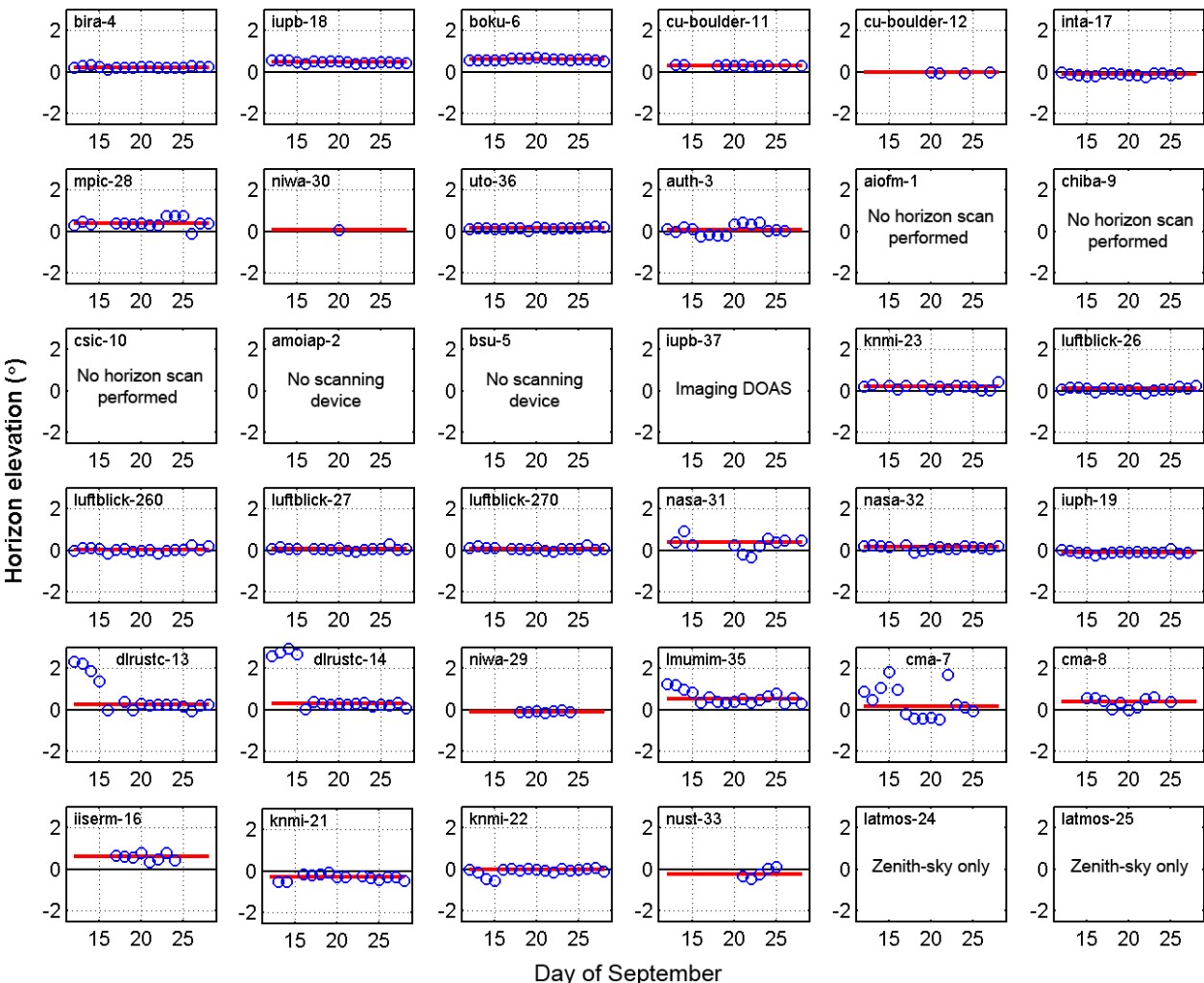

**Figure 5.** Time series of horizon elevation values (blue circles) derived from daily horizon scans performed with each instrument during the intercomparison period in the visible wavelength range (except for knmi-21). When no data are available for the horizon scan analysis, a short explanation is given. The red lines indicate the median values.

and by perturbations of the light intensity at the horizon (e.g. due to fog, high aerosol loads or refraction at temperature inversions). Nevertheless, it is striking to note in Fig. 6 that horizon elevations tend to be systematically higher at visible wavelengths than at UV ones. Likewise, FOVs measured in the UV tend to be wider than in the visible range. This variation is larger than expected from typical chromatic aberration effects in telescope lenses. The reason for this behaviour is not fully understood, but it is likely related to the wavelength dependence of the surface albedo, which may affect the horizon scan fitting process (for more details, see Donner et al., 2020).

### 3.3 History of slant column data set revisions

As described in Sect. 2.3, semi-blind dSCD data sets had to be submitted by 18 October 2016, i.e. 3 weeks after the end of the formal intercomparison period. However, resubmissions were accepted after this date when a clear justification was provided for the change. The main motivation for accepting late revisions was to remedy well-identified mistakes. Details of the submitted revisions, including justifications for the changes and corresponding dates, are given in Appendix B.

### 3.4 Pre-processing of the slant column data

Before further processing, the dSCD measurements from all groups were checked to remove unphysical values and obvious outliers. For this purpose, the following filters were applied: (1) dSCD data exceeding 10 times the daily median values from the instrument were excluded, and (2) data points with a fitting rms exceeding 4 times the daily median rms were removed.

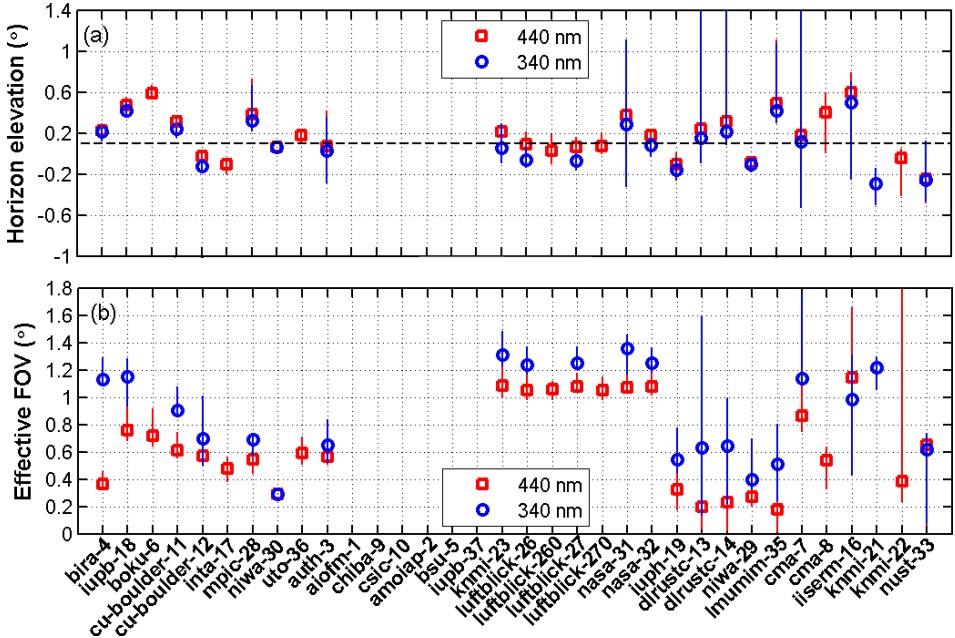

**Figure 6.** Summary of the average horizon elevation **(a)** and of the field of view **(b)**, resulting from the horizon scans performed at 340 and 440 nm. Symbols represent median values and the vertical bars 10th and 90th percentiles.

In addition, the results from the horizon scan analysis (see Sect. 3.2) were used to readjust the elevation angle of instruments presenting absolute elevation offsets larger than 1.5°. This correction was performed assuming a reference horizon elevation of 0.1°, as determined independently using lamp measurements performed at night combined with an analysis of terrain height variations (Donner et al., 2020). The impact of this angular correction is illustrated in Fig. 7 for NO₂ dSCD measurements, which are here represented in terms of their relative difference with respect to median values from a selection of the participating instruments (for more details, see Sect. 3.5 and Fig. 8). As can be seen, the large biases observed during the first few days of the campaign for some instruments were due to systematic mispointing effects well compensated for by the correction. The impact of the correction is largest for NO₂, but it is also significant for other tropospheric species, in particular O₄. This again stresses the importance of accurately calibrating the elevation scanner of MAX-DOAS instruments.

## 3.5 Determination of reference comparison data sets

As in previous campaigns, the intercomparison of dSCD measurements was based on pre-selected reference data sets. In CINDI-2, these were based on the calculation of median dSCDs obtained from a selection of measurements presenting an acceptable agreement. Here, the selection of the reference groups, different for each data product, was performed after an initial regression analysis using the median of all data as reference. Only groups satisfying the performance crite-

rion for the regression slopes were retained (see Sect. 4 and Table 4 for more details). The data sets included in the median references are displayed in Fig. 8 for both MAX-DOAS and zenith-sky twilight data products. In the particular case of HCHO, the selection was performed through visual inspection of the dSCD comparisons. Only data sets displaying consistent behaviour at 30° elevation (the angle generally used to retrieve first-guess total tropospheric columns using the geometrical approximation; see Hönninger and Platt, 2002) were retained for building the reference. This can be appreciated in Fig. 9 where time series of the HCHO dSCDs measured by each group are compared to the reference values. As can be seen, many data sets display noisy and/or unphysical negative values and only the four selected groups (bira-4, iupb-18, mpic-28 and niwa-29) present mutually consistent values. Note that a similar approach was used for the selection of the HCHO dSCD reference in Pinardi et al. (2013).

## 3.6 Initial assessment of the overall agreement between measurement data sets

Tables 5 and 6 show the mean relative differences (in percent) from the reference dSCDs and their first σ standard deviation for all participating instruments and, respectively, for all MAX-DOAS products and for all zenith-sky DOAS products. Extreme outliers (values exceeding percentile 97) are excluded from the analysis as well as MAX-DOAS ozone measurements since these show very small off-axis enhancements (see Fig. 8). Both tables provide an overall initial as-

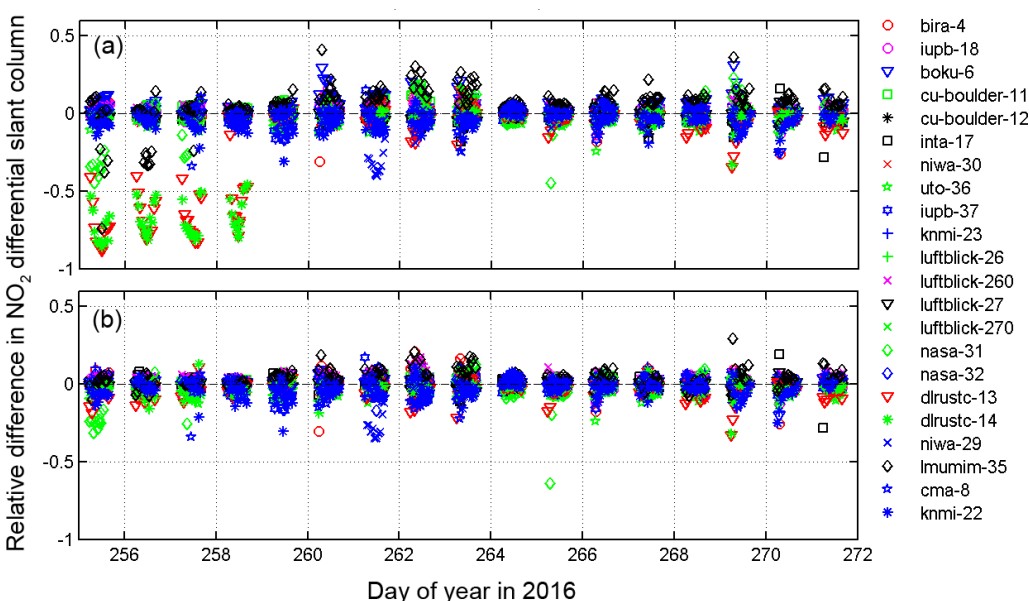

**Figure 7.** Relative differences of NO$_2$ dSCDs (in the visible wavelength region) with respect to the median from all instruments measured during the whole semi-blind intercomparison phase for the 287° azimuthal direction and 1° elevation angle. **(a)** Results before correction for elevation offsets; **(b)** same results after correction for elevation offsets derived from horizon scans. Colours and symbols represent different instruments.

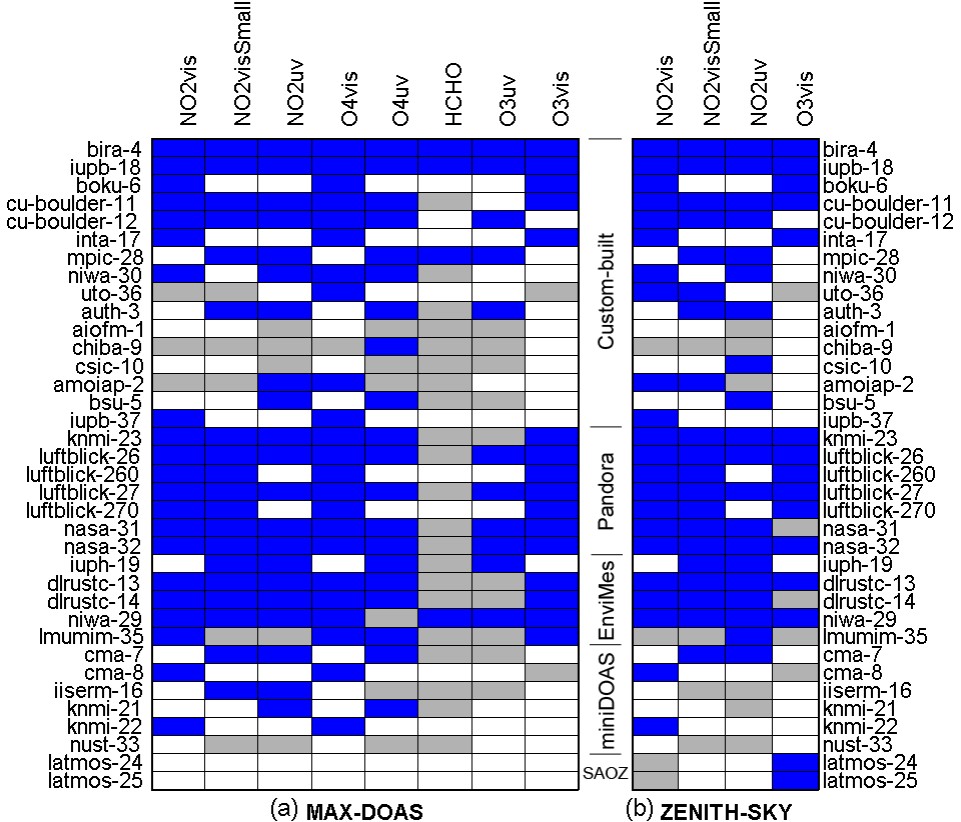

**Figure 8.** Instrument data sets selected to build the median MAX-DOAS reference **(a)** and zenith-sky **(b)** data sets. Blue marks the data sets included in the median,s while grey marks the data sets not included and white the ones not available. Note that the instruments are grouped according to their specific design as custom-built, Pandora, EnviMes, mini-DOAS or SAOZ.

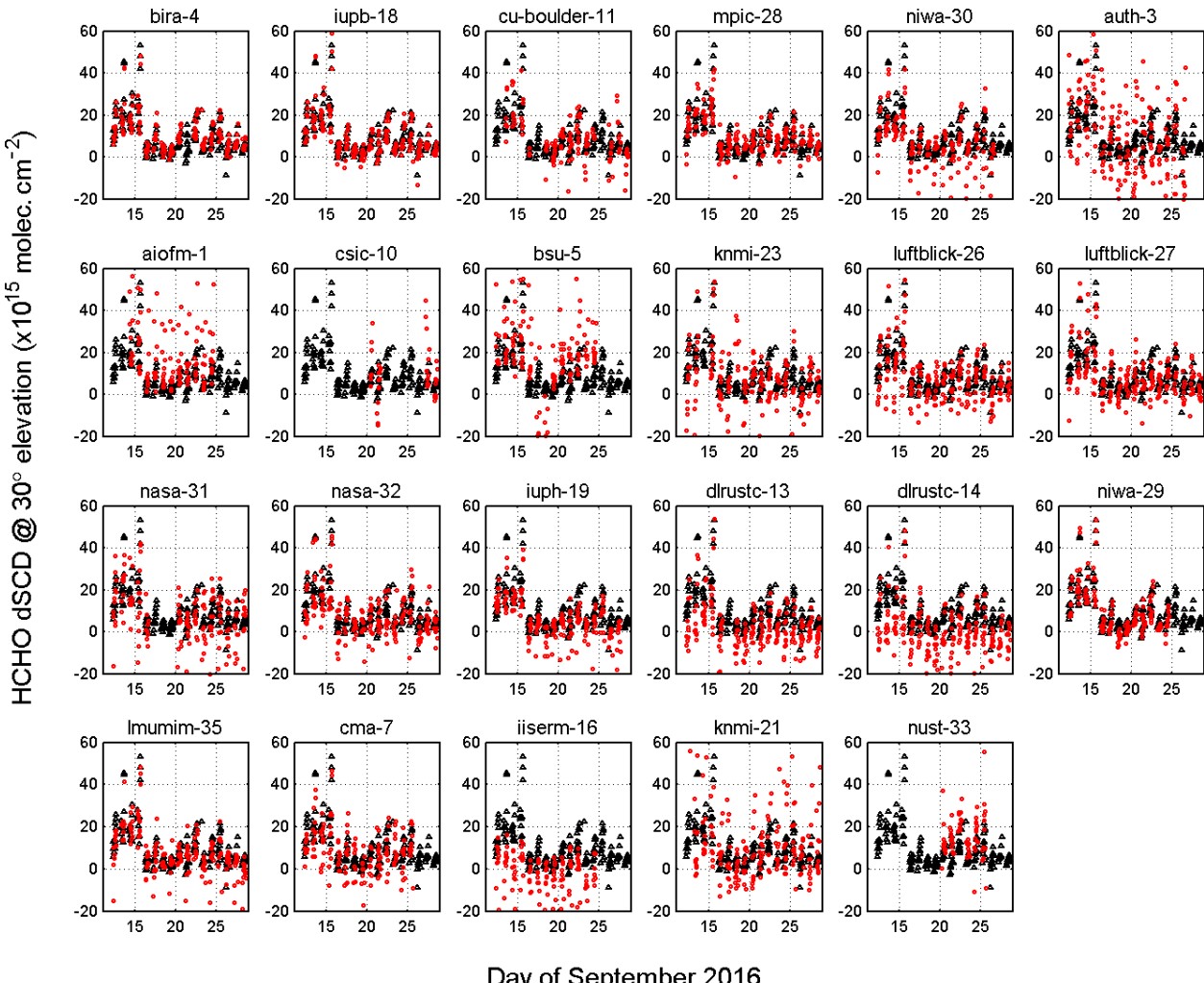

**Figure 9.** Comparison of HCHO dSCDs retrieved by each group at 30° elevation (red dots), and median values (black triangles). Only the four data sets (bira-4, iupb-18, mpic-28 and niwa-29) showing consistent values and a comparatively low noise level were selected for the calculation of the HCHO median.

sessment of the intercomparison results indicating that for most data products (except HCHO), instruments generally agree within a few percent for the most relevant range of elevation angles of 1–10° for MAX-DOAS data and for an SZA of 80–93° for zenith-sky twilight data. One can also see that the overall agreement between instruments is better in the visible than in the UV spectral range.

For HCHO (last two columns of Table 5), the differences between the instruments are comparatively larger and, in some cases, extreme. However, restricting the analysis to the first 4 d of the measurement campaign (when the air temperature was warmer and the HCHO dSCDs higher) reduces discrepancies significantly, and, although a higher spread remains compared to any of the other products, one can conclude that under such favourable conditions a large number of the participating instruments provide consistent HCHO dSCD measurements. For amoiap-2, however, the instrument

was operated in different modes during different time periods with some modes being more advantageous for the HCHO data analysis than others. The group found that when only HCHO data acquired during the optimal time period are used, the mean relative difference is substantially lower (approximately −16 %). More details on the instrument and the different modes are provided in Borovski et al. (2017a, b).

The last row of Tables 5 and 6 shows the median values from the table entries for each column. The median of the differences is by construction close to zero (but not exactly zero since the median reference values are derived from a selected subset of the participating instruments), while the median of the standard deviations provides an estimate of the most probable size of the deviations against the reference. For example, the median value for zenith-sky DOAS NO2uv shows the highest deviation from the reference when compared to the other zenith-sky DOAS products. For the MAX-

**Table 5.** Mean relative difference from the reference and standard deviation (in percent) for all participating instruments and MAX-DOAS data products (apart from ozone). The last column provides the values for HCHO when only considering measurements made during the first 4 d of the campaign period (12–15 September 2016).

| Instrument ID | NO2vis | NO2visSmall | NO2uv | O4vis | O4uv | HCHO | HCHO (12–15 Sep) |
|---|---|---|---|---|---|---|---|
| bira-4 | −0.0 (2.0) | 0.7 (2.0) | 1.7 (2.1) | 0.6 (2.0) | 1.0 (1.7) | 5.2 (6.9) | 1.0 (2.9) |
| iupb-18 | −2.2 (2.7) | −1.2 (2.4) | 0.1 (2.2) | −0.7 (2.2) | −1.2 (2.5) | −2.9 (6.4) | 0.0 (3.6) |
| boku-6 | 0.7 (2.6) | – | – | 0.3 (2.0) | – | – | – |
| cu-boulder-11 | 0.9 (4.9) | −1.8 (4.3) | −3.7 (5.1) | −0.7 (3.2) | −0.4 (3.3) | −19.9 (32.0) | −7.1 (11.7) |
| cu-boulder-12 | −3.9 (1.5) | −0.6 (1.6) | −0.6 (2.9) | −0.7 (1.6) | −0.2 (4.7) | – | – |
| inta-17 | 0.7 (2.6) | – | – | −0.2 (2.6) | – | – | – |
| mpic-28 | – | 1.4 (2.1) | 3.4 (3.3) | – | 0.9 (2.2) | −0.2 (14.5) | −4.0 (5.4) |
| niwa-30 | −2.6 (2.3) | – | −0.2 (10.0) | −0.1 (2.5) | 1.1 (6.5) | −24.5 (36.1) | −11.5 (7.7) |
| uto-36 | −6.4 (3.2) | −5.0 (3.1) | – | −3.6 (3.1) | – | – | – |
| auth-3 | – | −2.4 (3.4) | −3.4 (8.2) | – | 0.5 (8.5) | 7.9 (62.1) | 16.3 (26.3) |
| aiofm-1 | – | – | −15.8 (5.3) | – | −7.3 (5.1) | 18.2 (54.7) | −0.2 (16.3) |
| chiba-9 | −2.3 (3.4) | −1.3 (3.6) | 1.0 (4.0) | 6.5 (6.8) | 10.6 (4.1) | 0.1 (24.0) | −2.6 (13.3) |
| csic-10 | – | – | −17.7 (12.5) | – | 0.5 (8.4) | −131.5 (164.8) | – |
| amoiap-2 | −7.3 (3.3) | −7.9 (3.2) | −6.3 (9.9) | −0.8 (8.5) | −10.7 (8.0) | −70.5 (80.0) | −31.7 (12.1) |
| bsu-5 | – | – | −6.5 (6.5) | – | −5.0 (5.1) | 33.3 (90.5) | 13.2 (22.9) |
| iupb-37 | 3.3 (6.8) | – | – | -4.2 (7.0) | – | – | – |
| knmi-23 | 1.9 (2.3) | 2.8 (2.3) | 3.3 (6.8) | 1.3 (1.5) | 4.2 (4.2) | −12.3 (47.1) | −12.2 (17.9) |
| luftblick-26 | −0.4 (1.4) | −0.4 (1.3) | 0.6 (2.6) | −0.0 (1.3) | 0.6 (3.0) | −17.6 (32.5) | −11.9 (16.7) |
| luftblick-260 | 3.4 (2.1) | 2.8 (2.3) | – | −0.3 (1.5) | – | – | – |
| luftblick-27 | −1.3 (1.8) | −1.0 (1.6) | −0.5 (2.8) | 0.8 (1.4) | −1.0 (2.7) | −12.6 (28.0) | −9.0 (13.4) |
| luftblick-270 | −0.5 (1.7) | 0.7 (2.0) | – | −0.6 (1.3) | – | – | – |
| nasa-31 | 1.1 (6.2) | 1.0 (5.9) | 1.2 (5.7) | −0.1 (4.2) | −1.0 (5.1) | −21.5 (38.0) | −11.4 (15.7) |
| nasa-32 | 0.5 (1.7) | 0.2 (1.7) | −0.2 (3.0) | 1.0 (1.5) | −0.5 (3.1) | −10.6 (30.6) | −7.4 (9.6) |
| iuph-19 | – | −2.1 (3.0) | −1.0 (3.2) | – | −1.2 (3.0) | −32.1 (28.8) | −14.2 (7.9) |
| dlrustc-13 | −3.9 (3.7) | −3.1 (3.5) | −4.2 (3.8) | −3.1 (2.4) | 0.8 (2.3) | −42.6 (42.0) | −14.1 (8.1) |
| dlrustc-14 | −1.3 (3.0) | −0.4 (2.7) | −0.1 (2.7) | −1.5 (2.3) | 1.7 (2.0) | −57.5 (60.0) | −17.7 (9.9) |
| niwa-29 | −6.5 (12.0) | −5.1 (13.3) | −4.0 (14.8) | −0.2 (4.0) | 3.8 (6.2) | −10.5 (15.8) | – |
| lmumim-35 | 2.1 (4.4) | 1.2 (4.1) | −0.4 (3.7) | 7.1 (7.8) | −3.9 (3.0) | −9.0 (22.5) | −8.5 (8.3) |
| cma-7 | – | −1.5 (5.4) | −2.1 (5.4) | – | 1.7 (5.4) | −26.2 (35.5) | −20.7 (13.8) |
| cma-8 | −4.0 (4.1) | – | – | 0.7 (7.8) | – | – | – |
| iiserm-16 | – | 1.2 (5.0) | −0.1 (8.8) | – | 8.7 (7.0) | −111.5 (80.1) | −59.1 (24.1) |
| knmi-21 | – | – | −4.6 (5.0) | – | 2.7 (4.4) | 4.9 (60.0) | 0.4 (17.6) |
| knmi-22 | −1.5 (4.9) | – | – | −2.5 (4.6) | – | – | – |
| nust-33 | – | 6.7 (6.1) | 4.3 (9.2) | – | −22.6 (6.8) | 48.3 (73.7) | – |
| latmos-24 | – | – | – | – | – | – | – |
| latmos-25 | – | – | – | – | – | – | – |
| Median from all instruments | −0.9 (2.8) | −0.5 (3.1) | −0.4 (5.1) | −0.2 (2.4) | 0.5 (4.3) | −12.3 (36.1) | −8.7 (12.7) |

DOAS data products, as expected, HCHO shows by far the highest deviation.

### 3.7 Regression analysis

The approach adopted for CINDI-2 follows from previous exercises, in particular CINDI (Roscoe et al., 2010) and previous NDACC intercomparisons (Vandaele et al., 2005; Roscoe et al., 1999). It is based on the systematic analysis of regression plots between individual measurements and corresponding median reference values (see Sect. 3.5). Assuming negligible uncertainties in the reference dSCDs, we use a simple linear least-squares regression method weighted by reported dSCD uncertainties. Owing to the strict measurement protocol imposed for the campaign, most measurement points could be compared one-to-one without the need for further interpolation or averaging. When interpolation was necessary, a simple linear procedure was used to bring measurements in line with the campaign protocol (see Sect. 2.4). This implies that, in comparison to previous similar exercises, sampling and mismatch errors (air mass co-location errors) could be reduced considerably, so that comparison noise and biases should more accurately reflect the intrinsic instrumental performances. This question is further investigated below.

Linear correlation plots between the dSCDs for each instrument and the median value of all the measurements were systematically generated for the complete semi-blind intercomparison time period for each data product and for each elevation angle and azimuth viewing direction. This allowed determining, e.g., whether a specific issue arose from particular observation geometries for one or several instruments.

**Table 6.** Mean relative difference from the reference and standard deviation (in percent) for all participating instruments and zenith-sky DOAS data products.

| Instrument ID | NO2vis | NO2visSmall | NO2uv | O3vis |
|---|---|---|---|---|
| bira-4 | 0.4 (1.0) | 0.5 (1.2) | 0.9 (2.4) | 0.2 (1.0) |
| iupb-18 | 0.8 (1.1) | 0.8 (1.4) | 4.1 (3.0) | 0.2 (0.4) |
| boku-6 | 2.0 (1.0) | – | – | 0.7 (0.7) |
| cu-boulder-11 | 3.3 (2.7) | 1.3 (2.4) | −3.6 (7.8) | 0.5 (1.1) |
| cu-boulder-12 | −0.6 (2.2) | −0.2 (3.1) | −16.5 (21.5) | – |
| inta-17 | 1.4 (1.6) | – | – | −0.5 (0.7) |
| mpic-28 | – | 0.5 (3.1) | 6.3 (6.1) | – |
| niwa-30 | −0.1 (2.8) | – | 1.7 (14.4) | – |
| uto-36 | −1.0 (3.4) | −1.6 (2.8) | – | −6.7 (2.4) |
| auth-3 | – | 2.1 (3.6) | 1.8 (16.5) | – |
| aiofm-1 | – | – | −1.7 (17.5) | – |
| chiba-9 | 1.0 (6.0) | 5.3 (6.3) | 3.2 (16.1) | – |
| csic-10 | – | – | −14.3 (28.1) | – |
| amoiap-2 | 0.9 (3.1) | 0.0 (3.1) | 13.9 (9.3) | – |
| bsu-5 | – | – | 1.8 (10.7) | – |
| iupb-37 | 4.8 (10.2) | – | – | – |
| knmi-23 | 0.3 (1.8) | 1.4 (1.7) | 2.6 (12.7) | −0.5 (1.4) |
| luftblick-26 | −1.4 (1.5) | −0.2 (1.4) | −0.5 (4.5) | −1.3 (0.7) |
| luftblick-260 | 0.5 (1.2) | −0.5 (2.8) | – | −2.6 (5.1) |
| luftblick-27 | −1.7 (1.5) | −1.7 (1.8) | −2.7 (4.7) | −0.4 (0.7) |
| luftblick-270 | −2.5 (1.5) | −1.0 (3.5) | – | −2.5 (5.1) |
| nasa-31 | −1.9 (2.3) | −0.3 (2.3) | 6.3 (14.0) | −2.3 (1.3) |
| nasa-32 | −1.1 (1.9) | −0.9 (2.0) | −2.6 (6.0) | −1.3 (0.9) |
| iuph-19 | – | −1.1 (1.7) | −0.2 (4.4) | – |
| dlrustc-13 | −0.8 (2.1) | 0.4 (3.0) | −2.2 (5.2) | 0.5 (1.7) |
| dlrustc-14 | −2.6 (2.0) | −0.9 (2.1) | −1.6 (4.5) | −5.7 (2.3) |
| niwa-29 | 1.3 (6.0) | 1.8 (7.9) | −5.2 (8.2) | −0.0 (3.0) |
| lmumim-35 | −3.9 (3.2) | −5.0 (1.9) | −3.8 (4.2) | 1.1 (9.7) |
| cma-7 | – | −2.4 (5.1) | 1.9 (8.5) | – |
| cma-8 | −2.1 (3.7) | – | – | 11.6 (7.4) |
| iiserm-16 | – | 4.0 (2.7) | 5.6 (13.1) | – |
| knmi-21 | – | – | −19.3 (12.4) | – |
| knmi-22 | 1.0 (4.8) | – | – | – |
| nust-33 | – | 9.0 (3.7) | 22.1 (11.8) | – |
| latmos-24 | −9.2 (6.1) | – | – | 3.1 (2.7) |
| latmos-25 | −2.5 (3.7) | – | – | 1.0 (1.8) |
| Median from all instruments | −0.4 (2.3) | −0.1 (2.7) | 0.3 (8.9) | −0.2 (1.5) |

Concerning zenith-sky twilight analyses, zenith measurements were selected in a limited range of solar zenith angles (from 75 to 93°) representative of typical twilight measurements, similar to those performed within NDACC for stratospheric ozone and $NO_2$ monitoring (see, e.g., Hendrick et al., 2011), where an SZA range from 86–91° is used. Figures 10 and 11 show examples of the regression analysis for the case of MAX-DOAS $NO_2$ and $O_4$ measured in the visible spectral range. A more complete overview of the regression results obtained for all species can be found in the Supplement, where the regression analysis is shown for all elevation angles and viewing directions. As can be seen, a tight correlation is observed for most of the participating instruments.

The values for the slope ($S$), intercept ($I$) and the rms calculated as part of the regression analysis are shown in each of the instrument panels. The slope and intercept parameters, respectively, quantify the mean systematic bias and offset of individual data sets against the median reference, while the rms error provides an estimate of the measurement noise or dispersion.

A similar analysis is presented in Fig. 12 for HCHO. Note the much larger relative noise obtained for this weak absorber and the larger dispersion of the results. For this molecule, low-noise research-grade instruments perform significantly better than other systems. A similar conclusion was reached in Pinardi et al. (2013) (see in particular Fig. 18). Note, how-

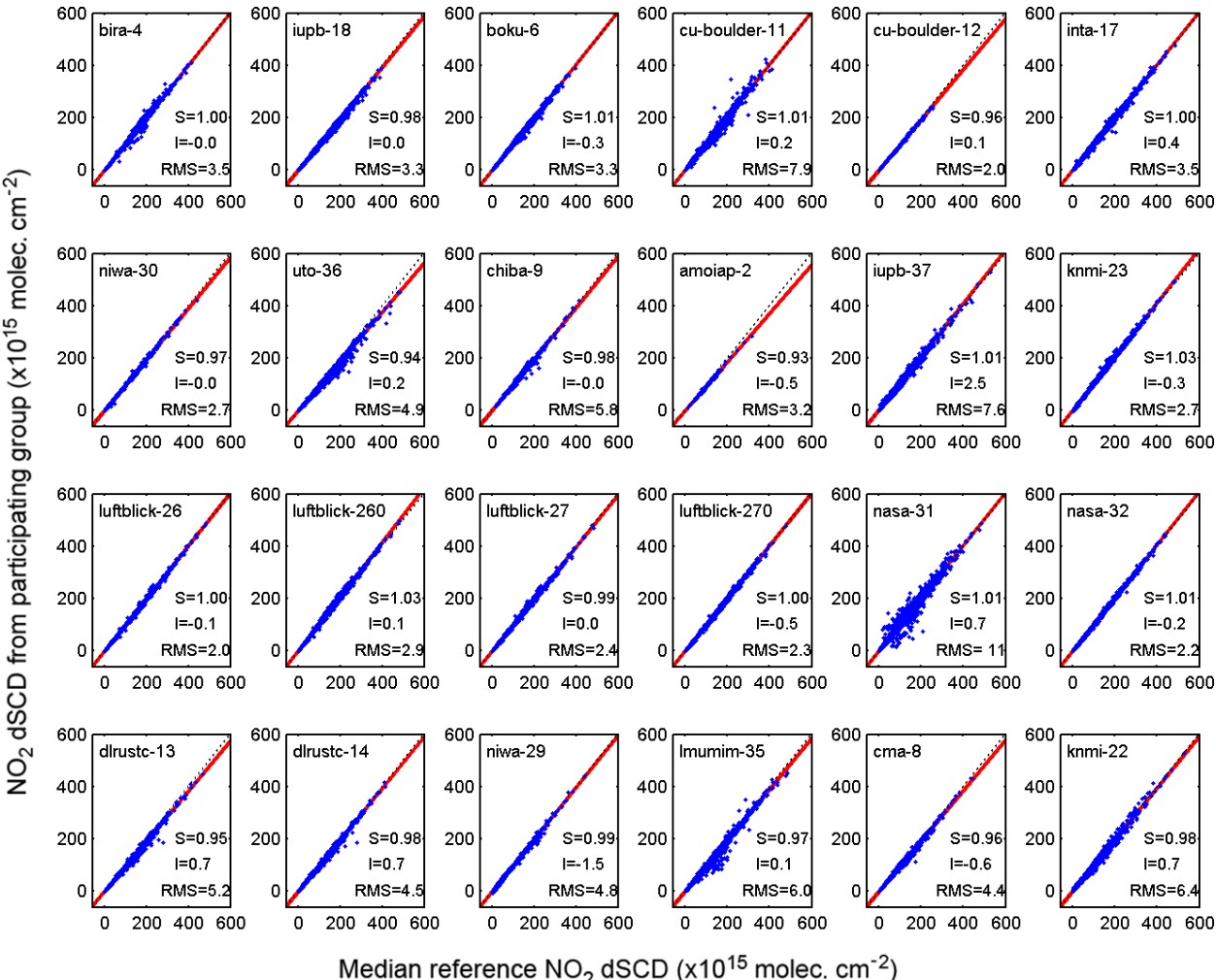

**Figure 10.** Regression analysis for $NO_2$ dSCDs (measured in the visible wavelength region) for each instrument which was measuring $NO_2$ in this wavelength region plotted against median values for the whole semi-blind phase, including all viewing and azimuth angles (blue crosses). The linear regression line is displayed as a red line, the 1-to-1 line as a reference as a dotted line. Instruments are identified with their affiliation and instrument ID number. $S$ is the slope of the regression, $I$ the intercept and rms the root-mean-square of the regression residuals.

ever, that instruments equipped with compact Avantes spectrometers (e.g. the Pandora and EnviMes instruments) also provide good results despite a larger noise level.

It is interesting to further investigate the dSCD noise levels and their dependencies. Two approaches are generally used to characterise the random uncertainties of dSCD measurements. The first one consists of inspecting the dSCD uncertainties produced by the DOAS least-squares fitting procedure. Assuming normally distributed residuals, these uncertainties provide a good estimate of the random uncertainty due to instrument noise. Figure 13a displays DOAS fit dSCD errors normalised to their median for the 12 data products investigated in this exercise for all instruments and all elevation angles. For each box, the bottom and top edges of the box indicate the 25th and 75th percentiles, respectively, while the whiskers extend to the most extreme data points. Median

dSCD error values are given for reference on the upper $x$ axis. Next to the fitting errors, in Fig. 13b, the rms residuals from regression analyses are represented, normalised in the same way as the dSCD errors. Owing to the good synchronisation achieved during CINDI-2, these rms values provide a good estimate of the comparison noise against median dSCD references. Assuming ideal comparison conditions (i.e. perfect co-location in time and space under stable atmospheric conditions), one would expect these two independent estimates of random uncertainties to converge towards a common value. This happens to be approximately the case for HCHO and for most of the twilight (stratospheric) data products, except for the O3vis product. In contrast, however, regression noise values derived for $NO_2$ and $O_4$ dSCDs appear to be much larger than their corresponding fitting uncertain-

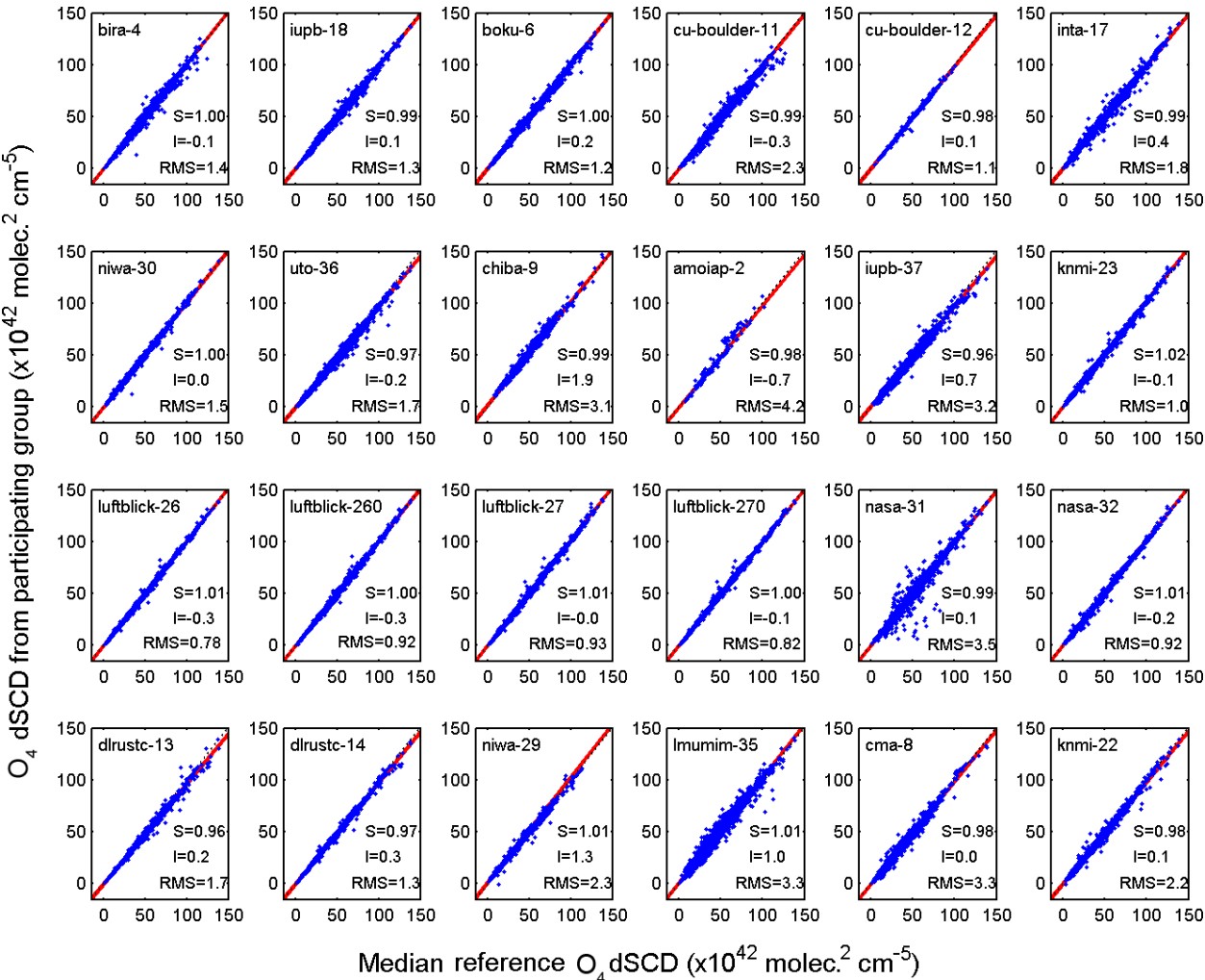

**Figure 11.** Same as Fig. 10 but for O$_4$ dSCDs measured in the visible wavelength range.

ties, and in the case of the NO2vis product, the difference is most pronounced.

The results shown in Fig. 13 indicate that despite the measurement synchronisation (to better than 1 min) and the fact that all instruments were oriented and pointing towards the same air masses, the variability of the NO$_2$ and possibly aerosol or cloud features can be large enough to introduce a difference between the individual data sets in the comparison exceeding the measurement uncertainty by an order of magnitude. This means that in this intercomparison, atmospheric variability limits the reproducibility and representativeness of individual MAX-DOAS measurements for species such as NO$_2$. Accordingly, it can be argued that for low-noise instruments the random uncertainty in tropospheric NO$_2$ dSCD measurements is by far dominated by atmospheric variability effects and the details of how this variability is smoothed out by the measurement system (in particular the FOV of the MAX-DOAS telescope and the integration time are key parameters). This also suggests that using

DOAS fit errors as a measure of the dSCD error covariance (as often applied in MAX-DOAS profile inversion schemes; see, e.g., Clémer et al., 2010; Vlemmix et al., 2015; Frieß et al., 2019) is not appropriate especially for tropospheric NO$_2$ retrievals. Instead, a more representative estimate of the random error should be derived from the measured variability of the observed dSCD, with the DOAS fit uncertainties being a lower boundary for the measurement uncertainties at best. This issue has been further investigated in a recent publication by Bösch et al. (2018).

This interpretation is strongly corroborated by Fig. 14, where the angular dependence of regression noise results is displayed (in green) for the NO2vis, O4vis and HCHO products. As can be seen, the comparison noise on NO$_2$ dSCDs is largest at the lowest elevation angles and regularly decreases at larger elevations. This behaviour, which is less marked but also observed for O$_4$, is consistent with atmospheric variability effects since one expects that inhomogeneities of the tropospheric NO$_2$ field will affect observa-

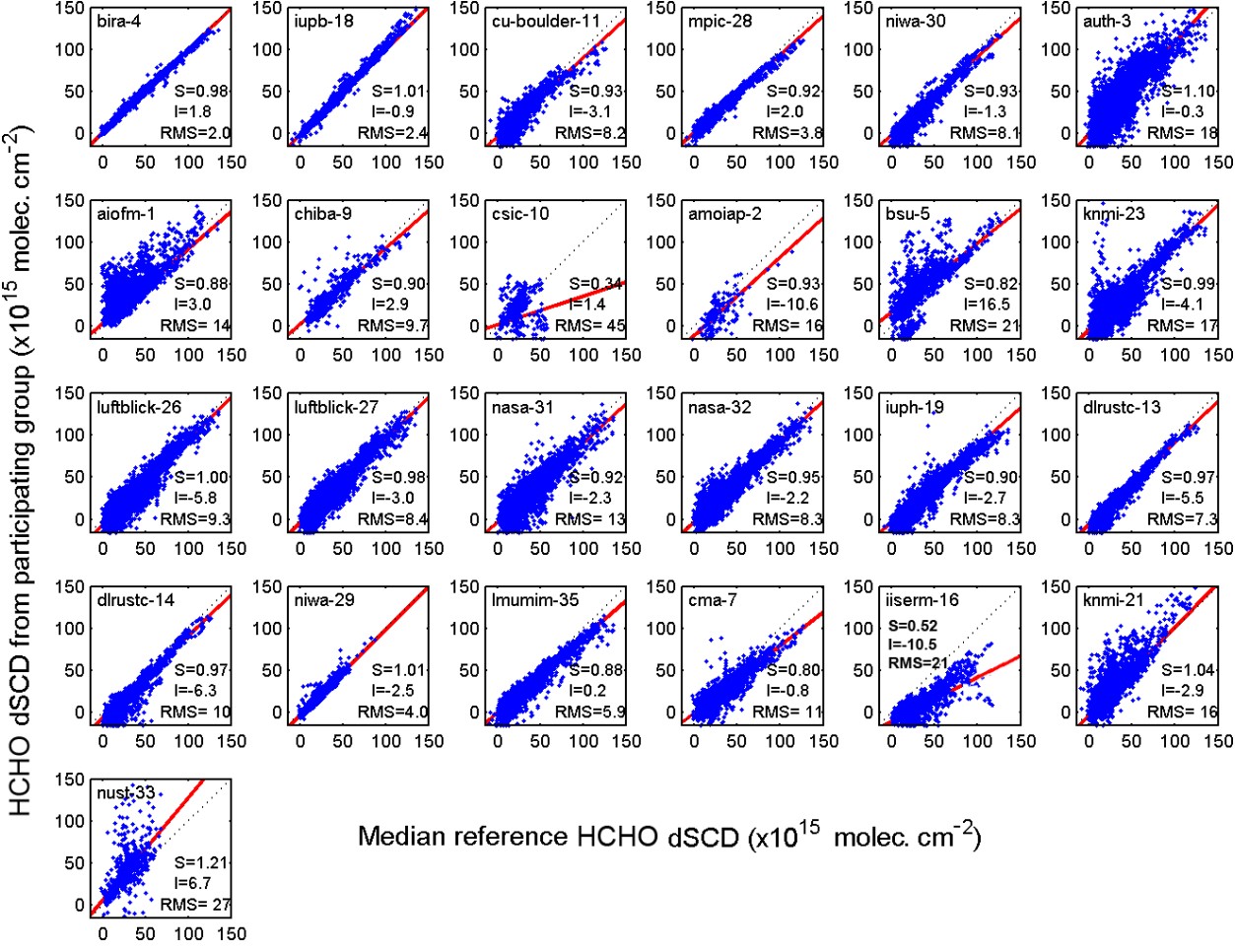

**Figure 12.** Same as Fig. 10 but for HCHO dSCDs.

tions at lowest elevation angles (which have strongest sensitivity to near-surface $NO_2$) more strongly. In contrast, the HCHO comparison noise is virtually independent of the elevation angle and close in size to the fitting noise. Note that
even at the highest elevation of 30°, the comparison noise on $NO_2$ and $O_4$ dSCDs remains larger than the fitting noise, suggesting that atmospheric variability remains a dominant effect at all the angles used for profile inversion. Figure 15 displays results from the same analyses but restricted to ref-
erence data sets. Similar conclusions are reached for $NO_2$ and $O_4$. In the case of HCHO, the noise level drops considerably, which reflects the high sensitivity of instruments selected for building the HCHO reference. Interestingly, one can also see that regression rms and fitting residuals now
match almost perfectly (and at all elevation angles) meaning that for this molecule most of the residual variance from regressions involving good instruments can be explained by instrument shot noise. Figure 16 displays results obtained when selecting Pandora instruments only. In comparison to other
systems, Pandoras are characterised by a larger field of view (see Fig. 6), which probably explains the smaller regression

rms observed for $NO_2$ and $O_4$ (likely due to a more efficient smoothing of the atmospheric variability).

Figure 17 provides a different view of the data set already presented in Fig. 10, displaying the slope, intercept and rms
for the $NO_2$ (visible) regression analysis graphically for all measurement days and viewing directions, and for several elevation angles (1, 3, 5, 8, 15 and 30°). Similar plots have been generated for all the trace gas data products and are provided in Sects. S1 and S2 of the Supplement. Note that
for two instruments (chiba-9, amoiap-2), only one elevation angle from the above set is available due to technical reasons intrinsic to these instruments. The limits indicated with dashed lines are introduced and discussed further in Sect. 4. Figure 17 can be compared with similar figures in Roscoe et
al. (2010) (Fig. 6) and Pinardi et al. (2013) (Fig. 7) allowing results from CINDI and CINDI-2 to be linked. It is interesting to note that although the range of variability on the slope and intercept parameters was similar in both campaigns, the proportion of instruments matching the 5 % limit on the slope
was significantly improved in CINDI-2, indicating a general improvement of the overall consistency of the measurements.

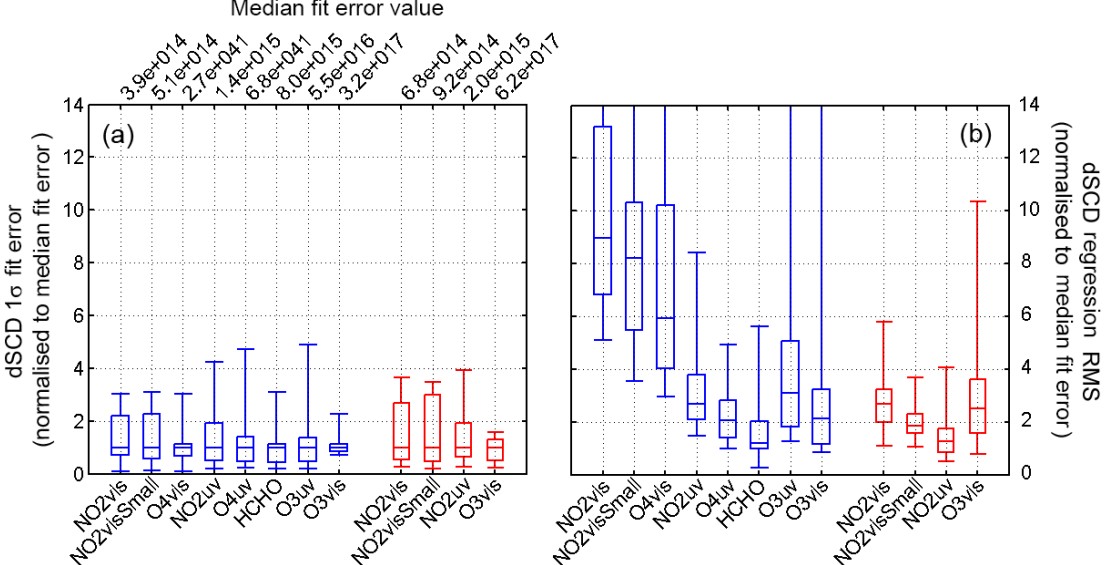

**Figure 13. (a)** Box-and-whisker plot of the $1\sigma$ fit error of the dSCDs for the 12 data products, for all instruments and for all elevation angles. MAX-DOAS products are represented in blue and zenith-sky twilight in red. **(b)** Box-and-whisker plot of the rms from dSCD regression analyses, again for the 12 data products under investigation.

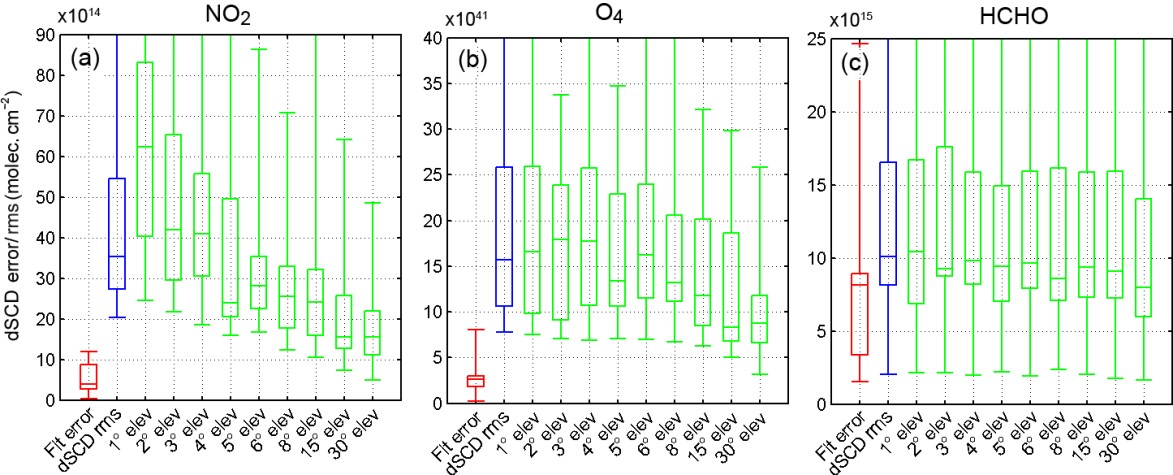

**Figure 14. (a)** Box-and-whisker plot of the $1\sigma$ dSCD fit error (red), the regression rms for all elevation angles (blue) and rms from dSCD regression analyses sorted as a function of the elevation angle (green) for NO₂ in the visible wavelength range. **(b)** Same as **(a)** but for O₄ (visible). **(c)** Same as **(a)** but for HCHO.

As is to be expected from well-calibrated instruments, the three regression parameters displayed in Fig. 17 generally do not show any marked angular dependence. However, some data sets display larger deviations and sometimes also significant angular dependencies. For these cases, the lowest elevation angles often show the largest deviations (e.g. intercept and rms for nasa-31 and dlrustc-13, slope for uto-36) but not always (e.g. rms for cu-boulder-11 and slope for iupb-37). Although this certainly does not explain all discrepancies, it is interesting to note that, in many cases, the largest deviations are observed for instruments that did not supply (or

could only partially supply) horizon scan information and therefore could not benefit from the angular correction applied in pre-processing (see Sect. 3.4).

## 4 Investigation of instrument performance

With MAX-DOAS-type instruments having gained popularity in recent years and their usage becoming more widespread, the need for a reliable and clearly documented assessment process is becoming more pressing. A semi-blind intercomparison campaign such as CINDI-2 provides the

Reference instruments only

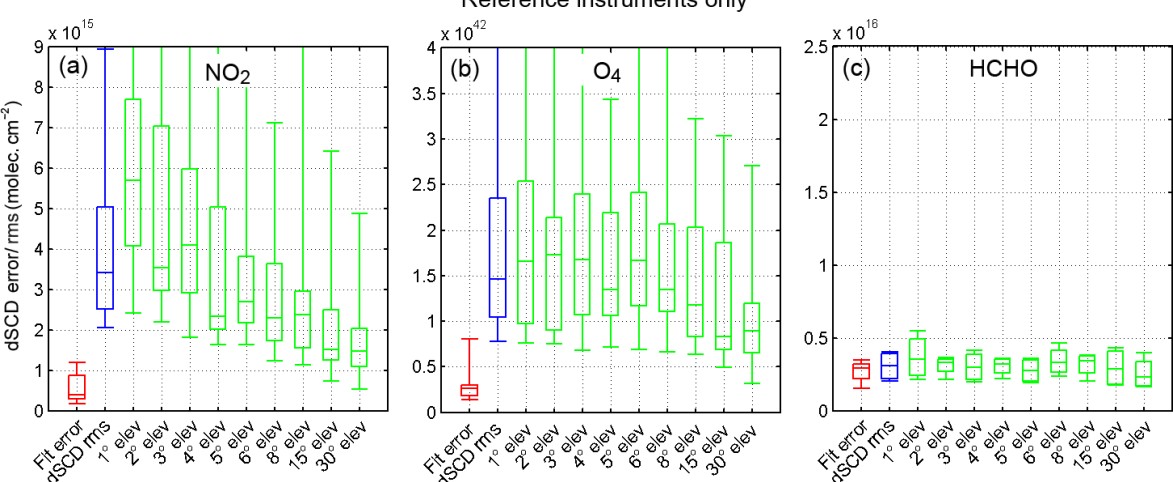

**Figure 15.** Same as Fig. 14, but for reference instruments only.

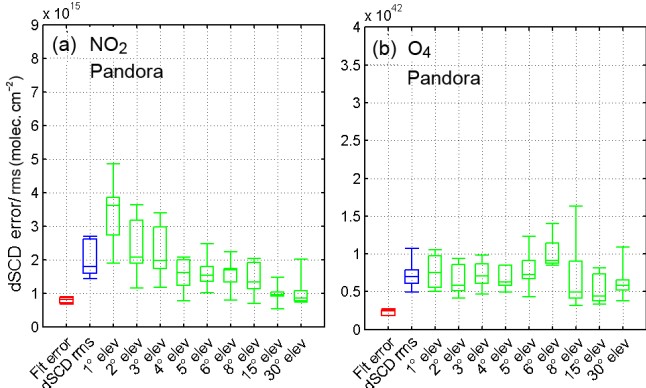

**Figure 16.** Same as Fig. 14, but for Pandora instruments only.

ideal conditions to obtain a data set for such a process and the opportunity to involve as many MAX-DOAS instruments as possible.

Three criteria based on the regression analysis discussed in Sect. 3.7 (slope, intercept and rms) have been selected to assess the performance of each of the participating instruments with regard to the eight MAX-DOAS and four zenith-sky products. For each of these parameters, specific limits have been set for the performance evaluation as listed in Table 4. These were semi-empirically derived from a visual inspection of the distribution of the slope, intercept and rms values for each of the eight CINDI-2 MAX-DOAS and four zenith-sky data products. The histograms and limits (indicated with red lines) for the eight MAX-DOAS data products are displayed in Fig. 18 for the slope, intercept and rms from the regression analysis. Note that the NO₂ and O₃ criteria were adapted from previous NDACC campaigns (see the Introduction for further details). For other products, limits were set arbitrarily to capture the most probable values while ex-

cluding clear outliers. The limits were, however, chosen to exceed the median of the measurements (indicated with blue lines). The blue lines represent the percentiles 16 and 84 (84 only for rms), and it can be seen that the certification criteria have been chosen to exceed these limits. One exception is HCHO, since for this product the difference between well-performing and less well-performing instruments was much larger than for the other products.

It must be acknowledged that the performance limits defined in this work (as in previous NDACC intercomparisons) are representative of the current state of the art of the instrumentation and to some extent also reflect the measurement conditions in Cabauw. Another campaign being performed at, e.g., a cleaner or more stable site could lead to different values for the limits.

Figure 19 shows a summary of the same regression statistics previously discussed in Sect. 3.7 and displayed in Figs. 10 and 15, but with all individual elevation angles added up resulting in one single value for each parameter, instrument and data product. This means that only three values are displayed for each instrument. The green shaded areas denote the limits defined in Fig. 18 with all parameters falling within the limits being displayed as blue dots while values in red do not meet the respective criterion. Note that not all of the 36 instruments measure NO2vis. For the slope of the NO2vis regression analysis shown in Fig. 19a, two instruments (uto-36 and amoiap-2) fall outside the limit. One other instrument (iupb-37, the imaging instrument) does not meet the criterion set for the intercept (see Fig. 19b) and one (nasa-31) for the rms (Fig. 19c). One such summary plot has been produced for each of the eight MAX-DOAS and four zenith-sky data products which can be individually viewed in Sects. S1 and S2 of the Supplement.

To further summarise the outcome of the regression analysis and provide an overview of all eight MAX-DOAS data

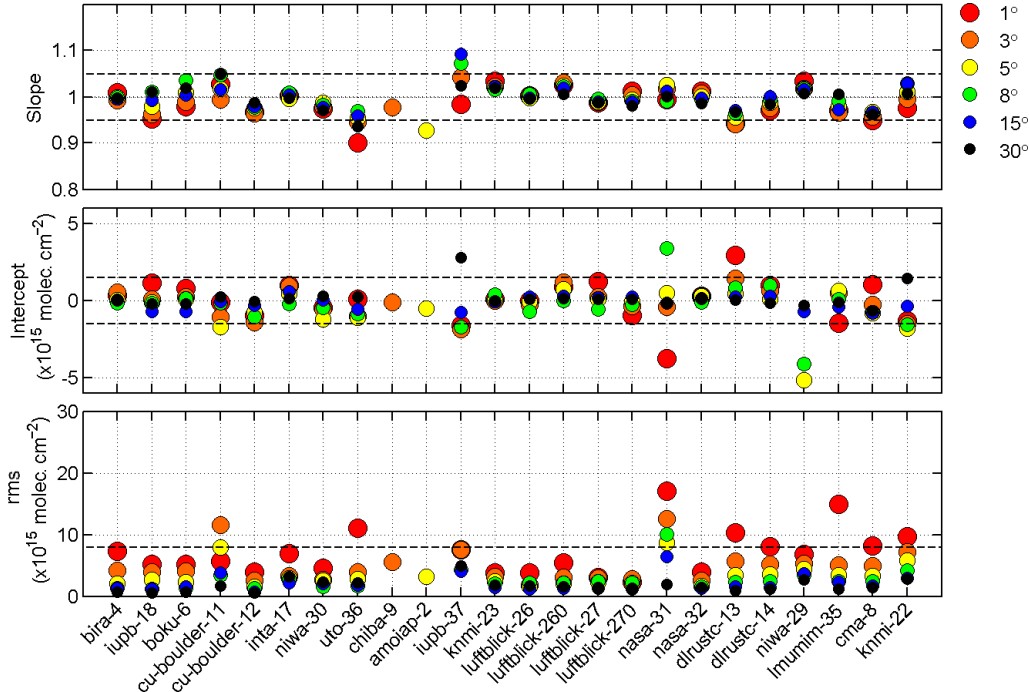

**Figure 17.** Slope, intercept and rms of regression plots against the median dSCD reference, for each of the 24 instruments measuring NO$_2$ in the visible wavelength range (as shown in Fig. 10). The values are colour-coded corresponding to the elevation angles (1 to 30°). Apart from a couple of exceptions (chiba-9, amoiap-2), most instruments measure the whole range of elevation angles. The dashed lines indicate the limits when comparing the values of the parameters for the different instruments with the aim of identifying outliers in a more objective way. Section 4 and Fig. 18 explain in more detail how the actual values of the limits were selected, and the values are listed in Table 4.

products, Fig. 20 displays the three selected parameters for all participating instruments. The performance is colour-coded with regard to parameters falling inside the performance limit (green) or not meeting the criterion (orange). In exceptional cases where the slope or rms exceeds the threshold by more than a factor of 4, the performance is colour-coded in black. Just under one-third of all the instruments do meet all the criteria. Figure 21 shows the same summary for the four zenith-sky data products. In this case, more instruments meet all criteria and none of the products have any parameters which exceed any performance threshold by a factor of 4 or more.

Figure 22 further synthesises all results into one overview plot. This assessment matrix shows the outcome for all 36 instruments, eight data products for MAX-DOAS and four data products for zenith-sky mode. Any box coloured with green denotes that all three assessment criteria for that instrument and data product have been fulfilled. Boxes marked with yellow and orange denote that one or two criteria, respectively, have not been met, while red means that all three criteria have not been met, and black indicates that this data set has at least one extreme outlier. Additionally, both the reported dSCD regression rms and the DOAS fit rms are used to sort the data products accordingly, with the smallest median rms being assigned the lowest number in each case.

The order in which the instruments are displayed in Fig. 22 is identical to Figs. 20 and 21, with the instruments being grouped into five different categories: custom-built, Pandora, EnviMes, mini-DOAS and SAOZ. Custom-built instruments are assembled in-house and often designed with specific research purposes in mind. This category displays the greatest diversity in performance, and it includes the highest performing instruments as well as the instrumentation with the biggest difficulties meeting the set criteria of the performance assessment. In some cases, this can be related to the level of experience of the research group involved in building the instrument and/or in operating the instrument and performing the data analysis.

The first seven custom-built instruments listed in Fig. 22 meet all criteria for all measured MAX-DOAS data sets with the following three instruments also being close to fulfilling almost all criteria for most of the data. The last six instruments listed under the custom-built category, however, struggle to either meet two criteria or to meet all criteria for one of the measured data products. Additionally, HCHO or O3uv data sets measured by three of the instruments (csic-10, bsu-5 and iiserm-16) contain extreme outliers.

The seven Pandora and five EnviMes instruments overall show a more consistent picture. Four of the Pandoras meet all categories and two of the other Pandoras satisfy all but one of

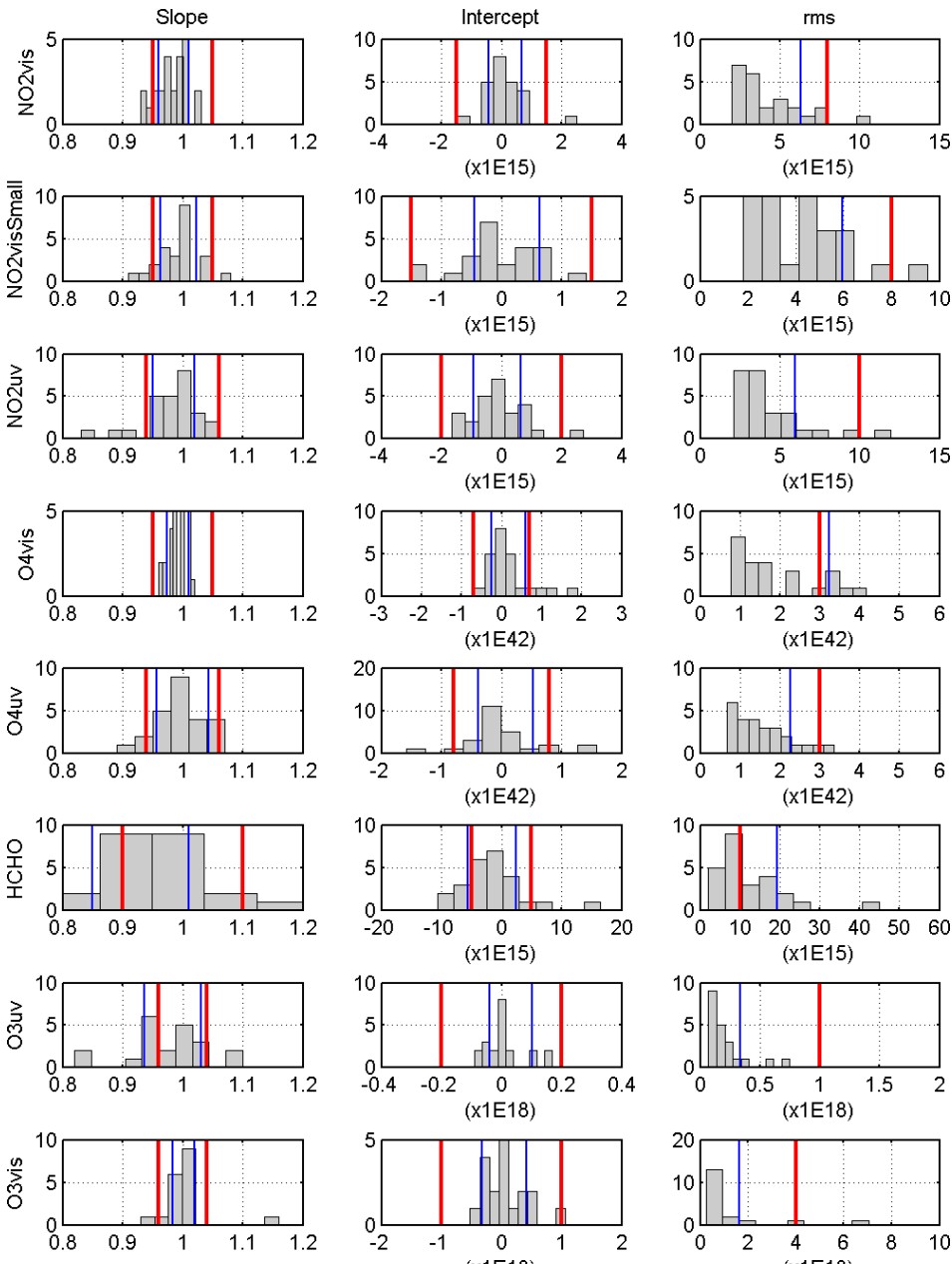

**Figure 18.** Limits for the assessment criteria for the eight MAX-DOAS data sets shown by red lines. The blue lines represent the percentiles 16 and 84 (84 only for rms), together with histograms of the slope being displayed in the left column of panels, the intercept in the middle and the rms in the rightmost panels (see also Table 4).

the criteria for one or two of the data products. Nasa-31, however, experienced problems during operation and had some dirt inside the head sensor which was moving around and blocking part of the instrument FOV as well as having a loose tracker shaft. This caused a significantly reduced signal-to-noise ratio and an increased pointing uncertainty (see the large error bar for this instrument in Fig. 6) that had negative consequences for all data products analysed in the campaign. These problems were detected during the campaign

and an attempt was made to fix them. In spite of these issues, most criteria were still met. It should be noted though that the behaviour displayed by nasa-31 did not fully represent the observational capabilities of a Pandora, as clearly evidenced by results from other instruments of the same type. The EnviMes instruments performed overall well when measuring NO$_2$ but struggled more to fulfil all criteria for the HCHO and O3uv data sets apart from niwa-29 which satisfied all criteria for HCHO and O3uv, while not satisfying one of the

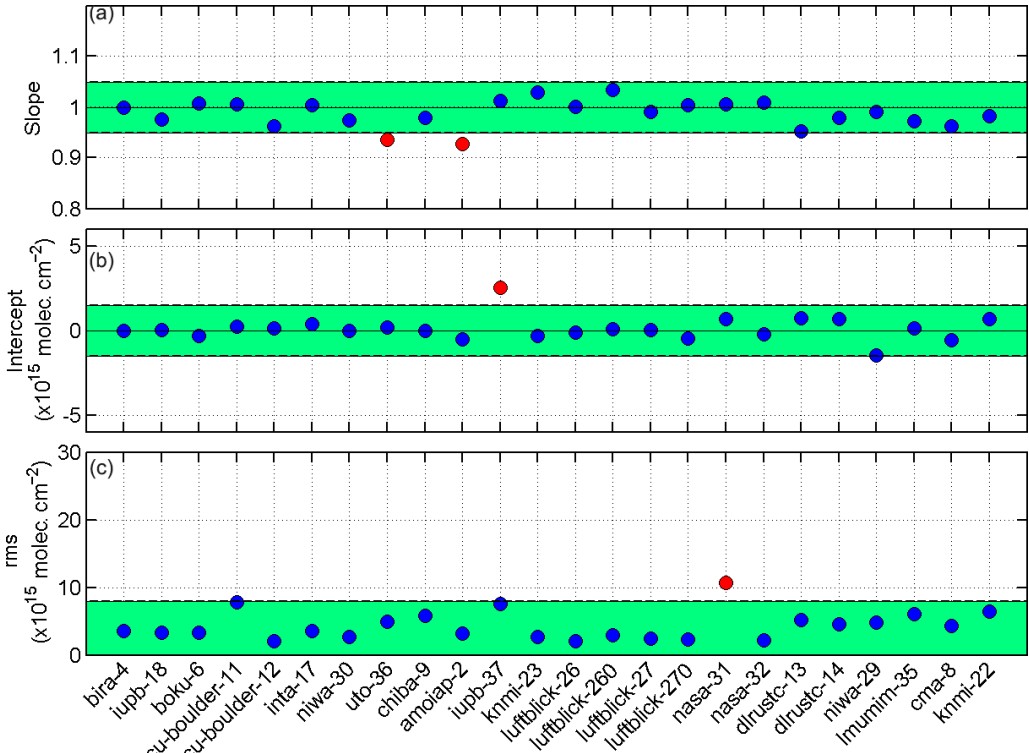

**Figure 19.** Summary of the NO₂ visible regression statistic shown in Fig. 10. The slope, intercept and rms values are displayed in **(a)**, **(b)** and **(c)**, respectively, for all measurement days, all viewing directions and all elevation angles. The green shading indicates the limits as defined in Table 4 and Fig. 18 for NO2vis; the values falling within these limits are plotted in blue, the ones outside the limits in red.

criteria for both of the O₄ data sets and one of the NO₂ data sets. Most of the six mini-DOAS instruments measured NO₂ satisfactorily in all three wavelength ranges and only failed to satisfy one criterion in the O₄ data sets. However, they experienced discernible difficulties when measuring HCHO and O₃, which includes all failed criteria and extreme outliers.

The zenith-sky twilight data set (rightmost four columns in Fig. 22) shows a consistent performance for all custom-built, Pandora, EnviMes and SAOZ-type instruments and all four data products (apart from nasa-31; see discussion above) with in most cases (90 %) all criteria satisfied and in just eight cases one criterion not satisfied. The performance of the mini-DOAS instruments is for the zenith-sky data more variable, with one instrument (cma-8) not satisfying any of the criteria for O3vis and another (nust-33) failing two out of three criteria for the NO2uv product. The two SAOZ instruments measure zenith-sky data only and either satisfy all criteria or do not meet just one of them.

The ranking provided in each of the individual boxes in Fig. 22 is based on the dSCD regression rms (first value) and the rms calculated as part of the data fitting routine (second value), the instruments with the smallest rms (i.e. the smallest measurement noise) being assigned the lowest number. Overall, the combined ranking reflects the performance assessment of the individual instruments, but there are a couple of

noteworthy deviations. For example, the data products measured by auth-3 have very large numbers corresponding to a high rms (high measurement noise in comparison to other systems) but at the same time meet almost all performance criteria. On the other hand, the data products measured by aiofm-1 have an excellent fit rms rating corresponding to a very low measurement noise, while none of the data products satisfy all criteria. This apparent inconsistency reflects the nature of the performance assessment methodology, which puts larger emphasis on the assessment of systematic biases in measured dSCDs than on the noise. We have also seen that the comparison noise in regression analyses is, for some of the products, (NO₂, O₄) dominated by atmospheric or observation geometry effects rather than by actual instrumental noise.

The performance matrix shown in Fig. 22 can be used to assess the participating groups and their instruments regarding their capability to measure NO₂, O₃ and HCHO concentrations and aerosols (using O₄ measurements) at sufficiently high quality to allow reliable geophysical studies or satellite validation efforts. In addition to offering an instantaneous picture of the level of performance of the current international MAX-DOAS research community, these results also provide the background information needed for the formal

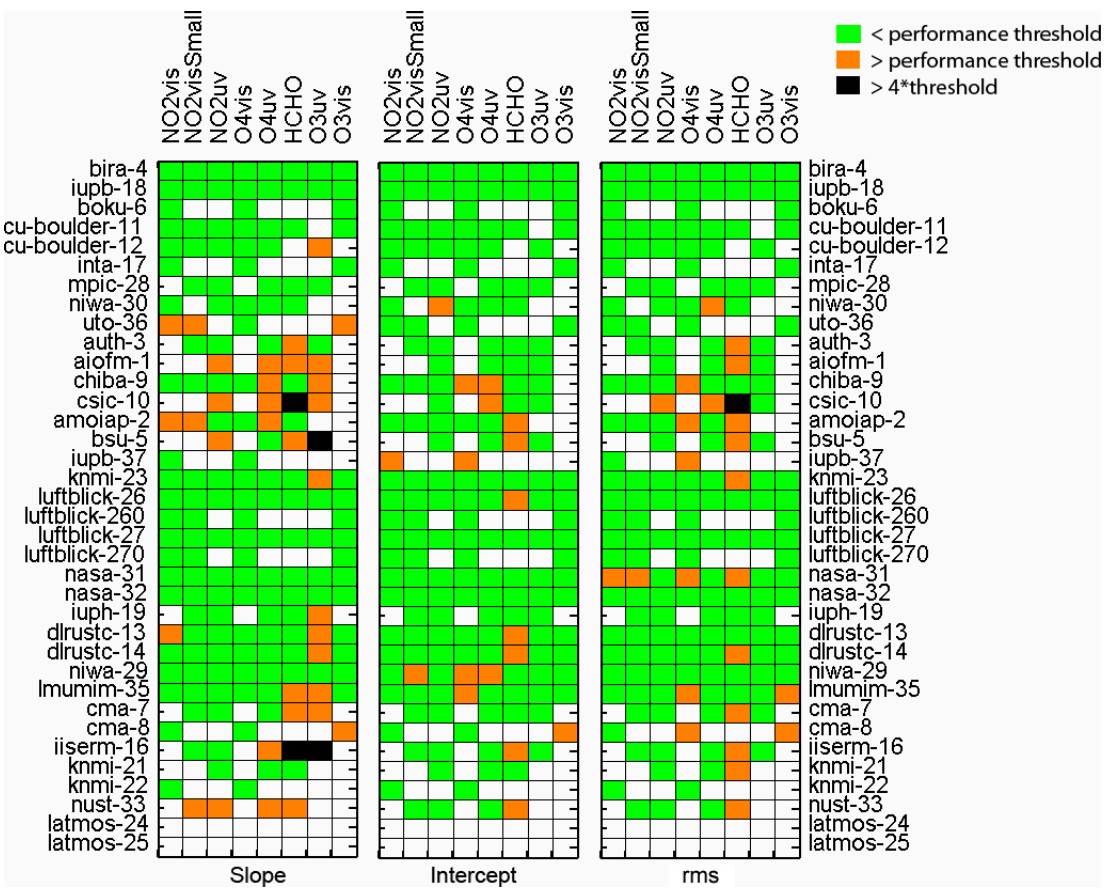

**Figure 20.** Overview of performance results for the slope, intercept and rms from the regression analysis displayed for all participating instruments and MAX-DOAS data products. Colour coding denotes whether each of the parameters is within the set criteria (green), whether the performance threshold is exceeded (orange), or whether it is exceeded by more than a factor of 4 (black).

assessment and certification of instruments contributing to the NDACC network.

## 5   Recommendations for network operation and future campaigns

The CINDI-2 exercise included more target trace gas species and more instruments and participants from many different institutes than previously attempted in any other UV–visible spectroscopy intercomparison exercise. This provided a logistical challenge, which was addressed by setting up a carefully managed campaign. Beyond the detailed consistency assessment documented in this work, several lessons were learnt that are expected to be of benefit to measurements conducted at network sites.

The accuracy and stability of the MAX-DOAS elevation scans was found to be critical, especially for measurements at low elevation angles. Therefore, we recommend regularly calibrating elevation scan devices using one of the methods described in Donner et al. (2020). Moreover, for instruments not equipped with an internal pointing verification system (e.g. digital inclinometer or self-calibrating sun tracker), horizon scans should be regularly performed, ideally on a daily basis, in order to verify the long-term stability of the pointing elevation.

The degree of geometric and temporal synchronisation prescribed for the instruments has revealed that spatial and temporal variability in the atmosphere is significantly greater than the total effect of instrument-derived uncertainties. As a result, atmospheric variability limits the reproducibility and representativeness of individual MAX-DOAS measurements for species such as NO$_2$. For this molecule, we estimate that the variability has a spatial scale that is at least as fine as many tens to a few hundreds of metres. This order of magnitude is consistent with the horizontal distances sampled by the average FOV (1°) and the horizontal separation of the instrument telescopes. It implies that random error estimates on NO$_2$ dSCDs should account for atmospheric variability effects in addition to spectral fitting uncertainties. To a lesser extent, the same reasoning applies to O$_4$ dSCD measurements.

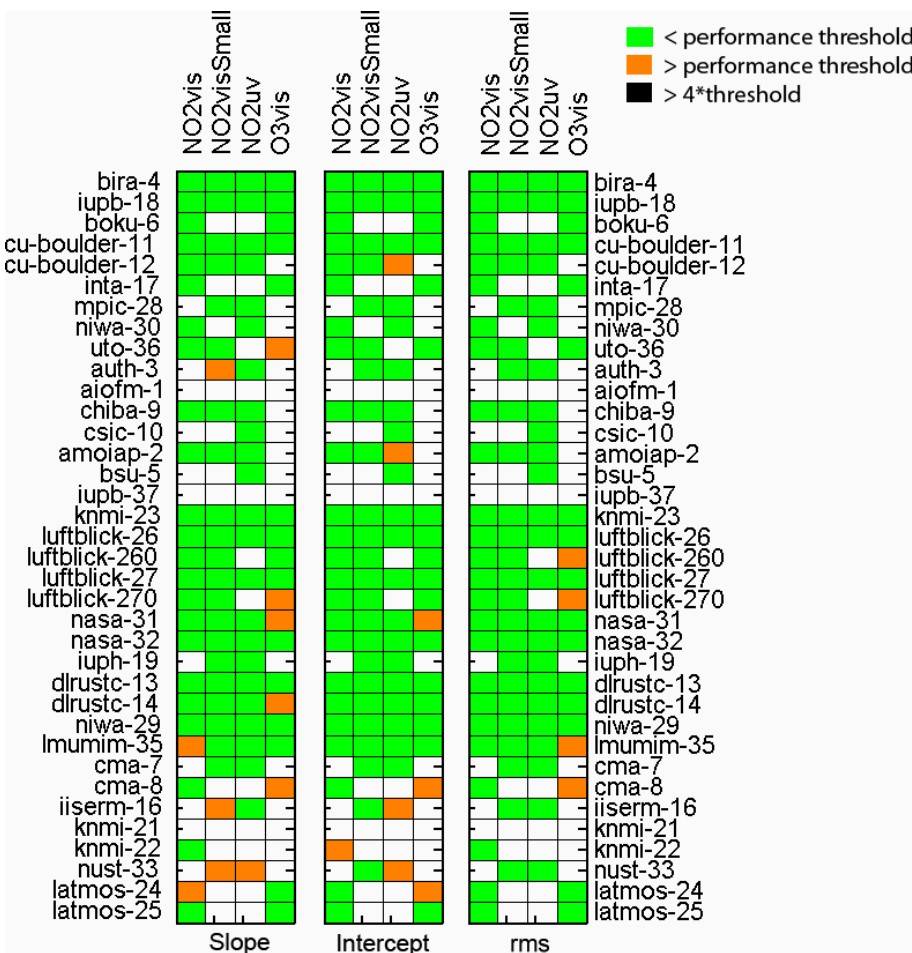

**Figure 21.** Same as Fig. 20 but for the zenith-sky products.

For high-quality HCHO measurements, radiance measurements should reach a signal-to-noise ratio of 1000 or better in the spectral range from 335 to 360 nm, corresponding to HCHO dSCD uncertainties of $5 \times 10^{15}$ molec. $cm^{-2}$ or better. At this level of random uncertainty (and in contrast to the $NO_2$ case), HCHO spectral fitting errors still dominate over atmospheric variability effects.

One also anticipates that future similar intercalibration campaigns will strongly benefit from the lessons learnt during and after CINDI-2. As already pointed out, the campaign was successful in improving (1) the spatial and temporal synchronicity of the measurements and (2) the characterisation of the pointing elevation accuracy from all instruments and their impact on the DOAS analysis results. Despite these achievements, a few critical points were identified that deserve more attention in future deployments.

The data acquisition protocol, which proved to be very useful for instrument synchronisation, was not fully adequate for monitoring the spatial variability in highly variable trace gases such as $NO_2$. As discussed in Sect. 3.7, results from CINDI-2 indicate that in spite of the improvement in mea-

**Table 7.** Summary of the level of agreement obtained for dSCD measurements during CINDI-2 and typical uncertainties achieved by high-quality and standard instruments for the different data products.

| Data product | Median agreement level between instruments | | Median dSCD fit error (molec. $cm^{-2}$) | |
|---|---|---|---|---|
| | Bias (%) | rms (molec. $cm^{-2}$) | High-quality instruments | Standard instruments |
| NO2vis | 3 | $3 \times 10^{15}$ | $2 \times 10^{14}$ | $7 \times 10^{14}$ |
| NO2visSmall | 3 | $3.5 \times 10^{15}$ | $2 \times 10^{14}$ | $5 \times 10^{14}$ |
| NO2uv | 3 | $4 \times 10^{15}$ | $6 \times 10^{14}$ | $1 \times 10^{15}$ |
| O4vis* | 2 | $1.5 \times 10^{42}$ | $1.5 \times 10^{41}$ | $3 \times 10^{41}$ |
| O4uv* | 2 | $1.5 \times 10^{42}$ | $3 \times 10^{41}$ | $8 \times 10^{41}$ |
| HCHO | 8 | $1 \times 10^{16}$ | $3 \times 10^{15}$ | $8 \times 10^{15}$ |
| O3vis | 2 | $6 \times 10^{17}$ | $3 \times 10^{17}$ | $3 \times 10^{17}$ |
| O3uv | 4 | $1.6 \times 10^{17}$ | $1.3 \times 10^{16}$ | $6 \times 10^{16}$ |

* Note: the units for $O_4$ are $molec.^2 \, cm^{-5}$.

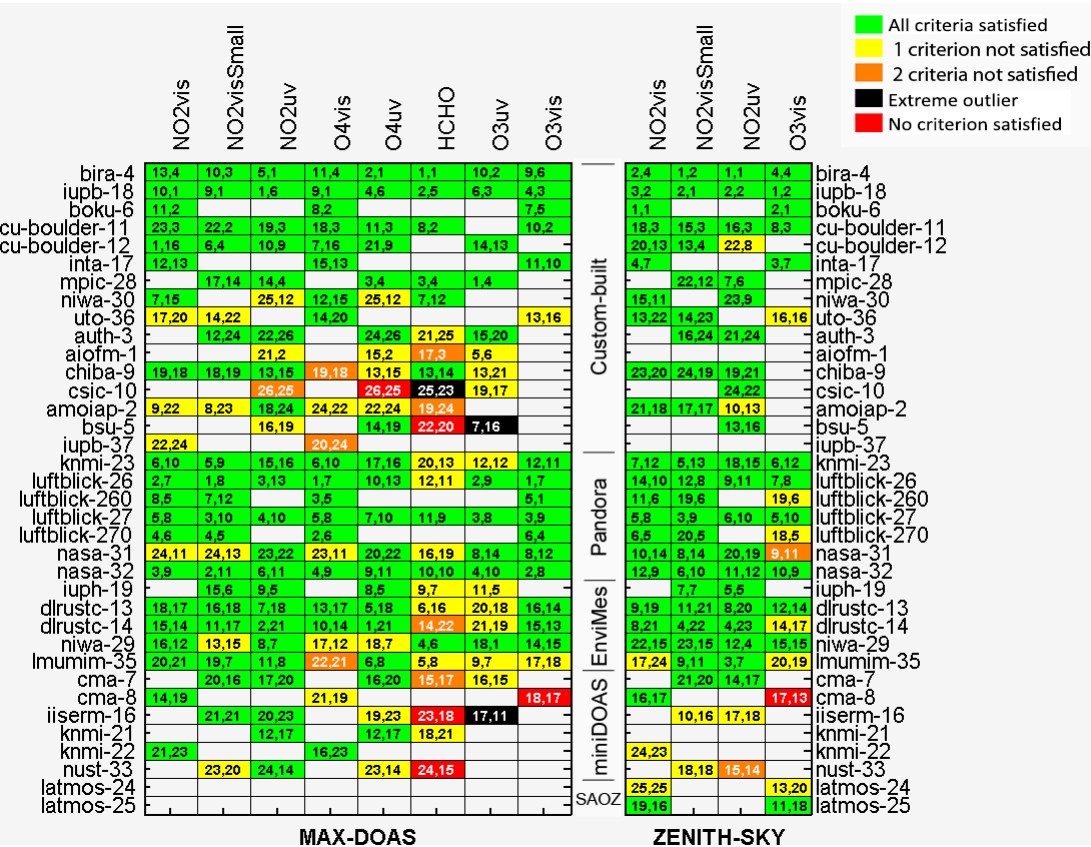

**Figure 22.** Assessment matrix for all 36 instruments and eight data products for MAX-DOAS and four data products for zenith-sky mode. Green indicates that all three assessment criteria have been fulfilled, yellow means that one criterion is not satisfied, orange means two are not, red means all three criteria have not been met, and black indicates that this data set has at least one extreme outlier. White indicates when data sets were not measured. The two numbers in each box indicate the rating for each product and instrument according to the dSCD regression rms (first value) and the rms calculated as part of the data fitting routine (second value). The instruments with the smallest rms are denoted with the smallest numbers. Note that the instruments are grouped according to their specific design as custom-built, Pandora, EnviMes, mini-DOAS or SAOZ.

suring the same air mass, the variability in some of the trace gases can still be large enough to introduce noise which is clearly exceeding the measurement uncertainty, suggesting that using DOAS fit errors as a measure of the dSCD error covariance is not appropriate. A more representative estimate of the random error should be derived instead from the measured variability of the observed dSCDs (see, e.g., Bösch et al., 2018). For future campaigns, we hence recommend adopting a strategy combining full elevation scans suitable for profile inversion at one or two reference azimuths and azimuth scans at one elevation for an evaluation of the spatial variability in trace gas concentration.

Although the campaign had a strong focus on elevation scan calibration, other aspects of the instrument calibration were handled with far less attention. Results from the data analysis, however, indicated that some of the observed discrepancies were related to a lack of proper instrumental characterisation before the campaign (e.g. detector non-linearity or spectral stray light), and it is likely that some of the re-

maining deviations are related to unresolved calibration issues. For future campaigns, a better strategy should be developed to improve the characterisation of participating instruments in preparation for field deployment. This could, for example, be organised in the form of a preparatory calibration campaign hosted by a suitably equipped lab. The focus of this exercise should be put on instrumental characteristics of major importance for DOAS-type instruments, i.e. in particular instrumental line shape, spectral stray light, polarisation response, detector response (dark current and linearity), field of view of telescope, elevation scanner accuracy and reproducibility, and instrument throughput and sensitivity.

## 6 Conclusions

CINDI-2 had a strong focus on synchronisation and collocation of the measurements as well as on the determination of the pointing accuracy, which altogether resulted in a reduc-

tion in the impact of atmospheric changes on the intercomparison exercise in comparison to CINDI. While each participating institute used their own instrumentation and analysis software (Tables 2 and 3), specific measurement procedures and retrieval settings were prescribed and strictly adhered to.

This comprehensive measurement protocol was highly successful in synchronising the timing of the measurements between all the instruments (Fig. 2). The different approaches applied to determine the pointing accuracy of the instruments and their stability during the campaign provided important information for monitoring the instrument performance (see Fig. 6). Moreover, this information was used to correct the data analysis in cases where the measurements were compromised by pointing inaccuracies leading to further improvements in consistency (see, e.g., Fig. 7). The horizon scans, in particular, were useful for identifying calibration biases, which could be addressed and corrected for the remainder of the campaign. Based on the experiences made during CINDI-2, it is highly recommended to include horizon scans into the daily measurement routine at monitoring sites and for any future MAX-DOAS intercomparison exercise. The different methods for the elevation calibration used during CINDI-2 are discussed in more detail in Donner et al. (2020).

In line with previous intercomparisons, a regression analysis of the dSCDs measured by each instrument with a reference data set was performed (see Sect. 3.5 for details on how the reference data sets were derived) and a whole range of correlation plots between the dSCDs and the reference were generated in a systematic manner (Figs. 10–12 and Sects. S1 and S2 of the Supplement). The slope and intercept of the regression analysis respectively quantify the mean systematic bias and offset of the individual data sets against the reference, and the regression rms error provides an estimate of the overall comparison noise (see, e.g., Fig. 17). These three performance criteria were further investigated, and for each of the parameters and data products, specific limits were set and applied to all the measurements (Table 4 and Fig. 18). Figures 19–22 visualise the summary of the regression analysis and provide an overview of the performance of each of the instruments regarding the eight MAX-DOAS and four zenith-sky data products.

The general level of agreement achieved for the different data products is summarised in Table 7. The median bias against the reference data sets is generally low ($< 5\%$ for most products), and comparison noise levels are of the order of $3–4 \times 10^{15}$ molec. cm$^{-2}$ for NO$_2$, $1.5 \times 10^{42}$ molec.$^2$ cm$^{-5}$ for O$_4$ and $1.0 \times 10^{16}$ molec. cm$^{-2}$ for HCHO. The table also lists the typical dSCD retrieval uncertainties that can be expected from high-quality and standard instruments. These uncertainties are compatible with satellite validation requirements (for further details, see, e.g., https://sentinels.copernicus.eu/web/sentinel/technical-guides/sentinel-5p/validation, last access: 30 March 2020). The results summarised in Table 7 agree

well with the mean relative differences and standard deviation from the reference listed for all participating instruments in Tables 5 and 6, which also show that most instruments agree within a few percent for all MAX-DOAS and twilight DOAS products (apart from HCHO and O$_3$).

This assessment process, undertaken as part of CINDI-2, provides the UV–visible absorption spectroscopy research community with guidelines and a procedure on how to assess the performance of MAX-DOAS and DOAS instruments, in particular for the inclusion into NDACC (see NDACC web page for access to the UV–vis Appendix describing these recommendations). It is expected that a similar level of consistency as seen during CINDI-2 can be obtained in the field if recommended settings are implemented and used by each participant of the network. More control in this aspect of homogeneity can be obtained through centralised processing, which is the aim of the currently developed ESA FRM$_4$DOAS project (see http://frm4doas.aeronomie.be/, last access: 30 March 2020).

The semi-blind CINDI-2 exercise, presented here, concludes with the comparison and assessment of the retrieved dSCDs of a limited number of mature data products (NO$_2$, O$_4$, O$_3$ and HCHO). However, additional species (e.g. HONO, glyoxal, BrO, H$_2$O) were also measured during the campaign, some of them being the subject of ongoing studies to be published separately. In particular, the tropospheric ozone column retrieval has been investigated in depth (Wang et al., 2018) and a publication on HONO retrievals is under way (Wang et al., 2019) as a follow-up of the first HONO intercomparison during MAD-CAT (Wang et al., 2017c). In addition to dSCD measurements, the subsequent steps in MAX-DOAS retrievals, i.e. their conversion into vertical-column and profile information, is also further investigated in a CINDI-2 profiling working group and as part of the ESA FRM$_4$DOAS project (Frieß et al., 2019; Tirpitz et al., 2020). Furthermore, other aspects of the campaign measurements are being further exploited, such as mobile car DOAS observations, reference in situ measurements and instrument elevation pointing calibration (Donner et al., 2020).

## Appendix A: DOAS retrieval settings

For each data product, a set of retrieval settings and pa-
rameters was prescribed. The use of these settings was
mandatory for participation in the semi-blind intercompari-
son. The tables below summarise the details of the DOAS
retrieval configurations used for each data product. The ref-
erenced absorption cross-section files are available from the
FRM4DOAS website (http://frm4doas.aeronomie.be/index.
php/documents, last access: 30 March 2020).

**Table A1.** DOAS settings for $NO_2$ and $O_4$ (visible range).

| | |
|---|---|
| Wavelength range | 425–490 nm |
| Fraunhofer reference spectra | Noon zenith spectra averaged between 11:30:00 and 11:40:00 UT |
| Cross sections: | |
| $NO_2$ (294 K) | Vandaele et al. (1998) with $I_0$ correction (slant column density (SCD) of $10^{17}$ molec. $cm^{-2}$) <br> File: no2_294K_vanDaele.xs |
| $NO_2$ (220 K) | Pre-orthogonalised Vandaele et al. (1998) with $I_0$ correction (SCD of $10^{17}$ molec. $cm^{-2}$) <br> File: no2a_220p294K_vanDaele_425-490nm.xs |
| $O_3$ (223 K) | Serdyuchenko et al. (2014) with $I_0$ correction (SCD of $10^{20}$ molec. $cm^{-2}$) <br> File: o3_223K_SDY_air.xs |
| $O_4$ (293 K) | Thalman and Volkamer (2013) <br> File: o4_thalman_volkamer_293K_inAir.xs |
| $H_2O$ | HITEMP (Rothman et al., 2010) <br> File: H2O_HITEMP_2010_390-700_296K_1013mbar_air.xs |
| Ring | Pseudo cross-section generated according to Chance and Spurr (1997) using the solar atlas of Chance and Kurucz (2010) and normalised as in Wagner et al. (2009). <br> File: Ring_QDOAScalc_HighResSAO2010_Norm.xs |
| Polynomial degree | Order 5 (six coefficients) |
| Intensity offset | Constant |

**Table A2.** DOAS settings for $NO_2$ and $O_4$ (alternative visible range).

| | |
|---|---|
| Wavelength range | 411–445 nm |
| Fraunhofer reference spectra | Noon zenith spectra averaged between 11:30:00 and 11:40:00 UT |
| Cross sections: | |
| $NO_2$ (294 K) | Vandaele et al. (1998) with $I_0$ correction (SCD of $10^{17}$ molec. cm$^{-2}$)<br>File: no2_294K_vanDaele.xs |
| $NO_2$ (220 K) | Pre-orthogonalised Vandaele et al. (1998) with $I_0$ correction (SCD of $10^{17}$ molec. cm$^{-2}$)<br>File: no2a_220p294K_vanDaele_425-490nm |
| $O_3$ (223 K) | Serdyuchenko et al. (2014) with $I_0$ correction (SCD of $10^{20}$ molec. cm$^{-2}$)<br>File: o3_223K_SDY_air.xs |
| $O_4$ (293 K) | Thalman and Volkamer (2013)<br>File: o4_thalman_volkamer_293K_inAir.xs |
| $H_2O$ | HITEMP (Rothman et al., 2010)<br>File: H2O_HITEMP_2010_390-700_296K_1013mbar_air.xs |
| Ring | Pseudo cross-section generated according to Chance and Spurr (1997) using the solar atlas of Chance and Kurucz (2010) and normalised as in Wagner et al. (2009).<br>File: Ring_QDOAScalc_HighResSAO2010_Norm.xs |
| Polynomial degree | Order 4 (five coefficients) |
| Intensity offset | Constant |

**Table A3.** DOAS settings for $NO_2$ and $O_4$ (UV range).

| | |
|---|---|
| Wavelength range | 338–370 nm |
| Fraunhofer reference spectra | Noon zenith spectra averaged between 11:30:00 and 11:40:00 UT |
| Cross sections: | |
| $NO_2$ (294 K) | Vandaele et al. (1998) with $I_0$ correction (SCD of $10^{17}$ molec. cm$^{-2}$)<br>File: no2_294K_vanDaele.xs |
| $NO_2$ (220 K) | Pre-orthogonalised Vandaele et al. (1998) with $I_0$ correction (SCD of $10^{17}$ molec. cm$^{-2}$)<br>File: no2a_220p294K_vanDaele_338-370nm.xs |
| $O_3$ (223 K) | Serdyuchenko et al. (2014) with $I_0$ correction (SCD of $10^{20}$ molec. cm$^{-2}$)<br>File: o3_223K_SDY_air.xs |
| $O_3$ (243 K) | Pre-orthogonalised Serdyuchenko et al. (2014) with $I_0$ correction (SCD of $10^{20}$ molec. cm$^{-2}$)<br>File: o3a_243p223K_SDY_338-370nm.xs |
| $O_4$ (293 K) | Thalman and Volkamer (2013)<br>File: o4_thalman_volkamer_293K_inAir.xs |
| HCHO (297 K) | Meller and Moortgat (2000)<br>File: hcho_297K_Meller.xs |
| BrO (223 K) | Fleischmann et al. (2004)<br>File: bro_223K_Fleischmann.xs |
| Ring | Pseudo cross-section generated according to Chance and Spurr (1997) using the solar atlas of Chance and Kurucz (2010) and normalised as in Wagner et al. (2009).<br>File: Ring_QDOAScalc_HighResSAO2010_Norm.xs |
| Polynomial degree | Order 5 (six coefficients) |
| Intensity offset | Constant |

**Table A4.** DOAS settings for HCHO.

| | |
|---|---|
| Wavelength range | 336.5–359 nm |
| Fraunhofer reference spectra | Noon zenith spectra averaged between 11:30:00 and 11:40:00 UT |
| Cross sections: | |
| HCHO (297 K) | Meller and Moortgat (2000)<br>File: hcho_297K_Meller.xs |
| NO$_2$ (294 K) | Vandaele et al. (1998) with $I_0$ correction (SCD of $10^{17}$ molec. cm$^{-2}$)<br>File: no2_294K_vanDaele.xs |
| O$_3$ (223 K) | Serdyuchenko et al. (2014) with $I_0$ correction (SCD of $10^{20}$ molec. cm$^{-2}$)<br>File: o3_223K_SDY_air.xs |
| O$_3$ (243 K) | Pre-orthogonalised Serdyuchenko et al. (2014) with $I_0$ correction (SCD of $10^{20}$ molec. cm$^{-2}$)<br>File: o3a_243p223K_SDY_324-359nm.xs |
| O$_4$ (293 K) | Thalman and Volkamer (2013)<br>File: o4_thalman_volkamer_293K_inAir.xs |
| BrO (223 K) | Fleischmann et al. (2004)<br>File: bro_223K_Fleischmann.xs |
| Ring | Pseudo cross-section generated according to Chance and Spurr (1997) using the solar atlas of Chance and Kurucz (2010) and normalised as in Wagner et al. (2009).<br>File: Ring_QDOAScalc_HighResSAO2010_Norm.xs |
| Polynomial degree | Order 5 (six coefficients) |
| Intensity offset | Order 1 |

**Table A5.** DOAS settings ozone in the Chappuis band.

| | |
|---|---|
| Wavelength range | 450–520 nm |
| Fraunhofer reference spectra | Noon zenith spectra averaged between 11:30:00 and 11:40:00 UT |
| Cross sections: | |
| O$_3$ (223 K) | Serdyuchenko et al. (2014) with $I_0$ correction (SCD of $10^{20}$ molec. cm$^{-2}$)<br>File: o3_223K_SDY_air.xs |
| O$_3$ (293 K) | Pre-orthogonalised Serdyuchenko et al. (2014) with $I_0$ correction (SCD of $10^{20}$ molec. cm$^{-2}$)<br>File: o3a_293p223K_SDY_450-550nm.xs |
| NO$_2$ (294 K) | Vandaele et al. (1998) with $I_0$ correction (SCD of $10^{17}$ molec. cm$^{-2}$)<br>File: no2_294K_vanDaele.xs |
| NO$_2$ (220 K) | Pre-orthogonalised Vandaele et al. (1998) with $I_0$ correction (SCD of $10^{17}$ molec. cm$^{-2}$)<br>File: no2a_220p294K_vanDaele_450-550nm.xs |
| O$_4$ (296 K) | Thalman and Volkamer (2013)<br>File: o4_thalman_volkamer_293K_inAir.xs |
| H$_2$O | HITEMP (Rothman et al., 2010)<br>File: H2O_HITEMP_2010_390-700_296K_1013mbar_air.xs |
| Ring | Pseudo cross-section generated according to Chance and Spurr (1997) using the solar atlas of Chance and Kurucz (2010) and normalised as in Wagner et al. (2009).<br>File: Ring_QDOAScalc_HighResSAO2010_Norm.xs |
| Polynomial degree | Order 5 (six coefficients) |
| Intensity offset | Order 1 |

**Table A6.** DOAS settings ozone in the Huggins band.

| | |
|---|---|
| Wavelength range | 320–340 nm |
| Fraunhofer reference spectra | Noon zenith spectra averaged between 11:30:00 and 11:40:00 UT |
| Cross sections: | |
| O$_3$ (223 K) | Serdyuchenko et al. (2014) with $I_0$ correction (SCD of $10^{20}$ molec. cm$^{-2}$)<br>File: o3_223K_SDY_air.xs |
| O$_3$ (293 K) | Pre-orthogonalised Serdyuchenko et al. (2014) with $I_0$ correction (SCD of $10^{20}$ molec. cm$^{-2}$)<br>File: o3a_293p223K_SDY_320-340nm.xs |
| O$_3$ | Non-linear correction terms (Puķīte et al., 2010)<br>Files: o3_SDY_Pukite1_320-340nm.xs and o3_SDY_Pukite2_320-340nm.xs |
| NO$_2$ (294 K) | Vandaele et al. (1998) with $I_0$ correction (SCD of $10^{17}$ molec. cm$^{-2}$)<br>File: no2_294K_vanDaele.xs |
| HCHO (297 K) | Meller and Moortgat (2000)<br>File: hcho_297K_Meller.xs |
| Ring | Pseudo cross-section generated according to Chance and Spurr (1997) using the solar atlas of Chance and Kurucz (2010) and normalised as in Wagner et al. (2009).<br>File: Ring_QDOAScalc_HighResSAO2010_Norm.xs |
| Polynomial degree | Order 3 (four coefficients) |
| Intensity offset | Order 1 |

## Appendix B:  History of slant column data set revisions

This appendix provides a history of the slant column data set resubmissions accepted after the formal deadline for participation in the semi-blind intercomparison (18 October 2016). The main motivation for accepting late revisions was to remedy well-identified mistakes. Details of the submitted revisions, including justifications for the changes and corresponding dates, are listed below.

### AIOFM (aiofm-1)

Data files were resubmitted on 16 October 2017 with two additional corrections applied, which were (1) a dark-current correction and (2) a wavelength shift that needed to be applied with respect to the reference spectrum. The O3uv data set was also resubmitted in September 2019 because an incorrect ozone cross section was used previously for the data analysis.

### AUTH (auth-3)

Data files were resubmitted on 17 March 2017. These were corrected for a systematic wavelength shift of the measured spectra.

### BIRA-IASB (bira-4)

Revised data were submitted on 28 February 2017, with small changes summarised as follows: (1) a correction of an error affecting the dark-current subtraction in the UV channel (affecting HCHO, NO2uv, O4uv and O3uv, mostly at large SZAs) and (2) an optimisation of the filtering scheme were applied. For the visible products, all measurement points having rms values exceeding 5 times the daily median rms calculated in hourly bins were excluded. The same procedure was also applied to the UV products, with any data values exceeding 4 times the median being excluded. This approach was found sufficient to exclude outliers due to an electronic instability in the UV channel.

### CHIBA (chiba-9)

Data files were resubmitted on 11 January 2018, with additional stray-light corrections applied to the measured spectra. This correction was derived as part of the wavelength calibration procedure. Considering the nominal spectral range of 310 to 525 nm, 11 discrete wavelength regions ($316 \pm 5$, $336 \pm 5$, $344 \pm 5$, $358 \pm 5$, $374 \pm 5$, $384 \pm 5$, $395 \pm 5$, $410 \pm 5$, $431 \pm 5$, $486 \pm 10$ and $518 \pm 5$ nm) were selected and analysed. In each spectral window, the spectrum was fitted using an iterative inversion method. The measurement vector consisted of the intensities measured by the MAX-DOAS instrument. The components of the state vector were set to the wavelength shift, the FWHM for the left part of an asymmetric Gaussian instrument line shape (FWHM1), the FWHM for the right part (FWHM2) and the differential slant column (dSCD) of significant absorbers ($O_3$, $NO_2$) in the analysed wavelength region. In addition, a scaling polynomial and a constant offset term (or stray-light correction term) were included in the state vector to scale the high-resolution solar spectrum data to the intensities measured by MAX-DOAS.

### CMA (cam-7, cma-8)

Revised data files were resubmitted on 26 September 2016 for CMA-7 (UV and VisSmall range) and CMA-8 (visible range). Periods with bad motor connection were filtered out in the resubmitted data. Additionally, fitting of the wavelength shift between measurement spectrum and reference spectrum was added in the revised processing.

### CU-Boulder (CU-boulder-11, CU-boulder-12)

Revised data files were submitted for all gases on 4 March 2017. For CU-boulder-11, the resubmitted data were filtered for periods with bad motor connection (when the instrument operated in 1-D or in zenith geometry), and one corrupt file was corrected. For CU-boulder-12, revised files were only submitted for gases analysed in the UV wavelengths range. Resubmitted data accounted for a time-dependent etalon identified on the UV spectrometer and fitted as a pseudo-absorber with independent shift and stretch. This approach captured the errant signal effectively at longer wavelength but was less effective at shorter wavelengths; no HCHO data were reported. The source of the etalon has since been eliminated.

### INTA (inta-17)

Revised data files were submitted on 14 February 2017, due to one change in their data analysis routine: the inverse of the actual measurement was used as the offset instead of the inverse of the reference spectrum, leading to a smaller uncertainty and improved retrievals of the sunrise and sunset slant columns. This change mainly affects twilight data, but for consistency the complete data set was reanalysed.

### KNMI (knmi-21, knmi-22)

Data files were resubmitted on 27 January 2017 with the following corrections: (1) fitting of the wavelength shift between measurement spectrum and reference spectrum was previously omitted and had to be added; (2) for knmi-22, due to an instable tripod, the logged angles can only be trusted when the horizon measurements show a consistent horizon from day to day ($< 0.5°$ difference). The measurements during all other periods were filtered out.

### LATMOS (latmos-25)

Data files were resubmitted on 4 April 2018 because the data files had to be corrected for detector non-linearity effects that

were identified after the campaign. The detector is a Hamamatsu charge-coupled device (CCD) 2048x16 type S11071-1104. The non-linearity of this detector was measured and corrected applying the procedure described in AvaSpec-DLL Manual V9.7.0.0 (pp. 71–73). A stable light source (Xe lamp, VG9 filter and diffuser) was used to measure spectra at different integration times between 50 and 1830 ms. The maximum level of the elementary spectrum varies from 400 to 16 000 counts. The correlation between the flux (count $s^{-1}$) and the number of counts of an elementary spectrum at several pixels was fitted by a polynomial of degree 7, and this curve was then used to correct raw data as recommended by Avantes.

### LMU-MIM (lmumim-35)

Data files were resubmitted with two corrections applied on 24 March 2018. (1) The spectra were reanalysed with a correction for detector non-linearity and the analysis was updated by using offset and dark-current spectra. The latter spectra were measured after CINDI-2 and also corrected for detector non-linearity. (2) The instrumental slit function was determined from measured spectra using the fitting facility available from the QDOAS retrieval software (see http://uv-vis.aeronomie.be/software/QDOAS/, last access: 8 April 2020), while for the originally submitted data set a fixed instrument slit function measured with a Hg lamp was used.

### LuftBlick/NASA (knmi-23, luftblick-26, 27, 260, 270, nasa-31, 32)

Revised data sets were submitted on 4 October 2017. Pandora data during CINDI-2 were processed using BlickP, the native Pandonia Global Network (PGN) software. BlickP allows for the fitting of molecular absorption cross sections of a specific species represented in terms of constant, or linear, or quadratic functions of temperature. Orthogonalisation of cross sections is not allowed. Pandora $NO_2$ and $O_3$ slant columns had to be recalculated to "simulate" the case where cross sections of the same gas at different temperatures are used in the fitting. In addition, measurements at azimuth angles of 95 and 135° at an elevation angle of 1° were eliminated due to obstruction. There was also a mistake in the intensity calibration correction in the original submission.

### NIWA (niwa-30)

Data files were resubmitted for $NO_2$ in the visible and UV range and for HCHO on 27 March 2017. The data were reprocessed to include a test that detects any bad timing on a spectrum and removes the results for that spectrum. This occasional fault was likely due to last-minute logging program changes to enable the one available spectrometer to switch wavelengths between the visible and UV regions every quarter of an hour.

### NUST (nust-33)

Data files were resubmitted on 10 February 2017, after exploring the relatively larger rms values. A misalignment of elevation angles was noticed in the analyses due to the malfunctioning of the Peltier controller unit and loose gear of the stepper motor. On 15 September 2016, the instrument was replaced with a new instrument no. 15306 (where a problem with the slit was identified and was adjusted). The new instrument functioned properly, but there was no lamp experiment to adjust the azimuth direction until 19 September 2017. Systematic high rms values are observed for all elevation angles in the retrieved NO2visSmall (411–445 nm) and HCHO dSCDs for the period of 12–17 September 2016. Finally, on 19 September 2016, a lamp experiment was performed, and the data showed a relatively large improvement in rms values from 20 September 2016 onward. After extensive checks and quality control, the retrieved slant columns were only submitted for a limited number of days.

*Data availability.* All CINDI-2 data sets are available on request from the various instrument principal investigators.

*Supplement.* The supplement related to this article is available online at: https://doi.org/10.5194/amt-13-1-2020-supplement.

*Author contributions.* FH, MVR, AA, UF, AR, TW, JL and AP designed, planned and organised CINDI-2. KK was the referee for the intercomparison during the campaign and for the follow-up data analysis. ED contributed as the assistant referee. All co-authors contributed to the campaign either as participants and instrument operators and/or by performing the data analysis, data quality control and data submission. MVR performed the intercomparison data analysis. KK and MVR interpreted the results and wrote the paper with feedback and contributions from all other co-authors.

*Competing interests.* The authors declare that they have no conflict of interest.

*Acknowledgements.* We gratefully acknowledge the KNMI staff at Cabauw for their excellent technical and infrastructure support during the campaign.

*Financial support.* CINDI-2 received funding from the Netherlands Space Office (NSO). Funding for this study was provided by ESA through the CINDI-2 (ESA contract no. 4000118533/16/I-Sbo) and FRM$_4$DOAS (ESA contract no. 4000118181/16/I-EF) projects and partly within the EU 7th Framework Programme QA$_4$ECV project (grant agreement no. 607405). The BOKU MAX-DOAS instrument was funded and the participation of Stefan F. Schreier was supported by the Austrian Science Fund (FWF): I 2296-N29. The participation of the University of Toronto team was supported by the Canadian Space Agency (through the AVATARS project) and the Natural Sciences and Engineering Research Council (through the PAHA project). The instrument was primarily funded by the Canada Foundation for Innovation and is usually operated at the Polar Environment Atmospheric Research Laboratory (PEARL) by the Canadian Network for the Detection of Atmospheric Change (CANDAC). Funding for CISC was provided by the UVAS ("Ultraviolet and Visible Atmospheric Sounder") projects SEOSAT/INGENIO, ESP2015-71299-R, MINECO-FEDER and UE. The activities of the IUP-Heidelberg were supported by the DFG project RAPSODI (grant no. PL 193/17-1). SAOZ and Mini-SAOZ instruments are supported by the Centre National de la Recherche Scientifique (CNRS) and the Centre National d'Etudes Spatiales (CNES). INTA recognises support from the National funding projects HELADO (CTM2013-41311-P) and AVATAR (CGL2014-55230-R). AMOIAP recognises support from the Russian Science Foundation (grant no. 16-17-10275) and the Russian Foundation for Basic Research (grant nos. 16-05-01062 and 18-35-00682). Ka L. Chan received transnational access funding from ACTRIS-2 (H2020 grant agreement no. 654109). Rainer Volkamer recognises funding from NASA's Atmospheric Composition Program (NASA-16-NUP2016-0001) and the US National Science Foundation (award AGS-1620530). Henning Finkenzeller is the recipient of a NASA graduate fellowship. Mihalis Vrekoussis recognises support from the University of Bremen and the DFG Research Center/Cluster of Excellence "The Ocean in the Earth System-MARUM". Financial support through the University of Bremen Institutional Strategy in the framework of the DFG Excellence Initiative is gratefully appreciated for Anja Schönhardt. Pandora instrument deployment was supported by Luftblick through the ESA Pandonia Project and NASA Pandora Project at the Goddard Space Flight Center under NASA Headquarters' Tropospheric Composition Program. The article processing charges for this open-access publication were covered by BK Scientific CE1.

*Review statement.* This paper was edited by Saulius Nevas and reviewed by two anonymous referees.

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

**Remarks from the language copy-editor**

CE1   Please confirm the changes to this section.