# Peer review of "Intercomparison of NO2, O4, O3 and HCHO slant column measurements by MAX-DOAS and zenith-sky UV-Visible spectrometers during the CINDI-2 campaign"

_Atmospheric Measurement Techniques, 2019_

## Referee Comment (RC1) · Anonymous Referee #1 · 19 Aug 2019

Review of Kreher et al. - Intercomparison of NO2, O4, O3 and HCHO slant column measurements by MAX-DOAS and zenith-sky UV-Visible spectrometers during the CINDI-2 campaign

GENERAL COMMENTS

As clearly stated in the title, this manuscript presents results from the 2016 "CINDI-2" intercomparison campaign relating to retrievals of key trace species (NO2, O4, O3 and HCHO) using either MAX-DOAS or zenith sky UV/Visible spectrometers. These types

of measurements have grown to considerable importance in the field of atmospheric composition in recent years, and are expected to continue to increase rapidly in number and range of applications, making a very careful campaign such as CINDI-2 of great interest to a broad community. Importantly, the major types of instruments now in widespread use (such as Pandora, SAOZ, the former EnviMeS MAX-DOAS and the Hoffman mini-DOAS) all participated in the campaign which ensures the relevance of the CINDI-2 results to the actual measurements being made around the world.

The manuscript is comprehensive and clearly written, and many of the author team are among the world experts in this field, and overall, I believe is very suitable for publication in AMT.

I do have a number of general comments and questions. I believe it will help the reader better understand the philosophy and approach of CINDI-2 if each of these could be briefly addressed in either the introduction or the discussion section of the manuscript.

1. It is evident that while great attention was paid to ensure the consistency of certain aspects of the measurements and retrievals, other aspects – which would also affect the results - were left to the individual groups. I am sure the decisions of the organisers in this regard were made with thought but it is not always clear to the reader what the motivation was for the different inclusions and exclusions and how these related to the stated aims.

2. To what extent, can the results of the intercomparison obtained in idealised and tightly co-ordinated conditions be applied the operational, geographically-distributed real-world measurement sites? Recommendations for the networks seem minimal (elevation scans are mentioned).

3. Limited of course by my own experience, it seems quite unusual for an intercomparison to be carried out without a designated reference instrument or standard, and instead to use the median of the participants as a reference. (Although in the case of formaldehyde a subgroup of better-performing instruments is identified and so this is

closer to an orthodox reference group). As far as I can see, this means there can be no traceability of any of the measurements? I would also add that in places I found the text has the potential to be misleading by referring to "the reference" in the abstract and conclusions, which readers might read in isolation to the rest of the paper.

4. In many places the manuscript notes the efforts made to eliminate spatial and temporal mismatches between the participating instruments, but this does not seem linked to the scales of temporal and spatial variability expected for these species, and indeed, in section 3.7 it seems NO2 varies on a finer scale.

5. From time to time the stated aims seem to interfere with each other. To really understand the differences between instruments requires a somewhat different approach compared to undertaking a strict performance evaluation, particularly if the aim is to simulate realistic conditions in the field. This point is closely related to (1) about the overall design of the exercise and what is or isn't being evaluated.

SPECIFIC COMMENTS

Lines 9-12 The "major aims" don't quite agree with what appears later (Section 2.3, page 5 lines 31-32).

Lines 12-14 I don't see how you can do "trend analysis" without traceability to a standard?

Line 20 The word "unprecedented" seems over hyped

Line 25 "bias and offset of the individual data sets against the reference". I think this is likely to mislead the reader of the abstract because it implies the existence of a reference instrument.

Lines 23-26 This seems like the "reproducibility" in usual metrological terms.

Line 28 ". . . a quantitative assessment of the measurement performance" – it seems to

me more like the "consistency" ?

Line 38 "The interest of ESA for . . ." change to either "The interest of ESA in " or "The desire of ESA for" or similar.

Lines 40-41 "planned at the horizon 2022-2023" – I don't know what this phrase means sorry.

Line 7 Touching again on the philosophy of CINDI-2, it seems to me just the consistency, there are other aspects of "high quality" needed for "long-term measurements, trend analysis and satellite data validation".

Line 8 ". . . it is essential . . . to contribute to a harmonisation" – it can't be "essential" to "contribute"! These seem to be aims (1) and (3) from the abstract re-worded.

Line 9-10 Did you in fact contribute to a harmonisation of the measurement settings and retrieval methods outside of the intercomparison itself, ie for the networks to use in practice?

Lines 5-10 This is very interesting in terms of the philosophy of CINDI-2. It is stated some groups performed more advanced pre-processing, but in general, as far as I can tell, the results from these groups was not weighted any differently from groups that didn't do these steps. Is that logical?

Lines 9-10 Rather than standardise these steps, wouldn't it be more valuable to assess their contribution to better results?

Lines 9-10 Could this be something to recommend to field instruments?

Line 14 "containers". For the first time this word appears, I suggest "shipping contain-
ers", and also the first time it appears in the captions (Figure 1). After the first time, just "container" would be ok. A "container" out of context could be of any size.

Line 14 "temporary containers were rented" – I would prefer "shipping containers were rented and temporarily installed".

Line 24 Strictly, 287 degrees isn't WNW, which is 292.5 degrees from north.

Line 24 Rather than "N=0", it would be clearer to say "north"

Line 29 Change "Like in" to "As in "

Lines 30-32 The objectives don't quite match the three listed earlier (such as in the abstract). Now there are only two.

Lines 31-32 The second objective was previously to "discuss the performance" now it is to "define a robust methodology for performance assessment". Is it to define a methodology or to apply it?

Lines 36-39 It is interesting that the retrieval settings and parameters were specified but not the software. I am struggling to understand the logic of this. I think this decision is worth more explanation. It would be possible to compare a purely raw instrumental quantity, wouldn't it?

Lines 1-7 This is another curious feature of the design of the campaign. To me there seems a conflict between the daily meetings which help understand better what is going on, and the strictness of the campaign designed to assess performance. In the field this luxury would certainly not be available.

Line 14 "operation" should be "operational"

Line 27 The sentence "The convention for the azimuth angle . . . " appears in the wrong place

Line 28 "synchronicity" should be "synchronisation" (unless we are talking about Jung or pop music from the early 1980s)

Line 32 I would have thought "an NDACC" rather than "a NDACC" (but this is because I am expecting the reader to read "NDACC" as "en dack".)

Line 12 "unprecedented" seems over-hyped to me – don't you really just mean that it was "improved" or "greatly improved" since CINDI-1?

Line 13 "synchroncity" -> "synchronisation"

Line 14 ". . . the impact of atmospheric noise on the data comparisons could be reduced to a minimum" - How do you know though that the level of co-ordination is enough though? Do you know what time scales and spatial scales you expect the species to vary over? Later on, you imply that actually the co-ordination was not sufficient for N20.

Line 29 Could you have mandated separate times for UV and visible?

Line 34 I don't think "MPIC" has previously been defined.

Page 7 line 30 – Page 8 line 4

Presumably however none of this, except (3), would be available in a field setting? This to me seems a conflict between the different aims of CINDI-2.

Line 12 "we used" – until now the manuscript has been written using the traditional third person passive voice.

Lines 25-38 There doesn't seem to be any mention of the type of location Cabauw is in terms of rural versus urban and the expected pollution levels.

Lines 18-19 Some of the instruments show a drift over the course of the campaign. Should we therefore expect instruments in the field also to show potentially significant drifts over time?

Lines 35-38 The decision to allow resubmissions is also interesting – I assume the justification is that these types of mistakes would be able to be identified and corrected independently by the instrument operator in a network setting?

Lines 6-14 This seems to create a problem though, because in the field, this would not generally be possible?

Line 25 -"drastically reduced" – that would depend on the temporal and spatial variability though?

Line 26 "and/or atmospheric variability" – I don't understand what you mean here. The sentence seems to contradict itself to me. The sampling and mis-match errors are only small or large relative to the spatial and temporal scale of atmospheric variability. If the comparison noise is caused by atmospheric variability then isn't that a mismatch?

Line 33 "similar as performed" -> "similar to as performed" or "similar to those performed"

Lines 22-30 The implication is that the fit residuals should represent a lower bound to the measurement uncertainty, but perhaps another sentence of justification is needed for this.

Line 30 – If the real NO2 is varying on short scales that in itself is not an error of the measurement, but it would affect the agreement with a given satellite pixel.

Line 39 "keeps" should be "stays"

Lines 1-2 ". . . for this molecule most of the residual variance between good instruments can be explained by measurement noise" needs re-wording. I think I know what you mean but the words by themselves don't make much sense.

Line 9 Replace "a couple" by "two".

Line 19 Replace "largest" with "the largest"

Lines 18-21 This must be very relevant for field instruments?

Line 30 ". . . specific limits have been set. . ." You should add something like ". . . specific limits have been set to use for performance evaluation". The way it is now, it takes the reader some time to work out what these limits are all about.

Lines 28-37

Intuitively, I don't find this approach very reasonable. It seems you choose limits some-what arbitrarily (or at least let's say making use of subjective judgement), and then go through a binary pass or fail evaluation. Especially in figure 19, some of the dots which pass seem to be right on the limit, and some of the failed points fall only just outside it. I appreciate for network use such as NDACC there might need to be a definite threshold, but otherwise the use of pass/fail seems to degrade the information you have gained through the experiment. Perhaps you could discuss this point briefly.

Line 1 "statistic" should be "statistics" if I've understood the sentence correctly.

Lines 11-17 Just repeating an earlier comment, the use of green versus orange when the two instruments could be a distance of epsilon other side of an arbitrary line seems odd to me. The use of pink for being four times outside the limit makes more sense.

Lines 37-38 I thought the DOAS settings were all prescribed?

Line 8 "wavelengths" should be "wavelength"

Line 9 A better wording might be "and only failed to satisfy one criterion in the O4 .."

Lines 8-10 I suggest breaking this sentence into two parts for easier comprehension.

Lines 23-24 I suggest replacing "at the same time they are meeting" with "at the same time meet"

Line 24 Replace "On the opposite" with a phrase such as "On the other hand" or "Conversely".

Line 25 "satisfies" should be "satisfy"

Line 1 ". . . a reduction in of the atmospheric changes on the intercomparison exercise." A reduction compared to what? (CINDI-1 I assume).

Line 4 "very well coordinated" sounds like boasting to me!

Line 14 ". . . with a selected reference" seems misleading to me, because it implies a specified reference instrument, which was not part of the intercomparison.

Lines 23-25 "The median bias against the reference is generally low . . .". Again I think this might mislead the reader who hasn't read the whole paper, who would assume there was a particular reference instrument.

Line 30 Replace "&" (ampersand symbol) with the word "and".

Line 33 Personally, I don't think you can say "guideline" in the singular like this, but others might disagree.

Line 34 Replace "like the one" with "such as the one".

Line 4 "instruments" should have an apostrophe - "instruments' " or re-word to "the elevation point calibration of instruments".

Lines 7-8 "a thoroughly planned and carefully managed campaign" sounds like boasting to me.

Lines 18-26 This sounds really good and would be very valuable to the community.

Page 32 (Figure 3)

The individual plots are very small but adequate for qualitative use of the figure.

Page 40 (Figure 11)

Delete the unwanted carriage return in the caption.

Page 46 (Figure 17)

"The dashed lines indicate the limits . . . " For the caption, you need to provide more information, in particular that these limits have been chosen (rather than derived), for the sake of distinguishing outliers.

Page 49 (Figure 20)

In my printed version of the manuscript I find the pink and orange a little bit hard to distinguish. (The green and orange have excellent contrast. )

Page 51 (Figure 22)

The numbers in the green boxes are quite hard to read, and also to some extent those in the red and orange boxes.
* * *

---

## Referee Comment (RC2) · Anonymous Referee #2 · 20 Aug 2019

This manuscript is a well written and extensive intercomparison between UV-Visible spectrometers during a field study with a highly refined strategy. The work demonstrates very good agreement between slant column densities of the gases mentioned in the title during the campaign. These efforts are necessary for understanding agreement between instruments and for use in subsequent profile retrievals and satellite validation. The work is clearly relevant to Atmospheric Measurement Techniques and I recommend that it be published with minor revisions. Below are general and then specific comments.

General comment:

The manuscript goes through extensive procedures that were designed to synchronize measurements to be of the same volume of air at the same time. This synchronization has been improved as compared to the prior campaign, and results are improved. This result indicates that there are significant variations in the actual slant column densities at the same elevation angles if viewed at even slightly different times. The result is not surprising for short-lived pollution gases that probably have a variety of nearby sources, but it indicates that subsequent inversions to vertical concentration profiles and vertical column densities may have challenges due to variations in the vertical concentration profile that occur during the measurement profile. This point is discussed on page 12, lines 21-31, but is not given as much importance as is necessary for this finding. On the other hand, it seems that this point may be the origin of the "conclusion" on lines 13-14 of page 17 that the design "was not fully adequate for profile inversion experiments". This conclusion should be removed or reworded because the present work does not show inversion experiments and thus cannot conclude on them. If the point was meant to be that variability in space and time is observed, then that is a conclusion. Please make clear both the important point of variability in time and space and discuss relevance for inversions, but do not conclude about inversions that are not shown here.

Specific comments:

Page 3, line 34. It should be discussed here that when the instruments that measure profiles sequentially at un-synchronized field studies (as they will typically be used after CINDI-2) that the variability during the profile will affect profile inversions. Potentially the Boesch et al. (2018) AMT paper could be cited.

Page 4, line 24. The Apituley et al. manuscript to be submitted to AMT is really important to the present publication. Is this manuscript submitted? If it is not submitted by the time of this manuscript being decided upon, details should be added here.

Page 5, lines 9-10. The suggestion for future studies should be in the discussion rather than here. Potentially giving an indication to "see section N.M" would be appropriate.

Page 6, line 31. Please give the approximate solar zenith angles of these UTC cutoffs so that they can be more easily translated to other work.

Page 7, line 14. The text says "atmospheric noise", but this effect is not noise but variability given later analysis. Reword.

Page 10, lines 23-29. It may be appropriate to note that retrievals using a zenith reference spectrum within the same elevation sequence (rather than a fixed noon reference) often reduces difficulty in fitting, and thus more instruments could get useful HCHO data if other analysis methods were used.

Page 12, line 24. The word "noise" is used, but this effect is not noise, but rather "variability" due to viewing different airmasses (in time or space).

Page 12, line 39. Replace "keeps larger" with "remains larger".

Page 13, line 16. Change "dependency" to "dependence".

Figure 7 needs a color/symbol key

Table A1. The reference to Vandaele et al. (1998) is not in the references. The paper that I believe is cited seems to indicate the spectrum is at 294K rather than 298K. Please clarify this citation and temperature. This citation and temperature occur in other appendices. Please assure that all sources are fully cited in these appendix tables.

---

## Author Response (AR1)

**Response to Anonymous Referee #1**

We would like to thank the reviewer for their comprehensive and thoughtful review, and helpful comments which are addressed individually in the response below. The reviewer's comments are included in blue & italics.

*GENERAL COMMENTS*
*As clearly stated in the title, this manuscript presents results from the 2016 "CINDI-2" intercomparison campaign relating to retrievals of key trace species (NO2, O4, O3 and HCHO) using either MAX-DOAS or zenith sky UV/Visible spectrometers. These types of measurements have grown to considerable importance in the field of atmospheric composition in recent years, and are expected to continue to increase rapidly in number and range of applications, making a very careful campaign such as CINDI-2 of great interest to a broad community. Importantly, the major types of instruments now in widespread use (such as Pandora, SAOZ, the former EnviMeS MAX-DOAS and the Hoffman mini-DOAS) all participated in the campaign which ensures the relevance of the CINDI-2 results to the actual measurements being made around the world.*

*The manuscript is comprehensive and clearly written, and many of the author team are among the world experts in this field, and overall, I believe is very suitable for publication in AMT.*

*I do have a number of general comments and questions. I believe it will help the reader better understand the philosophy and approach of CINDI-2 if each of these could be briefly addressed in either the introduction or the discussion section of the manuscript.*

*1. It is evident that while great attention was paid to ensure the consistency of certain aspects of the measurements and retrievals, other aspects – which would also affect the results - were left to the individual groups. I am sure the decisions of the organisers in this regard were made with thought but it is not always clear to the reader what the motivation was for the different inclusions and exclusions and how these related to the stated aims.*

This comment touches on a very important topic and most of the choices were motivated by findings of the first CINDI campaign and MADCAT. There definitely are reasons why some aspects of the intercomparison exercise were prescribed (such as the measurement schedule and the retrieval settings) while others were not (the analysis code and some of the calibration procedures). The organisers of the CINDI-2 intercomparison were aiming at providing a procedure that (1) forced every participating instrument to look simultaneously in the same direction (and to do this as precisely as practically achievable) and hence sample the same airmass and (2) to prescribe the use of analysis settings that were as consistent as realistically possible to enforce a more coordinated analysis.

One step further would have been to also prescribe the analysis software but allowing the individual groups to stick with their own preferred analysis software (which most participants would continue to use after CINDI-2 anyway) led to a more realistic intercomparison, and hence to a more realistic assessment of the participating instrument/group by using the combination of individual instrument plus individually used analysis software but prescribing all other settings and procedures.

Main reasons for not enforcing strict guidelines for the calibration steps were that (1) some of the key calibration steps (wavelength registration and slit function determination) can be obtained in the field using solar lines and dedicated software, (2) calibration facilities were not available to analyse other key instrumental responses, such as stray-light level, detector linearity response or polarization response, and (3) some neglected calibration steps are of minor importance for DOAS-type retrievals (e.g. radiometric response). However, the possibility to address better the missing aspects, and in

particular calibration related issues, will be considered when preparing future campaigns. A short paragraph has been added at the end of Section 2.2 (Campaign design) to motivate better why a lot of effort was spent on certain aspects.

*2. To what extent, can the results of the intercomparison obtained in idealised and tightly co-ordinated conditions be applied the operational, geographically-distributed real-world measurement sites? Recommendations for the networks seem minimal (elevation scans are mentioned).*

This is also a very important comment and helpful feed-back for us. The NDACC UV/Vis Working Group provides recommendations for measurements and data analysis which are mandatory for the inclusion of an instrument (and station) into the NDACC network. These recommendations (referred to as NDACC UV/Vis Appendix) have been substantially updated to also include guidelines for MAX-DOAS measurements and data analysis, and they will be published on the NDACC web page by the end of December 2019. A short statement has been added to address this, which is quoted under item 5 further below.
We have also added a separate section entitled 'Recommendations for network operation and future campaigns' before the conclusions. A part of the conclusions has been moved into this section and some addition text addressing this comment has also been added.

*3. Limited of course by my own experience, it seems quite unusual for an intercomparison to be carried out without a designated reference instrument or standard, and instead to use the median of the participants as a reference. (Although in the case of formaldehyde a subgroup of better performing instruments is identified and so this is closer to an orthodox reference group). As far as I can see, this means there can be no traceability of any of the measurements? I would also add that in places I found the text has the potential to be misleading by referring to "the reference" in the abstract and conclusions, which readers might read in isolation to the rest of the paper.*

We have clarified the use of reference data sets in the text. Reasons for why we used the median of the participating instruments rather than one single instrument (or a small group of instruments) is to keep the comparison fairer and not to 'favour' a couple of instruments. If an instrument with an absolute calibration would have been available then using that instrument would certainly have made sense, but there is no absolute reference for such measurements. The approach adopted here is similar to what was used in previous UV-Vis intercomparisons, i.e. identifying a group of mutually consistent instruments and use the median from their measurements as a best estimate ('most probable') of the 'true' value. For $NO_2$, it appeared that a large number of instruments were found to be in mutual agreement within limits derived from previous campaigns, for HCHO only a small sub-group presented a satisfactory level of agreement.

*4. In many places the manuscript notes the efforts made to eliminate spatial and temporal mismatches between the participating instruments, but this does not seem linked to the scales of temporal and spatial variability expected for these species, and indeed, in section 3.7 it seems NO2 varies on a finer scale.*

Efforts were made to substantially improve the spatial and temporal coincidence between measurements, in comparison to what was done in previous campaigns. However, practical limitations (related to the large variety of participating instruments) also had to be considered. It was found – a posteriori – that the scale of variability of $NO_2$ was in fact small enough to still dominate the variance of the measurements (despite the fact that these measurements were synchronized to better than one minute in time and all telescope pointing in the same azimuthal direction within a few degrees of accuracy, and in the same elevation to better than 1 degree). And this has also been stated in Section 3.7. E.g. the following sentence has been added: 'This means that in this intercomparison, atmospheric

variability limits the reproducibility and representativeness of individual MAX-DOAS measurements for species such as NO$_2$.'

*5. From time to time the stated aims seem to interfere with each other. To really understand the differences between instruments requires a somewhat different approach compared to undertaking a strict performance evaluation, particularly if the aim is to simulate realistic conditions in the field. This point is closely related to (1) about the overall design of the exercise and what is or isn't being evaluated.*

To address and clarify this point, we have added a statement in Section 1, paragraph 4, that the aim of the intercomparison is '... to assess the participating instruments in their ability to retrieve the same geophysical quantities (i.e. slant columns of NO$_2$, O$_4$, HCHO and O$_3$) when measured and processed in a controlled way (i.e. using a prescribed measurement protocol and retrieval settings)'.

We have also added/changed the following statement in the conclusion so that is now reads: ' This assessment process, undertaken as part of the CINDI-2 intercomparison campaign, provides the UV-visible absorption spectroscopy research community with guidelines and a procedure on how to assess the performance of MAX-DOAS and DOAS instruments, in particular for the inclusion into NDACC (see NDACC webpage for access to the UV/Vis Appendix describing these recommendations). It is expected that a similar level of consistency, as seen during CINDI-2, can be obtained in the field if recommended settings are implemented and used by each participant of the network. More control in this aspect of homogeneity can be obtained through centralized processing, which is the aim of the currently developed ESA FRM4DOAS project (see http://frm4doas.aeronomie.be/).'

*SPECIFIC COMMENTS*

*Page 2*
*Lines 9-12 The "major aims" don't quite agree with what appears later (Section 2.3, page 5 lines 31-32).*

We agree with the reviewer (thanks very much for picking this up) and have changed the text in the abstract and in Section 2.3 to be consistent.

*Lines 12-14 I don't see how you can do "trend analysis" without traceability to a standard?*

For trend analysis, the measurement precision and its stability in time (i.e. making sure that measurements are not affected by drifts or discontinuities of any type) should be most important. This means that suitability for trend analysis cannot be determined from a campaign in isolation, since an instrument showing a perfect behavior during two weeks can always be affected by longer term drifts or biases once in operation. However, successful participation to successive campaigns is one way to verify stability. This is e.g. the approach used in the Dobson/Brewer network communities. Another possible approach is to regularly operate traveling standard instruments at the different sites of a network.

*Line 20 The word "unprecedented" seems over hyped*

We have changed this to 'unique'.

*Line 25 "bias and offset of the individual data sets against the reference". I think this is likely to mislead the reader of the abstract because it implies the existence of a reference instrument.*

We agree and have changed the text to '.... the selected refence (which is the median of either all data or a subset), ...'

*Lines 23-26 This seems like the "reproducibility" in usual metrological terms.*

As far as I understand, this is correct. However, we used here the same mathematical terms previously used in UV/Vis instrument intercomparisons to be consistent with the analysis performed e.g. during the first CINDI or earlier intercomparisons.

*Line 28 ". . . a quantitative assessment of the measurement performance" – it seems to me more like the "consistency" ?*

We have changed the text to: 'It introduces a quantitative assessment of the consistency between all the participating instruments for the MAX-DOAS and zenith-sky DOAS techniques.' If an instrument was not performing well, this could be clearly identified.

*Page 3*
*Line 38 "The interest of ESA for . . ." change to either "The interest of ESA in " or "The desire of ESA for" or similar.*

Done.

*Lines 40-41 "planned at the horizon 2022-2023" – I don't know what this phrase means sorry.*

This phrase has been deleted.

*Page 4*
*Line 7 Touching again on the philosophy of CINDI-2, it seems to me just the consistency, there are other aspects of "high quality" needed for "long-term measurements, trend analysis and satellite data validation".*

See response to the corresponding comment above.

*Line 8 ". . . it is essential . . . to contribute to a harmonisation" – it can't be "essential" to"contribute"! These seem to be aims (1) and (3) from the abstract re-worded.*

We agree that this wasn't worded well and the text has been changed to accommodate the comment. The part of the sentence "… and to contribute to a harmonisation of the measurement settings and retrieval methods." has been deleted.

*Line 9-10 Did you in fact contribute to a harmonisation of the measurement settings and retrieval methods outside of the intercomparison itself, ie for the networks to use in practice?*

Yes, we did and this has been incorporated in the updated NDACC UV/Vis Appendix (Protocol for NDACC UV/Vis instrument operation and data analysis) which will be published on the NDACC web page later this month (Dec 2019). This has been added under Conclusions (paragraph 5).

*Page 5*
*Lines 5-10 This is very interesting in terms of the philosophy of CINDI-2. It is stated some groups performed more advanced pre-processing, but in general, as far as I can tell, the results from these groups was not weighted any differently from groups that didn't do these steps. Is that logical?*

Yes, it actually is. Since many of the instruments can differ in the detail of their particular setup, it would have been difficult to fairly assess the instruments performance on grounds of pre-processing without really looking thoroughly at each of the instruments and its pre-calibration features. However, it would certainly be valuable if future campaigns would look into the pre-processing and calibration of the instruments in a more coordinated way (e.g. through organization of a calibration campaign ahead of the field campaign) and this has now also been added in a new section dealing specifically with recommendations based on the CINDI-2 results and experiences.

*Lines 9-10 Rather than standardise these steps, wouldn't it be more valuable to assess their contribution to better results?*

That is a good point and we have addressed this by adding additional text under the new Section 5 (Recommendations for network operation and future campaigns), last paragraph (bullet #2).

*Lines 9-10 Could this be something to recommend to field instruments?*

Yes, it certainly can and the NDACC UV/Vis Appendix also contains information on further documentation containing guidelines for calibrations which will shortly also be available on the NDACC UV/Vis working group web site as well and is currently available here:
http://frm4doas.aeronomie.be/ProjectDir/Deliverables/FRM4DOAS_D4_MAXDOAS_Best_Practices_Document_20180110_v1_0.pdf

*Line 14 "containers". For the first time this word appears, I suggest "shipping containers", and also the first time it appears in the captions (Figure 1). After the first time, just "container" would be ok. A "container" out of context could be of any size.*

We have changed this to: '… mobile units (similar to shipping containers) were temporarily installed for the campaign period.' The containers are not strictly speaking shipping containers but 'mobile units' which look similar to shipping containers.

*Line 14 "temporary containers were rented" – I would prefer "shipping containers were rented and temporarily installed".*

We have changed the text accordingly (see above).

*Line 24 Strictly, 287 degrees isn't WNW, which is 292.5 degrees from north.*

True, that is strictly speaking correct and we have added 'approximately'. We were working of a table that stated that WNW is associated with angles between $281.25^{\circ} - 303.75^{\circ}$.

*Line 24 Rather than "N=0", it would be clearer to say "north"*

Agreed and this has been changed accordingly.

*Line 29 Change "Like in" to "As in "*
*Done.*

*Lines 30-32 The objectives don't quite match the three listed earlier (such as in the abstract). Now there are only two.*

This was already previously raised (first comment of the 'specific comments section') and has been changed in the text so it is consistent in Section 2.3 and the abstract.

*Lines 31-32 The second objective was previously to "discuss the performance" now it is to "define a robust methodology for performance assessment". Is it to define a methodology or to apply it?*

The objective is to define a methodology which is then also applied to the CINDI-2 data products. The text has been changed accordingly.

*Lines 36-39 It is interesting that the retrieval settings and parameters were specified but not the software. I am struggling to understand the logic of this. I think this decision is worth more explanation. It would be possible to compare a purely raw instrumental quantity, wouldn't it?*

We understand were the reviewer is coming from but since all analysis software packages basically solve the same mathematical equations (which are part of the DOAS technique), the differences lie in the details of the implementation (in particular wavelength registration issues) rather than in the

actual analysis software. Hence the approach to harmonize and prescribe the settings as much as possible but allow for individual software packages to be used.

*Page 6*
*Lines 1-7 This is another curious feature of the design of the campaign. To me there seems a conflict between the daily meetings which help understand better what is going on, and the strictness of the campaign designed to assess performance. In the field this luxury would certainly not be available.*

It is certainly correct that in the field, it is often not possible to get this kind of feed-back and the semi-blind intercomparison procedure is in this regard a compromise between (1) a strict 'blind' intercomparison which would not allow for any exchange of information between the participants and (2) the opportunity especially (but not only!) for more inexperienced participants to gain a lot of experience and knowledge, and if possible, to have an independent referee intervene if there is an obvious problem with instrumentation that can be fixed (e.g. a problem with the hardware, such as the elevation pointing). The information provided at the daily meetings also encouraged the participants to be more engaged in the intercomparison overall without giving away how well their individual measurements were doing.

*Line 14 "operation" should be "operational"*
Done.

*Line 27 The sentence "The convention for the azimuth angle . . . " appears in the wrong place*
We agree and this has been fixed; the explanation is now been provided earlier on under Section 2.2.

*Line 28 "synchronicity" should be "synchronisation" (unless we are talking about Jung or pop music from the early 1980s)*
Fair enough and done.

*Line 32 I would have thought "an NDACC" rather than "a NDACC" (but this is because I am expecting the reader to read "NDACC" as "en dack".)*
Agreed & done.

*Page 7*
*Line 12 "unprecedented" seems over-hyped to me – don't you really just mean that it was "improved" or "greatly improved" since CINDI-1?*
We appreciate the comment and have reworded the sentence accordingly.

*Line 13 "synchroncity" -> "synchronisation"*
Done.

*Line 14 ". . . the impact of atmospheric noise on the data comparisons could be reduced to a minimum" - How do you know though that the level of co-ordination is enough though? Do you know what time scales and spatial scales you expect the species to vary over? Later on, you imply that actually the co-ordination was not sufficient for N20.*
Good point and we have toned the statement down accordingly.

*Line 29 Could you have mandated separate times for UV and visible?*
Possibly, but we were not aware of that issue when the measurement schedule was designed, and this would also have meant that it would have affected everybody's schedule not just the Pandora instruments.

*Line 34 I don't think "MPIC" has previously been defined.*

The full name has been added in brackets.

*Page 7 line 30 – Page 8 line 4*
*Presumably however none of this, except (3), would be available in a field setting? This to me seems a conflict between the different aims of CINDI-2.*

Both, (2) and (3) should be straight forward to implement in a field application. All this is discussed in much more detail in Donner et al., 2019 which has been submitted and is currently under review (the reference has been updated accordingly). It is certainly true that option (1) requires the availability of a strong lamp but for a campaign such as CINDI-2, this was definitely a very valuable additional test and helped each of the groups to find out more about the accuracy of the elevation pointing of their instrument.

*Page 8*
*Line 12 "we used" – until now the manuscript has been written using the traditional third person passive voice.*

Agreed & this has been changed to the passive form.

*Lines 25-38 There doesn't seem to be any mention of the type of location Cabauw is in terms of rural versus urban and the expected pollution levels.*

Good point. We have added a short description of the Cabauw measurement site under Section 2:

"In short, the CESAR site at Cabauw is overall a rural site, with only a few pollution sources nearby, but the wider vicinity of Cabauw is densely populated, with the cities of Utrecht, Amsterdam, The Hague and Rotterdam less than 60 km away and a dense highway grid within 25 km, so that the site experiences recurring pollution events, e.g. such as from the daily morning and afternoon rush hours."

*Page 9*
*Lines 18-19 Some of the instruments show a drift over the course of the campaign. Should we therefore expect instruments in the field also to show potentially significant drifts over time?*

Possibly, and CINDI-2 really helped us to appreciate how important it is for the measurement quality to verify the accuracy and stability of the elevation scans. This also means that in the field, it is important to regularly monitor the accuracy of the elevation scans to avoid any drift, bias or discontinuity in data series, hence we made a recommendation to this end (2nd paragraph on conclusions).

*Lines 35-38 The decision to allow resubmissions is also interesting – I assume the justification is that these types of mistakes would be able to be identified and corrected independently by the instrument operator in a network setting?*

Yes, that is correct. For a resubmission, the groups had to state clearly what mistakes they made and how they were remedied. Admittedly, in a real word situation (e.g. due to time constraints) we might not always look carefully enough at our data sets but if we would, we should be able to identify and correct the issues which were identified.

*Page 10*
*Lines 6-14 This seems to create a problem though, because in the field, this would not generally be possible?*

We don't quite understand why this is a problem. We don't mean to imply that everything we applied during the intercomparison has to be 100% reproducible in a field situation and we think it is ok that

we create somewhat more idealized conditions which show us how well we can agree if we pay attention and get everything is right as possible.

*Line 25 -"drastically reduced" – that would depend on the temporal and spatial variability though?*

Good point. We have toned the text down somewhat and changed 'drastically' into 'considerably' and changed 'should accurately reflect' to 'should more accurately reflect'

*Line 26 "and/or atmospheric variability" – I don't understand what you mean here. The sentence seems to contradict itself to me. The sampling and mis-match errors are only small or large relative to the spatial and temporal scale of atmospheric variability. If the comparison noise is caused by atmospheric variability then isn't that a mismatch?*

We agree with the reviewer and have deleted 'and/or atmospheric variability'.

*Line 33 "similar as performed" -> "similar to as performed" or "similar to those performed"*

Done.

*Lines 22-30 The implication is that the fit residuals should represent a lower bound to the measurement uncertainty, but perhaps another sentence of justification is needed for this.*

A statement to this effect has been added.

*Line 30 – If the real NO2 is varying on short scales that in itself is not an error of the measurement, but it would affect the agreement with a given satellite pixel.*

We agree and we expect that the scale of variability of $NO_2$ is much smaller than the scale of any $NO_2$ satellite measurement. The main issue is therefore to assess the representativeness of correlative measurements for comparison to satellite data.

*Line 39 "keeps" should be "stays"*

Done. We have changed it to 'remains'.

*Lines 1-2 ". . . for this molecule most of the residual variance between good instruments can be explained by measurement noise" needs re-wording. I think I know what you mean but the words by themselves don't make much sense.*

We have reworded the sentence to '… for this molecule most of the residual variance from regressions involving good instruments can be explained by instrument shot noise.'

*Line 9 Replace "a couple" by "two".*

Done.

*Line 19 Replace "largest" with "the largest"*

Done.

*Lines 18-21 This must be very relevant for field instruments?*

Yes, we agree. This is already covered under Section 3.4 so we didn't want to repeat it here again.

*Line 30 ". . . specific limits have been set. . ." You should add something like ". . . specific limits have been set to use for performance evaluation". The way it is now, it takes the reader some time to work out what these limits are all about.*

Done.

*Lines 28-37*
*Intuitively, I don't find this approach very reasonable. It seems you choose limits somewhat arbitrarily (or at least let's say making use of subjective judgement), and then go through a binary pass or fail evaluation. Especially in figure 19, some of the dots which pass seem to be right on the limit, and some of the failed points fall only just outside it. I appreciate for network use such as NDACC there might need to be a definite threshold, but otherwise the use of pass/fail seems to degrade the information you have gained through the experiment. Perhaps you could discuss this point briefly.*

We agree that this is not straight forward and a bit of a delicate issue as well. Even though we tried to introduce some elements of objectivity, the choice of a limit is fundamentally arbitrary (but not totally subjective since it is based on statistical arguments). Since the limits were chosen to exceed the median of the measurements (this has now also been added to Figure 18 and in the text), instruments that exceed them can be seen as "out of the norm". This does not necessarily mean that such measurements are problematic and this is why we are checking several parameters. Failing in one parameter, especially if very close to the limit is not a problem per se but failing in two or more usually is.

*Page 14*
*Line 1 "statistic" should be "statistics" if I've understood the sentence correctly.*

Done.

*Lines 11-17 Just repeating an earlier comment, the use of green versus orange when the two instruments could be a distance of epsilon other side of an arbitrary line seems odd to me. The use of pink for being four times outside the limit makes more sense.*

See discussion above.

*Lines 37-38 I thought the DOAS settings were all prescribed?*

This sentence has been deleted, AIOFM have made a mistake and chose the wrong ozone cross-section. They have re-analysed their data with the correct ozone cross-section and we have updated the figures correspondingly. This did affect Figures 13, 18, 20 and 22, and Fig-S22, Fig-S23 and Fig-S24 from the Supplement, and some text in Appendix B.

*Page 15*
*Line 8 "wavelengths" should be "wavelength"*

Done.

*Line 9 A better wording might be "and only failed to satisfy one criterion in the O4 .."*

Done.

*Lines 8-10 I suggest breaking this sentence into two parts for easier comprehension.*

Done.

*Lines 23-24 I suggest replacing "at the same time they are meeting" with "at the same time meet"*

Done.

*Line 24 Replace "On the opposite" with a phrase such as "On the other hand" or "Conversely".*

Done.

*Line 25 "satisfies" should be "satisfy"*

Done.

*Page 16*
*Line 1 ". . . a reduction in of the atmospheric changes on the intercomparison exercise." A reduction compared to what? (CINDI-1 I assume).*

We agree that this needs to be fixed and have added to the sentence so it reads: '… atmospheric changes on the intercomparison exercise in comparison to CINDI.'

*Line 4 "very well coordinated" sounds like boasting to me!*

Fair enough and we have dropped this phrase.

*Line 14 ". . . with a selected reference" seems misleading to me, because it implies a specified reference instrument, which was not part of the intercomparison.*

We agree and have changed the text accordingly: '… with a reference data set was performed (see Section 3.5 for details on how the reference data sets were derived) …'

*Lines 23-25 "The median bias against the reference is generally low . . .". Again I think this might mislead the reader who hasn't read the whole paper, who would assume there was a particular reference instrument.*

To clarify this, we have replaced '… the reference…' with '… the reference data sets …'

*Line 30 Replace "&" (ampersand symbol) with the word "and".*

Done.

*Line 33 Personally, I don't think you can say "guideline" in the singular like this, but others might disagree.*

Agreed and changed.

*Line 34 Replace "like the one" with "such as the one".*

The sentence has been changed and the phrase has been dropped.

*Page 17*
*Line 4 "instruments" should have an apostrophe - "instruments' " or re-word to "the elevation point calibration of instruments".*

Done.

*Lines 7-8 "a thoroughly planned and carefully managed campaign" sounds like boasting to me.*

Fair enough and this has been toned down in the text and this sentence has been moved to the new Section 5 (Recommendations …).

*Lines 18-26 This sounds really good and would be very valuable to the community.*

The feedback is much appreciated and this will clearly be considered in the design of the next UV/Vis intercomparison. This has now also been moved into Section 5.

*Page 32 (Figure 3)*
*The individual plots are very small but adequate for qualitative use of the figure.*

That was the intention. If ok with the reviewer, we would prefer to leave the plot as is.

*Page 40 (Figure 11)*
*Delete the unwanted carriage return in the caption.*

Done.

*Page 46 (Figure 17)*
*"The dashed lines indicate the limits . . . " For the caption, you need to provide more information, in particular that these limits have been chosen (rather than derived), for the sake of distinguishing outliers.*

Done.

*Page 49 (Figure 20)*
*In my printed version of the manuscript I find the pink and orange a little bit hard to distinguish. (The green and orange have excellent contrast. )*

Good point, we have fixed this by replacing pink with black.

*Page 51 (Figure 22)*
*The numbers in the green boxes are quite hard to read, and also to some extent those in the red and orange boxes.*

We agree and have fixed this figure so that the numbers are much clearer to see.

**Response to Anonymous Referee #2**

We would like to thank the reviewer for their thorough review and helpful comments which are addressed individually in the response below. The reviewer's comments are included in blue and italics.

*This manuscript is a well written and extensive intercomparison between UV-Visible spectrometers during a field study with a highly refined strategy. The work demonstrates very good agreement between slant column densities of the gases mentioned in the title during the campaign. These efforts are necessary for understanding agreement between instruments and for use in subsequent profile retrievals and satellite validation. The work is clearly relevant to Atmospheric Measurement Techniques and I recommend that it be published with minor revisions. Below are general and then specific comments.*

*General comment:*
*The manuscript goes through extensive procedures that were designed to synchronize measurements to be of the same volume of air at the same time. This synchronization has been improved as compared to the prior campaign, and results are improved. This result indicates that there are significant variations in the actual slant column densities at the same elevation angles if viewed at even slightly different times. The result is not surprising for short-lived pollution gases that probably have a variety of nearby sources, but it indicates that subsequent inversions to vertical concentration profiles and vertical column densities may have challenges due to variations in the vertical concentration profile that occur during the measurement profile. This point is discussed on page 12, lines 21-31, but is not given as much importance as is necessary for this finding.*

To emphasis this finding further, we have added a brief summary of what has been discussed in Section 3.7 (former page 12, lines 21-31) to the conclusions as part of the 1. bullet point.

*On the other hand, it seems that this point may be the origin of the "conclusion" on lines 13-14 of page 17 that the design "was not fully adequate for profile inversion experiments". This conclusion should be removed or reworded because the present work does not show inversion experiments and thus cannot conclude on them. If the point was meant to be that variability in space and time is observed, then that is a conclusion. Please make clear both the important point of variability in time and space and discuss relevance for inversions, but do not conclude about inversions that are not shown here.*

We have changed the sentence as suggested and added more discussion to this first bullet point (partly also covered by the response to the comment above).

*Specific comments:*
*Page 3, line 34. It should be discussed here that when the instruments that measure profiles sequentially at un-synchronized field studies (as they will typically be used after CINDI-2) that the variability during the profile will affect profile inversions. Potentially the Boesch et al. (2018) AMT paper could be cited.*

This is an important point and one of the CINDI-2 companion papers on profile retrievals, 'Intercomparison of MAX-DOAS vertical profile retrieval algorithms: studies on field data from the CINDI-2 campaign' by Tirpitz et al. (see also entry in the reference list) which has just been submitted to AMT, would be the more appropriate publication for this discussion. A brief discussion has also been added in Section 5, 1. bullet point under 'Despite these achievements, a few critical points were identified that deserve more attention in future deployments.'

*Page 4, line 24. The Apituley et al. manuscript to be submitted to AMT is really important to the present publication. Is this manuscript submitted? If it is not submitted by the time of this manuscript being decided upon, details should be added here.*

Since Apituley et al. is not yet submitted, we have added some information re the measurement site (CESAR) and the CINDI-2 campaign in general:

'In short, the CESAR site at Cabauw is overall a rural site, with only a few pollution sources nearby, but the wider vicinity of Cabauw is densely populated, with the cities of Utrecht, Amsterdam, The Hague and Rotterdam less than 60 km away and a dense highway grid within 25 km, so that the site experiences recurring pollution events, e.g. such as from the daily morning and afternoon rush hours.

The MAX-DOAS instruments were also complemented with a suite of in-situ, profiling and mobile observations which are described in detail by Apituley et al. (to be submitted to AMT, 2019). In particular, a long-path DOAS measuring near surface mixing ratios of $NO_2$ and HCHO but also a range of other species such as HONO and $SO_2$ (see e.g. Merten et al, 2011, for a description of the technique) was operated at the CESAR site for the period of the campaign. Several mobile MAX-DOAS measurements were also made around Cabauw, and between Rotterdam and Utrecht (e.g. Merlaud, 2013). in addition to the static ones. $NO_2$ profiles were measured with $NO_2$ sondes (Sluis et al, 2010) and lidar (e.g. Volten et al., 2009), as well as through in-situ observations using the Cabauw meteorological tower. Extensive aerosol information was also gathered using Raman aerosol lidar and in situ samplers.'

*Page 5, lines 9-10. The suggestion for future studies should be in the discussion rather than here. Potentially giving an indication to "see section N.M" would be appropriate.*

We agree and since this suggestion is also discussed as part of the previous Conclusions section, now part of the newly added Section 5 (Recommendations for network operation and future campaigns), at the end (2nd bullet point), we have deleted this sentence.

*Page 6, line 31. Please give the approximate solar zenith angles of these UTC cutoffs so that they can be more easily translated to other work.*

This information has been added.

*Page 7, line 14. The text says "atmospheric noise", but this effect is not noise but variability given later analysis. Reword.*

This has been reworded as suggested.

*Page 10, lines 23-29. It may be appropriate to note that retrievals using a zenith reference spectrum within the same elevation sequence (rather than a fixed noon reference) often reduces difficulty in fitting, and thus more instruments could get useful HCHO data if other analysis methods were used.*

We agree with the reviewer that using a sequential reference spectrum can potentially reduce instrumental effects, or the impact of misfits to strong absorbers like $O_3$. However, this has not really been true for CINDI-2 and the agreement seems worse, most likely because noise is added due to the fact that not all instruments are able to capture the sequential reference exactly in the same way.

*Page 12, line 24. The word "noise" is used, but this effect is not noise, but rather "variability" due to viewing different airmasses (in time or space).*

We have changed the wording from 'noise' to 'difference between the individual data sets'.

*Page 12, line 39. Replace "keeps larger" with "remains larger".*

Done.

*Page 13, line 16. Change "dependency" to "dependence".*

Done.

*Figure 7 needs a color/symbol key*

This has been added as requested.

*Table A1. The reference to Vandaele et al. (1998) is not in the references. The paper that I believe is cited seems to indicate the spectrum is at 294K rather than 298K. Please clarify this citation and temperature. This citation and temperature occur in other appendices. Please assure that all sources are fully cited in these appendix tables.*

We agree with the reviewer and have added the reference and corrected the citations.

[revised manuscript text omitted]

**1 MAX-DOAS regression results**

This section presents detailed results from regression analyses performed for the eight MAX-DOAS data products. In each sub-section below, three plots are provided, showing respectively:

- Scatter plots of the regression between individual data sets and median reference values for all measurement days and all viewing and elevation directions (similar to Figures 10, 11 and 12 of the main manuscript).
- Overview plots of the slope, intercept and RMS from regression analysis for all measurement days and viewing directions, and for several elevation angles (1°, 3°, 5°, 8°, 15°, and 30°) (similar to Figure 15 of the main manuscript).
- Summary overview plots of the slope, intercept and RMS from regression analysis for all measurement days and all viewing and elevation directions. These summarize the details of the performance assessment results, as described in Figure 17 of the main manuscript.

**1.1 MAX-DOAS results for NO₂ in the visible range (NO2vis)**

[Figure]

**Figure S1: Regression analysis for NO₂ dSCDs (measured in the visible wavelength region), corresponding to Figure 10 in the main manuscript.**

[Figure]

**Figure S2: Slope, Intercept and RMS of NO₂ dSCDs against those of the median reference data set, for each instrument measuring NO₂ in the visible range. Colours refer to elevation angles shown top right. This figure is corresponding to Figure 17 in the main manuscript.**

[Figure]

**Figure S3: Summary of the regression statistic for NO₂ in the visible range, showing the slope, intercept and RMS values as displayed in Figure S1. The dashed lines show the performance limits as defined in Table 5 of the main manuscript. The values within these limits are plotted in blue, the ones falling outside the limit in red. This figure is corresponding to Figure 17 in the main manuscript.**

**1.2    MAX-DOAS results for NO₂ in the small visible range (NO2visSmall)**

[Figure]

**Figure S4: Regression analysis for NO₂ dSCDs (measured in the small visible wavelength region).**

[Figure]

**Figure S5: Slope, Intercept and RMS of NO₂ dSCDs against those of the median reference data set, for each instrument measuring NO₂ in the small visible range. Colours refer to elevation angles shown top right.**

[Figure]

**Figure S6: Summary of the regression statistic for NO₂ in the small visible range, showing the slope, intercept and RMS values as displayed in Figure S4. The dashed lines show the performance limits as defined in Table 5 of the main manuscript. The values within these limits are plotted in blue, the ones falling outside the limit in red.**

**1.3    MAX-DOAS results for NO₂ in the UV range (NO2uv)**

Figure S7: Regression analysis for NO₂ dSCDs (measured in the UV wavelength region).

[Figure]

**Figure S8: Slope, Intercept and RMS of NO₂ dSCDs against those of the median reference data set, for each instrument measuring NO₂ in the UV range. Colours refer to elevation angles shown top right.**

[Figure]

**Figure S9: Summary of the regression statistic for NO₂ in the UV range, showing the slope, intercept and RMS values as displayed in Figure S7. The dashed lines show the performance limits as defined in Table 5 of the main manuscript. The values within these limits are plotted in blue, the ones falling outside the limit in red.**

**1.4 MAX-DOAS results for O₄ in the visible range (O4vis)**

[Figure]

**Figure S10: Regression analysis for O₄ dSCDs (measured in the visible wavelength region), corresponding to Figure 11 in the main manuscript.**

[Figure]

**Figure S11: Slope, Intercept and RMS of O₄ dSCDs against those of the median reference data set, for each instrument measuring O₄ in the visible range. Colours refer to elevation angles shown top right.**

[Figure]

**Figure S12: Summary of the regression statistic for O₄ in the visible range, showing the slope, intercept and RMS values as displayed in Figure S10. The dashed lines show the performance limits as defined in Table 5 of the main manuscript. The values within these limits are plotted in blue, the ones falling outside the limit in red.**

**1.5 MAX-DOAS results for O₄ in the UV range (O4uv)**

[Figure]

**Figure S13: Regression analysis for O₄ dSCDs (measured in the UV wavelength region).**

[Figure]

**Figure S14: Slope, Intercept and RMS of O₄ dSCDs against those of the median reference data set, for each instrument measuring O₄ in the UV range. Colours refer to elevation angles shown top right.**

[Figure]

**Figure S15: Summary of the regression statistic for O₄ in the UV range, showing the slope, intercept and RMS values as displayed in Figure S13. The dashed lines show the performance limits as defined in Table 5 of the main manuscript. The values within these limits are plotted in blue, the ones falling outside the limit in red.**

**1.6    MAX-DOAS results for HCHO**

[Figure]

**Figure S16: Regression analysis for HCHO dSCDs, corresponding to Figure 12 in the main manuscript.**

[Figure]

**Figure S17: Slope, Intercept and RMS of HCHO dSCDs against those of the median reference data set, for each instrument measuring HCHO. Colours refer to elevation angles shown top right.**

[Figure]

**Figure S18: Summary of the regression statistic for HCHO, showing the slope, intercept and RMS values as displayed in Figure S16. The dashed lines show the performance limits as defined in Table 5 of the main manuscript. The values within these limits are plotted in blue, the ones falling outside the limit in red.**

**1.7    MAX-DOAS results for O₃ in the visible range (O3vis)**

[Figure]

**Figure S19: Regression analysis for O₃ dSCDs (measured in the visible wavelength region).**

[Figure]

**Figure S20: Slope, Intercept and RMS of O₃ dSCDs against those of the median reference data set, for each instrument measuring O₃ in the visible range. Colours refer to elevation angles shown top right.**

[Figure]

**Figure S21: Summary of the regression statistic for O₃ in the visible range, showing the slope, intercept and RMS values as displayed in Figure S19. The dashed lines show the performance limits as defined in Table 5 of the main manuscript. The values within these limits are plotted in blue, the ones falling outside the limit in red.**

**1.8 MAX-DOAS results for O₃ in the UV range (O3uv)**

**O3uv, MAX-DOAS regression, 0°<SZA<100°**

[Figure]

dSCD from participating group (x10$^{18}$ molec/cm$^2$)

Median reference dSCD (x10$^{18}$ molec/cm$^2$)

[Figure]

**Figure S22: Regression analysis for O₃ dSCDs (measured in the UV wavelength region).**

[Figure]

O3uv, regression analysis, 0°<SZA<100°, All azimuths

[Figure]

[Figure]

**Figure S23: Slope, Intercept and RMS of O₃ dSCDs against those of the median reference data set, for each instrument measuring O₃ in the UV range. Colours refer to elevation angles shown top right.**

[Figure]

O3uv, regression analysis, 0°<SZA<100°, All azimuths

[Figure]

[Figure]

**Figure S24: Summary of the regression statistic for $O_3$ in the UV range, showing the slope, intercept and RMS values as displayed in Figure S22. The dashed lines show the performance limits as defined in Table 5 of the main manuscript. The values within these limits are plotted in blue, the ones falling outside the limit in red.**

**2      Zenith-sky twilight regression results**

This section presents detailed results from regression analyses performed for the four zenith-sky twilight data products. In each sub-section below, two plots are provided, showing respectively:

- scatter plots of the regression between individual data sets and median reference values for all measurement days,
- summary overview plots of the slope, intercept and RMS from regression analysis for all measurement days. These summarize the details of the performance assessment results for zenith-sky twilight measurements as performed within NDACC.

**2.1      Zenith-sky results for NO₂ in the visible range (NO2vis)**

[Figure]

**Figure S25: Regression analysis for zenith-sky NO₂ dSCDs (measured in the visible wavelength region).**

[Figure]

**Figure S26: Summary of the regression statistic for zenith-sky NO₂ in the visible range, showing the slope, intercept and RMS values as displayed in Figure S25. The dashed lines show the performance limits as defined in Table 5 of the main manuscript. The values within these limits are plotted in blue, the ones falling outside the limit in red.**

NO2visSmall, zenith-sky twilight regression, 75°<SZA<93°

**Figure S27: Regression analysis for zenith-sky NO₂ dSCDs (measured in the small visible wavelength region).**

[Figure]

**Figure S28: Summary of the regression statistic for zenith-sky NO₂ in the small visible range, showing the slope, intercept and RMS values as displayed in Figure S27. The dashed lines show the performance limits as defined in Table 5 of the main manuscript. The values within these limits are plotted in blue, the ones falling outside the limit in red.**

**2.3    Zenith-sky results for NO₂ in the UV range (NO2uv)**

[Figure]

**Figure S29: Regression analysis for zenith-sky NO₂ dSCDs (measured in the UV wavelength region).**

[Figure]

**Figure S30: Summary of the regression statistic for zenith-sky NO₂ in the UV range, showing the slope, intercept and RMS values as displayed in Figure S29. The dashed lines show the performance limits as defined in Table 5 of the main manuscript. The values within these limits are plotted in blue, the ones falling outside the limit in red.**

**2.4    Zenith-sky results for O₃ in the visible range (O3vis)**

[Figure]

Figure S31: Regression analysis for zenith-sky O₃ dSCDs (measured in the visible wavelength region).

[Figure]

**Figure S32: Summary of the regression statistic for zenith-sky O₃ in the visible range, showing the slope, intercept and RMS values as displayed in Figure S31. The dashed lines show the performance limits as defined in Table 5 of the main manuscript. The values within these limits are plotted in blue, the ones falling outside the limit in red.**

**3 Description and technical characteristics of the CINDI-2 MAX-DOAS and zenith-sky DOAS systems**

This section presents the description of all the participating instruments. The following colour coding is used for the different types: yellow for Zenith-sky DOAS, blue for 1D MAX-DOAS and green for 2D MAX-DOAS. The instruments are listed in alphabetical order with respect to their institute acronym which is included in the top of each instrument table as part of the institute name (see also Table 1 in the main manuscript).

| | |
|---|---|
| **Institute:** Anhui Institute of Optics and Fine Mechanics, Chinese Academy of Sciences (**AIOFM**), Hefei, China

**Responsible person(s):** Ang Li, Pinhua Xie

**Contact details:** angli@aiofm.ac.cn, phxie@aiofm.ac.cn | |
| **Instrument type:** 2D MAX-DOAS | **Nr:**
CINDI-2.01 |

| | |
|---|---|
| **Overall design of the instrument** | **Optical head including telescope:** separated; elevation and azimuth angles fully configurable
**Spectrometer type:** Princeton Instrument 150i
**Detector type:** Princeton Instrument PIXIS-2K BUV
**Optical fibers:** quartz optical fiber, length: 10 m
**Filters:** ZWB3(=UG5)
**Mirrors:** no
**Temperature control of spectrometer/detector:** 35°C /-30°C |
| **Instrument performance** | **Spectral range/resolution:** 290-380 (adjustable)/0.35 nm
**Azimuthal scan/direct-sun capabilities:** yes/no
**Elevation angle capability:** fully configurable
**Field of view:** 0.2°
**Typical integration time:** 10-60s
**Typical scan duration:** 15 minutes |
| **Calibration/characterization procedures** | **Elevation angles:** inclinometer
**Field of view:** scanning over a light source in the laboratory
**Straylight:**
**Dark signal:** by using the shutter
**Line shape:** Hg lamp in the laboratory
**Polarization:** -
**Detector nonlinearity:** halogen lamp/dark background
**Pixel-to-pixel variability**: halogen lamp/dark background |
| **Spectral analysis software** | QDOAS / WinDOAS |
| **Supporting measurements** | Video camera, inclinometer, GPS, electronic compass |
| **Reference** | Wang Yang, Li Ang, Xie Pin-Hua, Chen Hao, Xu Jin, Wu Feng-Cheng, Liu Jian-Guo, Liu Wen-Qing: Retrieving vertical profile of aerosol extinction by multi-axis differential optical absorption spectroscopy, Acta Phys. Sin., 62(18), 180705, http://dx.doi.org/10.7498/aps.62.180705, 2013. |

| | |
|---|---|
| **Institute:** A.M.Obukhov Institute of Atmospheric Physics (**AMOIAP**), Russian Academy of Sciences, Moscow, Russia

**Responsible person(s):** Alexander Borovski, Oleg V.Postylyakov

**Contact details:** alexander.n.borovski@gmail.com
oleg.postylyakov@gmail.com |
[Figure]
 |

| **Instrument type:** 2-port DOAS | **Nr:**
CINDI-2.02 |
|---|---|

| | |
|---|---|
| **Overall design of the instrument** | **Optical head including telescope:** separated; 2 telescope units (one for zenith + one for off-axis)
**Spectrometer type:** Shamrock303i spectrograph with filter wheel
**Detector type:** Newton CCD (DU940N-BU2, 2048×512 pxls)
**Optical fibers:** standard fiber cable with two inputs and one output, length: 25 m
**Filters:** Andover Corp. filter S86FG11-25 (transmittion from 320 to 700 nm)
**Mirrors:** no
**Temperature control of spectrometer/detector:** 35°C/-40°C |
| **Instrument performance** | **Spectral range/resolution VIS1:** 420-490 / 0.4 nm
**Spectral range/resolution VIS2:** 395-465 / 0.4 nm
**Spectral range/resolution VIS3:** 390-530 / 0.9 nm
**Spectral range/resolution UV:** 315-385 / 0.4 nm
**Azimuthal scan/direct-sun capabilities:** no/no
**Elevation angle capability:** two fixed elevation angles (one zenith and one 5°)
**Field of view:** 0.3°
**Typical integration time:** 1 – 10 s
**Typical scan duration:** 30 – 40 s |
| **Calibration/characterization procedures** | **Elevation angles:** adjusted manually using bubble and digital levels
**Field of view:** measured in the lab
**Straylight:** unknown
**Dark signal:** using unilluminated parts of the detector
**Line shape:** Hg lamp in the lab, FWHM adjusted during spectra analysis
**Polarization:** n/a (use of long depolarizing fiber bundle)
**Detector nonlinearity:** unknown
**Pixel-to-pixel variability**: unknown |
| **Spectral analysis software** | Andor Solis/own-developed software |
| **Supporting measurements** | n/a |
| **Reference** | I. Bruchkouski, A. Borovski, A. Elokhov, and O. Postylyakov. A layout of two-port DOAS system for investigation of atmospheric trace gases based on laboratory spectrograph, Proc. SPIE, 10035, 100353C, https://doi.org/10.1117/12.2248634, 2016. |

| | |
|---|---|
| **Institute:** Physics Department, Section of Applied and Environmental Physics, Laboratory of Atmospheric Physics, Aristotle University of Thessaloniki (**AUTH**), Thessaloniki, Greece

**Responsible person(s):** Alkiviadis Bais, Theano Drosoglou

**Contact details:** abais@auth.gr, tdroso@auth.gr |
[Figure]
 |

| **Instrument type:** Phaethon mini MAX-DOAS | **Nr:**
CINDI-2.03 |
|---|---|

| | |
|---|---|
| **Overall design of the instrument** | **Optical head including telescope:** separated; elevation and azimuth angles fully configurable
**Spectrometer type:** AvaSpec-ULS2048LTEC (Avantes)
**Detector type:** SONY2048L (CCD linear array)
**Optical fibers:** standard fiber cable with metal silicone jacketing, 800 µm fiber core diameter and overall length of 8 meters
**Filters:** filter wheel: neutral density filter + ground quartz diffuser plate for direct-sun, clear aperture for sky-radiance, opaque for dark signal
**Mirrors:** no mirrors, plano-convex lens
**Temperature control of spectrometer/detector:** 5°C/5°C |
| **Instrument performance** | **Spectral range/resolution:** 297-452/0.3-0.4 nm
**Azimuthal scan/direct-sun capabilities:** yes/yes
**Elevation angle capability:** fully configurable, 0.125° resolution
**Field of view:** 1°
**Typical integration time:** 200-3000 ms (scattered light)
**Typical scan duration:** 10-20 minutes for a sequence of elevation angles |
| **Calibration/characterization procedures** | **Elevation angles:** Sighting using the solar disk
**Field of view:** white reflecting stripe measurements in laboratory
**Straylight:** tunable-laser measurements
**Dark signal:** after each scan sequence for all integration times used
**Line shape:** laser lines and spectral discharge lamp measurements
**Polarization:** zenith radiance measurements at different azimuth angles
**Detector nonlinearity:** tunable-laser measurements with varying output
**Pixel-to-pixel variability**: tungsten halogen lamp measurements |
| **Spectral analysis software** | QDOAS (currently version 2.109.3) |
| **Supporting measurements** | None during the campaign |
| **Reference** | Drosoglou, T., A. F. Bais, I. Zyrichidou, N. Kouremeti, A. Poupkou, N. Liora, C. Giannaros, M. E. Koukouli, D. Balis, and D. Melas (2017), Comparisons of ground-based tropospheric NO2 MAX-DOAS measurements to satellite observations with the aid of an air quality model over the Thessaloniki area, Greece, Atmos. Chem. Phys., 17(9), 5829-5849; http://dx.doi.org/ 10.5194/acp-17-5829-2017. |

| | |
|---|---|
| **Institute:** Royal Belgian Institute for space Aeronomy (**BIRA-IASB**), Brussels, Belgium

**Responsible person(s):** Christian Hermans and Michel Van Roozendael

**Contact details:** christh@aeronomie.be, michelv@oma.be |
[Figure]
 |
| **Instrument type:** 2D MAX-DOAS | **Nr:**
CINDI-2.04 |

| | |
|---|---|
| **Overall design of the instrument** | **Optical head including telescope:** separated; elevation and azimuth angles fully configurable; active sun tracking system
**Spectrometer type UV:** Newport, model: 74086
**Spectrometer type vis:** Horiba, model: Micro HR
**Detector type UV:** CCD Back-illuminated Princeton Instrument Pixis 2K
**Detector type vis:** CCD Back-illuminated Princeton Instrument Pixis 100
**Optical fibers:** quartz
UV chanel**:** monofiber (l:6m,diam:1000µm)+ bundle(length:2m, 51 fibers 100µm)
Vis chanel**:** monofiber (l:6m,diam:800µm)+ bundle(length:2m, 37 fibers  100µm)
**Filters:**  UV chanel : Filter band U-340 Hoya
**Mirrors:** no (for telescope we use lens in quartz)
**Temperature control of spectrometer and detector UV:** 30°C/-50°C
**Temperature control of spectrometer and detector vis:** 30°C/-50°C |
| **Instrument performance** | **Spectral range/resolution UV:** 300–390/0.4 nm
**Spectral range/resolution vis:** 405–540/0.7 nm
**Azimuthal scan/direct-sun capabilities:** yes/yes
**Elevation angle capability:** fully configurable; resolution: <0.1°
**Field of view:** <1°
**Typical integration time:** total measurement t:60 sec (t min: vis 0.03s, UV 0.1s)
**Typical scan duration:** 20 minutes |
| **Calibration/characterization procedures** | **Elevation angles:** digital inclinometer in telescope
**Field of view:** white light source in lab
**Straylight:**  double monochromator fed by white light source
**Dark signal:**  measured as night every day
**Line shape:** HgCd lamp in the lab, further adjusted using QDOAS
**Polarization:** n/a (use of long depolarising fiber bundle)
**Detector nonlinearity:** white light source in the lab
**Pixel-to-pixel variability**: white light source in the lab |
| **Spectral analysis software** | QDOAS |
| **Supporting measurements** | Video camera |
| **Reference** | Clémer, K., Van Roozendael, M., Fayt, C., Hendrick, F., Hermans, C., Pinardi, G., Spurr, R., Wang, P., and De Mazière, M.: Multiple wavelength retrieval of tropospheric aerosol optical properties from MAXDOAS measurements in Beijing, Atmos. Meas. Tech., 3, 863-878, https://doi.org/10.5194/amt-3-863-2010, 2010. |

| | |
|---|---|
| **Institute:** Institute of Meteorology, University of Natural Resources and Life Sciences (**BOKU**), Vienna, Austria

**Responsible person(s):** Stefan Schreier

**Contact details:** Stefan.Schreier@boku.ac.at |
[Figure]
 |
| **Instrument type:** 1 channel scientific grade elevation and azimuth scanning MAX-DOAS | **Nr:**
CINDI-2.06 |

| | |
|---|---|
| **Overall design of the instrument** | **Optical head including telescope:** separated; elevation and azimuth angles fully configurable

**Spectrometer type:** Acton Standard Series SP-2356 Imaging Spectrograph

**Detector type:** PIX100B-SF-Q-F-A

**Optical fibers:** Y-type quartz bundle, diameter: 150µm, length: 25m

**Filters:** no

**Mirrors:** no

**Temperature control of spectrometer and detector:** 35°C/-60°C |
| **Instrument performance** | **Spectral range/resolution:** 419–553/0.8 nm

**Azimuthal scan/direct-sun capabilities:** yes/no

**Elevation angle capability:** fully configurable

**Field of view:** 0.8°

**Typical integration time:** 30s (off-axis); 60s (zenith)

**Typical scan duration:** 10 minutes for 10 elevation angles |
| **Calibration/characterization procedures** | **Elevation angles:** geometric alignment of telescope, horizon scan

**Field of view:** white light source in lab

**Straylight:** not yet characterized

**Dark signal:** nightly measurements

**Line shape:** HgCd lamp in telescope

**Polarization:** -

**Detector nonlinearity:** white light source in lab, characterization only
**Pixel-to-pixel variability:** white light source in lab, characterization only |
| **Spectral analysis software** | NLIN |
| **Supporting measurements** | Video camera, HgCd lamp |
| **Reference** | Schreier et al., Multiple ground-based MAX-DOAS observations in Vienna, Austria – part 1: Evaluation of horizontal and temporal NO2, HCHO, and CHOCHO distributions and comparison with independent data sets, to be submitted to ACP (2019) |

| | |
|---|---|
| **Institute:** Belarusian State University (**BSU**), Minsk, Belarus
**Responsible person(s):** Ilya Bruchkouski
**Contact details:** bruchkovsky2010@yandex.by |
[Figure]
 |
| **Instrument type:** MAX-DOAS one azimuth, catadioptric telescope / MARS-B | **Nr:**
CINDI-2.05 |

| | |
|---|---|
| **Overall design of the instrument** | **Optical head including telescope:** integrated
**Spectrometer type:** Oriel MS257 imaging spectrograph (1:4)
**Detector type:** Andor DV420-OE 256*1024 pixels CCD
**Optical fibers:** n/a
**Filters:** red
**Mirrors:** yes
**Temperature control of detector:** -40°C |
| **Instrument performance** | **Spectral range/resolution:** 409-492/0.4 nm + possibly also UV
**Azimuthal scan/direct-sun capabilities:** no/no
**Elevation angle capability:** fully configurable
**Field of view:** 0.2° (azimuth); 1° (elevation)
**Typical integration time:** 1-3s
**Typical scan duration:** 1.5 minutes (12 elevation angles) |
| **Calibration/characterization procedures** | **Elevation angles:** Udo Friess method (laser level, narrow mercury lamp)
**Field of view:** measured in the lab
**Straylight:** N/A
**Dark signal:** 485 ±6 counts
**Line shape:** Gaussian
**Polarization:** N/A
**Detector nonlinearity:** above 25000 counts
**Pixel-to-pixel variability**: ±6 counts |
| **Spectral analysis software** | Self-made + Windoas |
| **Supporting measurements** | Video camera (possibly) |
| **Reference** | I. Bruchkouski, V. Dziomin, A. Krasouski. Seasonal variability of the atmospheric trace constituents in Antarctica / I. Bruchkouski [et al.] // Abs. 35-th Canadian Symposium of Remote Sensing (IGARSS-2014), Quèbec, 13-18 July / General Chair Dr. Monique Bernier. – Quebec, 2014. – P. 4098-4100. |

| | |
|---|---|
| **Institute:** Center for Environmental Remote Sensing (CEReS), Chiba University (**CHIBA**), Chiba, Japan

**Responsible person(s):** Hitoshi Irie

**Contact details:** hitoshi.irie@chiba-u.jp |
[Figure]
 |
| **Instrument type:** 1 channel scientific grade elevation and azimuth scanning MAX-DOAS | **Nr:**
CINDI-2.09 |

| | |
|---|---|
| **Overall design of the instrument** | **Optical head including telescope:** separated
**Spectrometer type:** Ocean Optics Maya2000Pro
**Detector type:** Back-thinned, 2D FFT-CCD
**Optical fibers:** premium-grade UV/VIS Optical fibre, length - 10 m
**Filters:** no
**Mirrors:** quartz mirror
**Temperature control of spectrometer and detector:** 40°C/40°C |
| **Instrument performance** | **Spectral range/resolution:** 310–515/0.4 nm
**Azimuthal scan/direct-sun capabilities:** no/no
**Elevation angle capability:** set of 6 elevation angles, values can be adjusted but not the number of angles
**Field of view:** <1°
**Typical integration time:** 140 seconds
**Typical scan duration:** 15 minutes |
| **Calibration/characterization procedures** | **Elevation angles:** Two horizontal levels embedded in the base plate and in a plate holding the reflecting mirror are used to adjust the zero angle of the reflecting mirror. A stepping motor with an angle step of 0.038) is used for controlling the mirror angle.
**Field of view:** Characterized by Prede
**Stray light:** Subtracted as an offset component in DOAS analysis
**Dark signal:** nightly measurements
**Line shape:** An asymmetry Gaussian shape is determined during the wavelength calibration.
**Polarization:** -
**Detector nonlinearity:** characterized by Ocean Optics
**Pixel-to-pixel variability**: nightly measurements |
| **Spectral analysis software** | JM2 (Japanese MAX-DOAS profile retrieval algorithm, version 2) |
| **Supporting measurements** | none |
| **Reference** | Irie, H., H. M. S. Hoque, A. Damiani, H. Okamoto, A. M. Fatmi, P. Khatri, T. Takamura, and T. Jarupongsakul, Simultaneous observations by sky radiometer and MAX-DOAS for characterization of biomass burning plumes in central Thailand in January-April 2016, Atmos. Meas. Tech., 12, 599-606, https://doi.org/10.5194/amt-12-599-2019, January 29, 2019. |

| | |
|---|---|
| **Institute:** Chinese Academy of Meteorology Science, China Meteorological Administration (**CMA**), Beijing, China
**Responsible person(s):** Junli Jin, Jianzhong Ma
**Contact details:** jinjunli@camscma.cn |
[Figure]
 |
| **Instrument type:** mini-DOAS Hoffmann UV (#1) | **Nr:**
CINDI-2.07 |

| | |
|---|---|
| **Overall design of the instrument** | **Optical head including telescope:** integrated
**Spectrometer type:** Ocean Optics usb 2000
**Detector type:** Sony ILX511 CCD (2048 pixels)
**Optical fibers:** n/a
**Temperature control of spectrometer/detector:** n/a |
| **Instrument performance** | **Spectral range/resolution:** 292-447/0.6-0.8 nm
**Azimuthal scan/direct-sun capabilities:** no/no
**Elevation angle capability:** fully configurable
**Field of view:** 0.8°
**Typical integration time:** 1-2 minutes
**Typical scan duration:** 15-30 minutes |
| **Calibration/characterization procedures** | **Elevation angles:** horizontal scan calibration
**Field of view:** not yet characterized
**Straylight:** not characterized
**Dark signal:** measurement in night or measured with telescope covered, then substracted before spectra analysis
**Line shape:** not yet characterized
**Polarization:** not yet characterized
**Detector nonlinearity:** not yet characterized
**Pixel-to-pixel variability**: not yet characterized |
| **Spectral analysis software** | WinDOAS |
| **Supporting measurements** | none |

| | |
|---|---|
| **Institute:** Chinese Academy of Meteorology Science, China Meteorological Administration (**CMA**), Beijing, China

**Responsible person(s):** Junli Jin, Jianzhong Ma

**Contact details:** jinjunli@camscma.cn |
[Figure]
 |
| **Instrument type:** mini-DOAS Hoffmann VIS (#1) | **Nr:**
CINDI-2.08 |

| | |
|---|---|
| **Overall design of the instrument** | **Optical head including telescope:** integrated
**Spectrometer type:** Ocean Optics usb 2000
**Detector type:** DET2B-vis (2048 pixels)
**Optical fibers:** n/a
**Filters:** n/a
**Mirrors:** n/a
**Temperature control of spectrometer/detector:** n/a |
| **Instrument performance** | **Spectral range/resolution:** 399-712/0.6-0.8 nm
**Azimuthal scan/direct-sun capabilities:** no/no
**Elevation angle capability:** fully configurable
**Field of view:** 0.8°
**Typical integration time:** 1-2 minutes
**Typical scan duration:** 15-30 minutes |
| **Calibration/characterization procedures** | **Elevation angles:** horizontal scan calibration
**Field of view:** not characterized
**Dark signal:** measurement in night or measured with telescope covered, then substracted before spectra analysis
**Line shape:** not yet characterized
**Polarization:** not yet characterized
**Detector nonlinearity:** not yet characterized
**Pixel-to-pixel variability**: not yet characterized |
| **Spectral analysis software** | WinDOAS |
| **Supporting measurements** | none |

| | |
|---|---|
| **Institute:** Department of Atmospheric Chemistry and Climate (AC2), Spanish National Research Council (**CSIC**), Madrid, Spain

**Responsible person(s):** David García, Nuria Benavent, Shanshan Wang

**Contact details:** dgarcia@iqfr.csic.es |
[Figure]
 |

| | |
|---|---|
| **Instrument type:** MAX-DOAS | **Nr:**
CINDI-2.10 |

| | |
|---|---|
| **Overall design of the instrument** | **Optical head including telescope:** separated; elevation angles fully configurable

**Spectrometer type:** Princeton Acton SP2500

**Detector type:** Pixis 2D CCD Camera, 1340x400 pixels

**Optical fibers:** Multifiber UV-VIS, 10 m length

**Temperature control of spectrometer and detector:** 20-25°C and 70°C |
| **Instrument performance** | **Spectral range/resolution:** 300–500/0.5 nm

**Azimuthal scan/direct-sun capabilities:** no/no

**Elevation angle capability:** fully configurable

**Field of view:** approx. 0.7° (estimated using white stripe method)

**Typical integration time:** 0.01-1s

**Typical scan duration:** 5 minutes |
| **Calibration/characterization procedures** | **Elevation angles:** 45 °

**Field of view:** lamp in telescope

**Straylight:** -

**Dark signal:** by using the shutter

**Line shape:** Hg/Ne

**Polarization:** -

**Detector nonlinearity:** laboratory
**Pixel-to-pixel variability**: laboratory |
| **Spectral analysis software** | QDOAS |
| **Supporting measurements** | Video camera |
| **Reference** | Prados-Roman, C., Cuevas, C. A., Hay, T., Fernandez, R. P., Mahajan, A. S., Royer, S.-J., Galí, M., Simó, R., Dachs, J., Großmann, K., Kinnison, D. E., Lamarque, J.-F., and Saiz-Lopez, A.: Iodine oxide in the global marine boundary layer, Atmos. Chem. Phys., 15, 583-593, https://doi.org/10.5194/acp-15-583-2015, 2015. |

| | |
|---|---|
| **Institute:** University of Colorado (**CU-Boulder**), Boulder, Colorado

**Responsible person(s):** Rainer Volkamer, Henning Finkenzeller

**Contact details:** Rainer.Volkamer@colorado.edu, Henning.Finkenzeller@colorado.edu |
[Figure]
 |

| | |
|---|---|
| **Instrument type:** 3D-MAX-DOAS | **Nr:**
CINDI-2.11 |

| | |
|---|---|
| **Overall design of the instrument** | **Optical head including telescope:** separated; elevation and azimuth angles fully configurable; integrating sphere for direct sun measurements

**Spectrometer type:** 2 x Acton SP2150

**Detector type:** 2 x PIXIS 400 back-illuminated CCD

**Optical fibers:** Monofiber, diameter: 1.25mm, length: 25m connects to

Y-type bundle, diameter: 0.145mm, length: 1m

**Filters:** BG3/BG38, GG395

**Mirrors:** quartz prisms

**Temperature control of spectrometer and detector:** 34°C/-30°C |
| **Instrument performance** | **Spectral range/resolution:** 327-470/0.7 & 432–678/1.2 nm

**Azimuthal scan/direct-sun capabilities:** yes/yes

**Elevation angle capability:** fully configurable

**Field of view:** 0.7 degrees (full angle)

**Typical integration time:** ~20s

**Typical scan duration:** ~8min (12 EA & 12 Az) |
| **Calibration/characterization procedures** | **Elevation angles:** geometric alignment, solar aureole/horizon scan

**Field of view:** laser pointer backwards

**Straylight:** dark areas on CCD

**Dark signal:** characterized at night, and by dark areas on CCD

**Line shape:** Hg/Kr lamps (external) & QDOAS for wavelength dependency

**Polarization:** -

**Detector nonlinearity:** Fraunhofer OD at different saturation levels of CCD
**Pixel-to-pixel variability**: monitored |
| **Spectral analysis software** | QDOAS |
| **Supporting measurements** | Webcam, Hg & Kr lamp |
| **Reference** | Baidar, S., Oetjen, H., Coburn, S., Dix, B., Ortega, I., Sinreich, R., and Volkamer, R.: The CU Airborne MAX-DOAS instrument: vertical profiling of aerosol extinction and trace gases, Atmos. Meas. Tech., 6, 719-739, https://doi.org/10.5194/amt-6-719-2013 , 2013. |

| | |
|---|---|
| **Institute:** University of Colorado (**CU-Boulder**), Boulder, Colorado
**Responsible person(s):** Rainer Volkamer
**Contact details:** Rainer.Volkamer@colorado.edu | |
| **Instrument type:** ZS & MAX-DOAS (1D) | **Nr:**
CINDI-2.12 |

| | |
|---|---|
| **Overall design of the instrument** | **Optical head including telescope:** rotating prism, elevation angles fully configurable horizon-to-horizon across zenith
**Spectrometer type:** Acton SP2356i & QE65000
**Detector type:** PIXIS 400 back-illuminated CCD & Sony CCD
**Optical fibers:** Monofiber, diameter: 1.5mm, length: 10m connects to
Y-type bundle, diameter: 0.145mm, length: 1m
**Filters:** BG3/BG38
**Mirrors:** quartz prism
**Temperature control of spectrometer/detector:** 34°C/-30°C |
| **Instrument performance** | **Spectral range/resolution:** 300-466/0.8 & 379–493/0.5 nm
**Azimuthal scan/direct-sun capabilities:** no/no
**Elevation angle capability:** fully configurable
**Field of view:** 0.4 degrees (full angle)
**Typical integration time:** ~30s
**Typical scan duration:** ~8min |
| **Calibration/characterization procedures** | **Elevation angles:** geometric alignment, horizon scan
**Field of view:** laser pointer backwards
**Straylight:** dark areas on CCD
**Dark signal:** characterized at night, and by dark areas on CCD
**Line shape:** Hg/Kr lamps (external) & QDOAS for wavelength dependency
**Polarization:** -
**Detector nonlinearity:** Fraunhofer line distortion at different sat levels
**Pixel-to-pixel variability**: monitored |
| **Spectral analysis software** | QDOAS |
| **Supporting measurements** | Webcam, Hg & Kr lamp |
| **Reference** | Coburn, S., Dix, B., Sinreich, R., and Volkamer, R.: The CU ground MAX-DOAS instrument: characterization of RMS noise limitations and first measurements near Pensacola, FL of BrO, IO, and CHOCHO, Atmos. Meas. Tech., 4, 2421-2439, https://doi.org/10.5194/amt-4-2421-2011 , 2011. |

[Figure]

| | |
|---|---|
| **Institute 1:** Institut fuer Methodik der Fernerkundung (IMF), Deutsches Zentrum fuer Luft- und Raumfahrt e.V. (**DLR**), Wessling, Germany

**Institute 2:** School of Earth and Space Sciences, University of Science and Technology of China (**USTC**), Hefei, Anhui, China

**Responsible person(s):** Nan Hao (DLR) and Cheng Liu (USTC)

**Contact details:** nan.hao@dlr.de, Chliu81@ustc.edu.cn | |
| **Instrument type:** 1D MAX-DOAS EnviMeS (#1) | **Nr:**
CINDI-2.13
CINDI-2.14 |

[Figure]

| | |
|---|---|
| **Overall design of the instrument** | **Optical head including telescope:** separated; elevation and azimuth angles fully configurable
**Spectrometer type UV and Vis:** Avantes AvaBench-75
**Detector type UV:** Backthinned Hamamatsu CCD (2048 pixel)
**Detector type vis:** Backthinned Hamamatsu CCD (2048 pixel)
**Optical fibers:** Multifibre (UV), single fibre (VIS), length: 10m
**Filters:** UV bandpass filters (BG3)
**Mirrors:** none (rotatable prism for elevation angle selection)
**Temperature control of spectrometer and detector UV:** 20°C/20°C
**Temperature control of spectrometer and detector vis:** 20°C/20°C |
| **Instrument performance** | **Spectral range/resolution UV:** 296–460/0.56 nm
**Spectral range/resolution vis:** 440–583/0.54 nm
**Azimuthal scan/direct-sun capabilities:** yes/no
**Elevation angle capability:** fully configurable; step: 0.1° or less
**Field of view:** <0.5°
**Typical integration time:** 2.5ms -60s
**Typical scan duration:** 5 minutes |
| **Calibration/characterization procedures** | **Elevation angles:** Point-like light source and laser level
**Field of view:** Point-like light source and laser level
**Straylight:** Optical filters
**Dark signal:** Measurement during the night
**Line shape:** Atomic emission lines (Hg/Ne)
**Polarization:** n/a (depolarizing fibre)
**Detector nonlinearity:** Measurement of artificial light source with varying integration times
**Pixel-to-pixel variability**: Halogen lamp |
| **Spectral analysis software** | DOASIS |
| **Supporting measurements** | Webcam, tilt sensor, GPS |

| | |
|---|---|
| **Institute:** Indian Institute of Science Education and Research Mohali
Department of Earth and Environmental Sciences, Indian Institute of Science Education and Research Mohali (**IISERM**), Punjab, India
**Responsible person(s):** Abhishek Kumar Mishra and Vinod Kumar
**Contact details:** abhishekkumar.mishra21@gmail.com, vinodkumar@iisermohali.ac.in |
[Figure]
 |

| | | |
|---|---|---|
| **Instrument type:** mini-MAX DOAS Hoffmann UV (#2) | **Nr:**
CINDI-2.16 | |

| | |
|---|---|
| **Overall design of the instrument** | **Optical head including telescope:** integrated
**Spectrometer type UV:** Ocean Optics usb 2000+
**Spectrometer type:** CCD (2048 pixels)
**Filters:** no
**Mirrors:** -
**Temperature control of spectrometer and detector:** Peltier cooler |
| **Instrument performance** | **Spectral range/resolution:** 316–466/0.7 nm
**Azimuthal scan/direct-sun capabilities:** no/no
**Elevation angle capability:** fully configurable; step: 0.1° or less
**Field of view:** 0.7°
**Typical integration time:** 60ms
**Typical scan duration:** ~5 minutes for one full elevation sequence |
| **Calibration/characterization procedures** | **Elevation angles:** - Horizon calibration (-3° – 3°) every noon, Distant point source calibration in night
**Field of view:** -Point light source
**Straylight:** - not characterized
**Dark signal:** - Recorded every night
**Line shape:** - Gaussian like
**Polarization:** - Not characterized
**Detector nonlinearity:** - Not characterized
**Pixel-to-pixel variability**: - Not characterized |
| **Spectral analysis software** | WinDOAS and DOASIS |
| **Supporting measurements** | None |

| | |
|---|---|
| **Institute:** National Institute for Aerospace Technology (**INTA**), Madrid, Spain
**Responsible person(s):** Olga Puentedura
**Contact details:** puentero@inta.es |
[Figure]
 |

| | |
|---|---|
| **Instrument type:** 2D-MAX-DOAS RASAS III | **Nr:**
CINDI-2.17 |

| | |
|---|---|
| **Overall design of the instrument** | **Optical head including telescope:** separated; elevation and azimuth angles fully configurable
**Spectrometer type:** Andor Shamrock SR-163i
**Detector type:** IDUS Andor BU2
**Optical fibres:** Bundle 100 μm, length: 8 m
**Filters:** No
**Mirrors:** No
**Temperature control of spectrometer/detector:** 17°C/-30ºC |
| **Instrument performance** | **Spectral range/resolution:** 420-540/0.55 nm
**Azimuthal scan/direct-sun capabilities:** yes/no
**Elevation angle capability:** fully configurable
**Field of view:** 1°
**Typical integration time:** ~1 minute/pointing direction
**Typical scan duration:** ~1 minute x number of pointing directions |
| **Calibration/characterization procedures** | **Elevation angles:** Inclinometer during operation
**Field of view:** Geometrical
**Straylight:** HeNe LASER and optical filters
**Dark signal:** Measured at constant temperature with different integration times and subtracted during analysis
**Line shape:** HgCd lamp
**Polarization:** Optical fibre depolarizes the signal
**Detector nonlinearity:** Stable source and varying integration times
**Pixel-to-pixel variability**: Halogen lamp |
| **Spectral analysis software** | LANA software |
| **Supporting measurements** | Video camera, inclinometer and GPS |
| **Reference** | Puentedura, O., Gil, M., Saiz-Lopez, A., Hay, T., Navarro-Comas, M., Gómez-Pelaez, A., Cuevas, E., Iglesias, J., and Gomez, L.: Iodine monoxide in the north subtropical free troposphere, Atmos. Chem. Phys., 12, 4909-4921, https://doi.org/10.5194/acp-12-4909-2012, 2012. |

| | |
|---|---|
| **Institute:** Institute of Environmental Physics (**IUP-Bremen**), University of Bremen, Bremen, Germany

**Responsible person(s):** Andreas Richter

**Contact details:** richter@iup.physik.uni-bremen.de |
[Figure]
 |

| | | |
|---|---|---|
| **Instrument type:** 2 channel scientific grade elevation and azimuth scanning MAX-DOAS | **Nr:**
CINDI-2.18 | |

| | |
|---|---|
| **Overall design of the instrument** | **Optical head including telescope:** separated; elevation and azimuth angles fully configurable
**Spectrometer type UV:** Acton ARC500
**Spectrometer type vis:** Acton ARC500
**Detector type UV:** Princeton NTE/CCD-1340/400-EMB
**Detector type vis:** Princeton NTE/CCD-1340/400-EMB
**Optical fibers:** Y-type quartz bundle, diameter: 150µm, length: 22m
**Filters:** UG5 (UV only)
**Mirrors:** no
**Temperature control of spectrometer and detector UV:** 35°C/-35°C
**Temperature control of spectrometer and detector vis:** 35°C/-30°C |
| **Instrument performance** | **Spectral range/resolution UV:** 305–390/0.5 nm
**Spectral range/resolution vis:** 406–579/0.85 nm
**Azimuthal scan/direct-sun capabilities:** yes/no
**Elevation angle capability:** fully configurable
**Field of view:** 1°
**Typical integration time:** 60s; 120s for zenith
**Typical scan duration:** 15 minutes for 11 elevation angles |
| **Calibration/characterization procedures** | **Elevation angles:** geometric alignment of telescope, horizon scan
**Field of view:** white light source in lab
**Straylight:** not yet characterized
**Dark signal:** nightly measurements
**Line shape:** HgCd lamp in telescope
**Polarization:** -
**Detector nonlinearity:** white light source in lab, characterization only
**Pixel-to-pixel variability:** white light source in lab, characterization only |
| **Spectral analysis software** | NLIN |
| **Supporting measurements** | Video camera, HgCd lamp |
| **Reference** | Peters, E., Wittrock, F., Großmann, K., Frieß, U., Richter, A., and Burrows, J. P., Formaldehyde and nitrogen dioxide over the remote western Pacific Ocean: SCIAMACHY and GOME-2 validation using ship-based MAX-DOAS observations, Atmos. Chem. Phys., 12, 11179-11197, doi:10.5194/acp-12-11179-2012, 2012. |

| | |
|---|---|
| **Institute:** Institute of Environmental Physics (**IUP-Bremen**), University of Bremen, Bremen, Germany
**Responsible person(s):** Enno Peters
**Contact details:** Enno.Peters@iup.physik.uni-bremen.de |
[Figure]
 |

| **Instrument type:** single channel scientific grade imaging-DOAS, telescope mounted on pan-tilt-head for azimuthal scans and zenith (reference) pointing, indoor parts equipped in a 19'' rack | **Nr:**
CINDI-2.37 |
|---|---|

| | |
|---|---|
| **Overall design of the instrument** | **Optical head including telescope:** separated; elevation and azimuth angles fully configurable
**Spectrometer type:** Andor Shamrock 303i
**Detector type:** Andor Newton DU940P-BU, 2048x512 pixel (only inner pixels used for imaging)
**Optical fibers:** Fibre bundle with 69 sorted single fibres, diameter: 100μm, length: 15m
**Filters:** BG39
**Mirrors:** no
**Temperature control of spectrometer and detector:** 35°C/-30°C |
| **Instrument performance** | **Spectral range/resolution:** 420 – 500nm/0.8 nm
**Azimuthal scan/direct-sun capabilities:** yes/n/a
**Elevation angle capability:** fully configurable
**Field of view:** vertically approx. 50° total, 1.5° per view, horizontally 1.2°
**Typical integration time:** 10s
**Typical scan duration:** 10 min for complete horizon scan (10° azimuthal steps 0-360° followed by zenith reference) |
| **Calibration/characterization procedures** | **Elevation angles:** between -5 and +30 + regular zenith-sky
**Field of view:** white light source in lab
**Straylight:** not yet characterized
**Dark signal:** manually
**Line shape:** HgCd lamp (manually)
**Polarization:** -
**Detector nonlinearity:** white light source in lab, characterization only
**Pixel-to-pixel variability:** white light source in lab, characterization only |
| **Spectral analysis software** | NLIN |
| **Supporting measurements** | Video camera |
| **Reference** | Peters, E., Ostendorf, M.,Bösch, T., Seyler, A., Schönhardt, A., Schreier, S.F., Henzing, J. S., Wittrock, F., Richter, A., Vrekoussis, M., Burrows,J.P., Full-azimuthal imaging-DOAS observations of NO2 and O4 during CINDI-2, submitted to AMT, 2019. |

| | |
|---|---|
| **Institute:** Institute of Environmental Physics (**IUP-Heidelberg**), University of Heidelberg, Heidelberg, Germany

**Responsible person(s):** Udo Friess

**Contact details:** udo.friess@iup.uni-heidelberg.de |
[Figure]
 |

| | | |
|---|---|---|
| **Instrument type:** 2D MAX-DOAS EnviMeS (#3) | **Nr:**
CINDI-2.19 | |

| | |
|---|---|
| **Overall design of the instrument** | **Optical head including telescope:** separated; elevation and azimuth angles fully configurable
**Spectrometer type UV and Vis:** Avantes AvaBench-75
**Detector type UV:** Backthinned Hamamatsu CCD (2048 pixel)
**Detector type vis:** Backthinned Hamamatsu CCD (2048 pixel)
**Optical fibers:** Multifibre (UV), single fibre (VIS), length: 10m
**Filters:** UV bandpass filters (BG3)
**Mirrors:** none (rotatable prism for elevation angle selection)
**Temperature control of spectrometer and detector UV:** 20°C/20°C
**Temperature control of spectrometer and detector vis:** 20°C/20°C |
| **Instrument performance** | **Spectral range/resolution UV:** 296–460/0.56 nm
**Spectral range/resolution vis:** 440–583/0.54 nm
**Azimuthal scan/direct-sun capabilities:** yes/yes
**Elevation angle capability:** fully configurable; step: 0.1° or less
**Field of view:** <0.5°
**Typical integration time:** 2.5ms - 60s
**Typical scan duration:** 5 minutes |
| **Calibration/characterization procedures** | **Elevation angles:** Point-like light source and laser level
**Field of view:** Point-like light source and laser level
**Straylight:** Optical filters
**Dark signal:** Measurement during the night
**Line shape:** Atomic emission lines (Hg/Ne)
**Polarization:** n/a
**Detector nonlinearity:** Measurement of artificial light source with varying integration times
**Pixel-to-pixel variability**: Halogen lamp |
| **Spectral analysis software** | DOASIS |
| **Supporting measurements** | Webcam, tilt sensor, GPS |
| **Reference** | Lampel, J., Frieß, U., and Platt, U.: The impact of vibrational Raman scattering of air on DOAS measurements of atmospheric trace gases, Atmos. Meas. Tech., 8, 3767-3787, https://doi.org/10.5194/amt-8-3767-2015, 2015. |

| | |
|---|---|
| **Institute:** Royal Netherlands Meteorological Institute (**KNMI**), De Bilt, The Netherlands

**Responsible person(s):** Ankie Piters

**Contact details:** ankie.piters@knmi.nl |
[Figure]
 |
| **Instrument type:** mini-DOAS Hoffmann UV (#3) | **Nr:**
CINDI-2.21 |

| | |
|---|---|
| **Overall design of the instrument** | **Optical head including telescope:** integrated
**Spectrometer type:** Ocean Optics usb 2000
**Detector type:** Sony ILX511 CCD (2048 pixels)
**Optical fibers:** n/a |
| **Instrument performance** | **Spectral range/resolution:** 290-443/0.6 nm
**Azimuthal scan/direct-sun capabilities:** no/no
**Elevation angle capability:** fully configurable
**Field of view:** 0.45°
**Typical integration time:** 1-2 minutes
**Typical scan duration:** 15-30 minutes |
| **Calibration/characterization procedures** | **Elevation angles:** calibration of horizon (+/-0.5 degree) via quick horizon-scan (-3 to +3, very short integration time)
**Field of view:** scanning over a light source in the laboratory
**Straylight:** not yet characterized
**Dark signal:** characterized in the dark room as a function of detector temperature
**Line shape:** determined from lamp lines (function of temperature and wavelength)
**Polarization:** not yet characterized
**Detector nonlinearity:** not yet characterized
**Pixel-to-pixel variability:** characterized in the dark room as a function of detector temperature |
| **Spectral analysis software** | Own software (Python-based) |
| **Supporting measurements** | none |
| **Reference** | Vlemmix, T., Piters, A.J.M., Stammes, P., Wang, P., and Levelt, P.F., Retrieval of tropospheric NO2 using the MAX-DOAS method combined with relative intensity measurements for aerosol correction, Atmos. Meas. Tech. 3, 1287-1305, 2010. |

| | |
|---|---|
| **Institute:** Royal Netherlands Meteorological Institute (**KNMI**), De Bilt, The Netherlands

**Responsible person(s):** Ankie Piters

**Contact details:** ankie.piters@knmi.nl |
[Figure]
 |
| **Instrument type:** mini-DOAS Hoffmann VIS (#3) | **Nr:**
CINDI-2.22 |

| | |
|---|---|
| **Overall design of the instrument** | **Optical head including telescope:** integrated

**Spectrometer type:** Ocean Optics usb 2000+

**Detector type:** Sony ILX511 CCD (2048 pixels) |
| **Instrument performance** | **Spectral range/resolution:** 400-600/0.5 nm

**Azimuthal scan/direct-sun capabilities:** no/no

**Elevation angle capability:** fully configurable

**Field of view:** 0.4°

**Typical integration time:** 1-2 minutes

**Typical scan duration:** 15-30 minutes |
| **Calibration/characterization procedures** | **Elevation angles:** calibration of horizon (+/-0.5 degree) via quick horizon-scan (-3 to +3, very short integration time)

**Field of view:** scanning over a light source in the laboratory

**Straylight**: not yet characterized

**Dark signal:** characterized in the dark room as a function of detector temperature

**Line shape:** determined from lamp lines (function of temperature and wavelength)

**Polarization:** not yet characterized

**Detector nonlinearity:** not yet characterized
**Pixel-to-pixel variability**: characterized in the dark room as a function of detector temperature |
| **Spectral analysis software** | Own software (Python-based) |
| **Supporting measurements** | none |
| **Reference** | Vlemmix, T, Tropospheric nitrogen dioxide inversions based on spectral measurements of scattered sunlight, PhD Thesis, Technische Universiteit Eindhoven, DOI: 10.6100/IR719874, 2011. |

| | |
|---|---|
| **Institute:** Royal Netherlands Meteorological Institute (**KNMI**), De Bilt, The Netherlands

**Responsible person(s):** Ankie Piters

**Contact details:** ankie.piters@knmi.nl |
[Figure]
 |
| **Instrument type:** PANDORA-1S (#1) | **Nr:**
CINDI-2.23 |

| | |
|---|---|
| **Overall design of the instrument** | **Optical head including telescope:** separated; elevation and azimuth angles fully configurable
**Spectrometer type:** AvaSpec-ULS2048x64
**Detector type:** 2048 x 64 pixel backthinned non-cooled Hamamatsu CCD
**Optical fibers:** single strand 400um core diameter high OH fused silica fiber, 10m long
**Filters:** spectral filters (U340 and BP300 to remove visible light), attenuation filters
**Mirrors:** no
**Temperature control of spectrometer and detector:** 20°C/20°C |
| **Instrument performance** | **Spectral range/resolution UV:** 290-530/0.6 nm
**Azimuthal scan/direct-sun capabilities:** yes/yes
**Elevation angle capability:** fully configurable
**Field of view:** circular, 1.5° (sky mode); 2.5° (sun mode)
**Typical integration time:** 2.4ms-300ms (sun), 20ms to 1000ms (sky)
**Typical scan duration:** 15-30s per pointing position |
| **Calibration/characterization procedures** | **Elevation angles:** based on astronomical calculations and scanning the solar disc
**Field of view:** determined from scanning the solar disc
**Stray light:** determined in laboratory from measuring monochromatic input at multiple wavelengths
**Dark signal:** determined after each measurement
**Line shape:** determined in the laboratory from measurements of several spectral lamps
**Polarization:** no residual polarization measured after 10m fiber
**Detector nonlinearity:** determined in laboratory from tungsten halogen lamp measurements at different integration times
**Pixel-to-pixel variability**: determined in laboratory from tungsten halogen lamp measurement |
| **Spectral analysis software** | Own software (Python-based) and Blick Software Suite (Python-based) |
| **Supporting measurements** | None |
| **Reference** | J. Herman, A. Cede, E. Spinei, G. Mount, M. Tzortziou, and N. Abuhassan, NO2 column amounts from ground-based Pandora and MFDOAS spectrometers using the direct-sun DOAS technique: Intercomparisons and application to OMI validation, J. Geophys. Res., 114, D13307, doi:10.1029/2009JD011848, 2009. |

| | |
|---|---|
| **Institute:** Laboratoire Atmosphère, Milieux, Observations Spatiales (**LATMOS**), Guyancourt, France

**Responsible person(s):** Andrea Pazmino

**Contact details:** andrea.pazmino@latmos.ipsl.fr,
Manuel.pinharanda@latmos.ipsl.fr |
[Figure]
 |
| **Instrument type:** Système d'Analyse par Observation Zénithale (SAOZ) | **Nr:**
CINDI-2.24 |

| | |
|---|---|
| **Overall design of the instrument** | **Optical head including telescope:** n/a
**Spectrometer type:** Jobin-Yvon CP200 flat field
**Detector type:** 1024 NMOS diode array from Hamamatsu
**Optical fibers:** n/a
**Filters:** no
**Mirrors:** Yes
**Temperature control of spectrometer and detector:** no |
| **Instrument performance** | **Spectral range/resolution:** 270–640/1.3 nm
**Azimuthal scan/direct-sun capabilities:** n/a
**Elevation angle capability:** n/a
**Field of view:** 20º
**Exposure time:** 0.19 s - 5 x measurement cycle (adjusted automatically)
**Measurement cycle:** 60 s (programmable) |
| **Calibration/characterization procedures** | **Elevation angles:** n/a
**Field of view:** n/a
**Straylight:** n/a
**Dark signal:** shutter
**Line shape:** wavelength calibration based on reference spectrum
**Polarization:** Est-West fixed direction of the entrance slit
**Detector nonlinearity:** exposure time calibrated to 12000 counts in elementary spectrum
**Pixel-to-pixel variability**: dark background |
| **Spectral analysis software** | SAM version 5.9 |
| **Supporting measurements** | GPS |
| **Reference** | Pazmiño A., O3 and NO2 vertical columns using SAOZ UV-Visible spectrometer, EPJ Web of Conferences, Vol 9: ERCA 9 – From the Global Mercury Cycle to the Discoveries of Kuiper Belt Objects, p. 201-214, doi:10.1051/epjconf/201009016, 2010. |

| | |
|---|---|
| **Institute:** Laboratoire Atmosphère, Milieux, Observations Spatiales (**LATMOS**), Guyancourt, France

**Responsible person(s):** Andrea Pazmino

**Contact details:** andrea.pazmino@latmos.ipsl.fr, Manuel.pinharanda@latmos.ipsl.fr |
[Figure]
 |

| | | |
|---|---|---|
| **Instrument type:** Mini Système d'Analyse par Observation Zénithale (mini-SAOZ) | **Nr:**
CINDI-2.25 | |

| | |
|---|---|
| **Overall design of the instrument** | **Optical head:** separated
**Spectrometer type:** Cerny-Turner, grating 600 grooves/mm
**Detector type:** 2048x16 CCD back-thinned from Hamamatsu
**Optical fibers:** HGC950; diameter: 950 µm; length:10 m
**Filters:** OSC-UB
**Temperature control of spectrometer and detector:** n/a |
| **Instrument performance** | **Spectral range/resolution:** 270–820/0.7 nm
**Azimuthal scan/direct-sun capabilities:** n/a
**Elevation angle capability:** n/a
**Field of view:** 8°
**Exposure time:** 0.037 s - 5 x measurement cycle (adjusted automatically)
**Measurement cycle:** 60 s (programmable) |
| **Calibration/characterization procedures** | **Elevation angles:** n/a
**Field of view:** n/a
**Straylight:** n/a
**Dark signal:** shutter
**Line shape:** wavelength calibration based on reference spectrum
**Polarization:** n/a
**Detector nonlinearity:** exposure time calibrated to 12000 counts in elementary spectrum spectrum (semi-blind campaign)
Characterisation using stable light source at different integration time (after campaign)
**Pixel-to-pixel variability:** dark background |
| **Spectral analysis software** | SAOZ.gui Version 1.25-50f870 |
| **Supporting measurements** | GPS |
| **Reference** | Piters, A. J. M. et al.: The Cabauw Intercomparison campaign for Nitrogen Dioxide measuring Instruments (CINDI): design, execution, and early results, Atmos. Meas. Tech., 5(2), 457-485, 2012, doi:10.5194/amt-5-457-2012, 2012. |

| | |
|---|---|
| **Institute:** Meteorologisches Institut, Ludwig-Maximilians-Universität München (**LMU-MIM**), Munich, Germany

**Responsible person(s):** Mark Wenig

**Contact details:** mark.wenig@physik.uni-muenchen.de, lok.chan@ physik.uni-muenchen.de |
[Figure]
 |
| **Instrument type:** 2D MAX-DOAS EnviMeS (#4) | **Nr:**
CINDI-2.35 |

| | |
|---|---|
| **Overall design of the instrument** | **Optical head including telescope:** separated; elevation and azimuth angles fully configurable
**Spectrometer type UV:** Avantes AvaBench-75
**Spectrometer type vis:** Avantes AvaBench-75
**Detector type UV:** Backthinned Hamamatsu CCD (2048 pixel)
**Detector type vis:** Backthinned Hamamatsu CCD (2048 pixel)
**Optical fibers:** Multifibre (UV), single fibre (VIS), length: 10m
**Filters:**  UV bandpass filters (BG3)
**Mirrors:** N/A
**Temperature control of spectrometer and detector UV:** 20°C/20°C
**Temperature control of spectrometer and detector vis:** 20°C/20°C |
| **Instrument performance** | **Spectral range/resolution UV:** 305–460/0.56 nm
**Spectral range/resolution vis:** 430–650/0.54 nm
**Azimuthal scan/direct-sun capabilities:** yes/yes
**Elevation angle capability:** fully configurable
**Field of view:** <0.5°
**Typical integration time:** 2.5ms -60s
**Typical scan duration:** 15 min |
| **Calibration/characterization procedures** | **Elevation angles:** tilt sensor
**Field of view:** not yet characterized
**Straylight:** not yet characterized
**Dark signal:**  not yet characterized
**Line shape:** not yet characterized
**Polarization:** not yet characterized
**Detector nonlinearity:** not yet characterized
**Pixel-to-pixel variability**: not yet characterized |
| **Spectral analysis software** | DOASIS |
| **Supporting measurements** | Two video cameras, inclinometer |
| **Reference** | Lampel, J., Frieß, U., and Platt, U.: The impact of vibrational Raman scattering of air on DOAS measurements of atmospheric trace gases, Atmos. Meas. Tech., 8, 3767-3787, https://doi.org/10.5194/amt-8-3767-2015, 2015. |

| | |
|---|---|
| **Institute:** **LuftBlick**, Mutters, Austria

**Responsible person(s):** Alexander Cede

**Contact details:** alexander.cede@luftblick.at |
[Figure]
 |

| | | |
|---|---|---|
| **Instrument type:** PANDORA-2S (#2 & #3) | | **Nr:**
CINDI-2.26
CINDI-2.27 |

| | |
|---|---|
| **Overall design of the instrument** | **Optical head including telescope:** separated; elevation and azimuth angles fully configurable
**Spectrometer type:** AvaSpec-ULS2048x64 (one for UV and one for vis)
**Detector type:** 2048 x 64 pixel backthinned non-cooled Hamamatsu CCD (one for UV and one for vis)
**Optical fibers:** single strand 400um core diameter high OH fused silica fiber, 10m long
**Filters:** spectral filters (U340 and BP300 to remove visible light), attenuation filters
**Mirrors:** no
**Temperature control of spectrometer and detector UV:** 20°C/20°C
**Temperature control of spectrometer and detector VIS:** 20°C/20°C |
| **Instrument performance** | **Spectral range/resolution UV:** 280 - 540/0.6 nm
**Spectral range/resolution vis:** 380 - 900/1.1 nm
**Azimuthal scan/direct-sun capabilities:** yes/yes
**Elevation angle capability:** fully configurable
**Field of view:** circular, 1.5° (sky mode); 2.8° (sun mode)
**Typical integration time:** 2.4ms-300ms (sun), 20ms to 1000ms (sky)
**Typical scan duration:** 15-30s per pointing position |
| **Calibration/characterization procedures** | **Elevation angles:** based on astronomical calculations and scanning the solar disc
**Field of view:** determined from scanning the solar disc
**Stray light:** determined in the laboratory from measuring monochromatic input at different wavelengths
**Dark signal:** determined after each measurement
**Line shape:** determined in the laboratory from measurements of several spectral lamps
**Polarization:** no residual polarization measured after 10m fiber
**Detector nonlinearity:** determined in laboratory from tungsten halogen lamp measurements at different integration times
**Pixel-to-pixel variability**: determined in laboratory from tungsten halogen lamp measurements |
| **Spectral analysis software** | Blick Software Suite (Python-based) |
| **Supporting measurements** | None |
| **Reference** | J. Herman, A. Cede, E. Spinei, G. Mount, M. Tzortziou, and N. Abuhassan, NO2 column amounts from ground-based Pandora and MFDOAS spectrometers using the direct-sun DOAS technique: Intercomparisons and application to OMI validation, J. Geophys. Res., 114, D13307, doi:10.1029/2009JD011848, 2009. |

| | |
|---|---|
| **Institute:** Max-Planck Institute for Chemistry (**MPIC**), Mainz, Germany

**Responsible person(s):** Thomas Wagner

**Contact details:** thomas.wagner@mpic.de |
[Figure]
 |
| **Instrument type:** TubeMAX-DOAS      53 | **Nr:**
CINDI-2.28 |

| | |
|---|---|
| **Overall design of the instrument** | **Optical head including telescope:** separated; elevation angles fully configurable
**Spectrometer type:** Avantes
**Detector type:** CCD
**Optical fibers:** quartz fibre bundle, length: 5 m
**Filters:** BG3 (UV)
**Mirrors:** no
**Temperature control of spectrometer and detector:** 20°C/20°C |
| **Instrument performance** | **Spectral range/resolution:** 305–464/0.6 nm
**Azimuthal scan/direct-sun capabilities:** no/no
**Elevation angle capability:** fully configurable
**Field of view:** 0.7°
**Typical integration time:** 60s
**Typical scan duration:** 15 minutes (depends on sequence) |
| **Calibration/characterization procedures** | **Elevation angles:** performed at the campaign using laser device or water level
**Field of view:** performed at the campaign using laser device or water level
**Straylight:** has to be quantified
**Dark signal:** measured on site and corrected
**Line shape:** almost symmetric Gaussian-like, almost not dependent on wavelength
**Polarization:** -
**Detector nonlinearity:** characterised in the laboratory
**Pixel-to-pixel variability:** - |
| **Spectral analysis software** | Windoas and QDOAS |
| **Supporting measurements** | Video camera |
| **Reference** | Donner, S., Mobile MAX-DOAS measurements of the tropospheric formaldehyde column in the Rhein-Main region. Master Thesis, Universität, Mainz,http://hdl.handle.net/11858/00-001M-0000-002C-EB17-2, 2016. |

| | |
|---|---|
| **Institute: NASA**-Goddard (Greenbelt, Maryland)
**Responsible person(s):** Jay Herman
**Contact details:** jay.r.herman@nasa.gov, Elena Spinei
(elena.spinei@nasa.gov) |
[Figure]
 |

| **Instrument type:** PANDORA-1S (#4 & #5) | **Nr:**
CINDI-2.31
CINDI-2.32 |
|---|---|

| | |
|---|---|
| **Overall design of the instrument** | **Optical head including telescope:** separated; elevation and azimuth angles fully configurable
**Spectrometer type:** AvaSpec-ULS2048x64 (one for 285 – 530 nm)
**Detector type:** 2048 x 64 pixel backthinned non-cooled Hamamatsu CCD
**Optical fibers:** single strand 400um core diameter high OH fused silica fiber, 10m long
**Filters:** spectral filters (U340 and BP300 to remove visible light), attenuation filters
**Mirrors:** no
**Temperature control of spectrometer and detector UV:** 20°C/20°C
**Temperature control of spectrometer and detector VIS:** 20°C/20°C |
| **Instrument performance** | **Spectral range/resolution UV:** 280-540/0.6 nm
**Azimuthal scan/direct-sun capabilities:** yes/yes
**Elevation angle capability:** fully configurable
**Field of view:** circular, 1.6° (sky mode); 2.8° (sun mode)
**Typical integration time:** 2.4ms-300ms (sun), 20ms to 1000ms (sky)
**Typical scan duration:** 15-30s per pointing position |
| **Calibration/characterization procedures** | **Elevation angles:** based on astronomical calculations and scanning the solar disc
**Field of view:** determined from scanning the solar disc
**Stray light:** determined in laboratory from measuring monochromatic input at multiple wavelengths
**Dark signal:** determined after each measurement
**Line shape:** determined in the laboratory from measurements of several spectral lamps
**Polarization:** no residual polarization measured after 10m fiber
**Detector nonlinearity:** determined in laboratory from tungsten halogen lamp measurements at different integration times
**Pixel-to-pixel variability**: determined in laboratory from tungsten halogen lamp measurement |
| **Spectral analysis software** | Blick Software Suite (Python-based) |
| **Supporting measurements** | None |
| **Reference** | J. Herman, A. Cede, E. Spinei, G. Mount, M. Tzortziou, and N. Abuhassan, NO2 column amounts from ground-based Pandora and MFDOAS spectrometers using the direct-sun DOAS technique: Intercomparisons and application to OMI validation, J. Geophys. Res., 114, D13307, doi:10.1029/2009JD011848, 2009. |

| | |
|---|---|
| **Institute:** National Institute of Water and Atmospheric Research (**NIWA**), Lauder, New Zealand
**Responsible person(s):** Richard Querel, Paul Johnston
**Contact details:** richard.querel@niwa.co.nz |
[Figure]
 |

| | | |
|---|---|---|
| **Instrument type:** EnviMeS 1D MAX-DOAS (#3) | 55 | **Nr:**
CINDI-2.29 |

| | |
|---|---|
| **Overall design of the instrument** | **Optical head including telescope:** elevation angle configurable
**Spectrometer type UV:** Avantes AvaBench-75
**Spectrometer type vis:** Avantes AvaBench-75
**Detector type UV:** Backthinned Hamamatsu CCD (2048 x 64 pixels)
**Detector type vis:** Backthinned Hamamatsu CCD (2048 x 64 pixels)
**Optical fibers:** Multifibre (6 x UV), single fibre (1 x VIS), length: 10m
**Filters:** UV bandpass filter (BG3), VIS bandpass filter (BG40)
**Mirrors:** Rotating glass quartz prism as entrance optic
**Temperature control of spectrometer and detector UV:** 20 °C / 20 °C
**Temperature control of spectrometer and detector vis:** 20 °C / 20 °C |
| **Instrument performance** | **Spectral range/resolution UV:** 305–457 nm / 0.7 nm
**Spectral range/resolution vis:** 410–550 nm / 0.7 nm
**Azimuthal scan/direct-sun capabilities:** no
**Elevation angle capability:** fully configurable; step: 0.1° or less
**Field of view:** <0.5°
**Typical integration time:** 2.5ms -60s
**Typical scan duration:** 60 s |
| **Calibration/characterization procedures** | **Elevation angles:** Calibrated tilt meter and level
**Field of view:** not measured
**Straylight:** not measured
**Dark signal:** shutter blocks light path in scanning head
**Line shape:** taken from Hg lamp spectra
**Polarization:** 10 m fibre effectively depolarizes incoming light
**Detector nonlinearity:** observations of a temperature stabilized LED with several different exposure times, assuming LED to be constant intensity.
**Pixel-to-pixel variability**: Not tested |
| **Spectral analysis software** | DOASIS, STRATO |
| **Supporting measurements** | Tilt sensor (for elevation angle), PTU |
| **Reference** | Lampel, J., Frieß, U., and Platt, U.: The impact of vibrational Raman scattering of air on DOAS measurements of atmospheric trace gases, Atmos. Meas. Tech., 8, 3767-3787, https://doi.org/10.5194/amt-8-3767-2015, 2015. |

| | |
|---|---|
| **Institute:** National Institute of Water and Atmospheric Research (**NIWA**), Lauder, New Zealand
**Responsible person(s):** Richard Querel, Paul Johnston
**Contact details:** richard.querel@niwa.co.nz |
[Figure]
 |

| | | |
|---|---|---|
| **Instrument type:** Lauder Acton275 MAX-DOAS | **Nr:**
CINDI-2.30 |
[Figure]
 |

| | |
|---|---|
| **Overall design of the instrument** | **Optical head including telescope:** elevation angle configurable
**Spectrometer type UV/Vis:** Acton 275 with grating control
**Detector type UV/Vis:** Backthinned Hamamatsu CCD (1044 x 128pixels x 24um)
**Optical fibers:** Multifibre with 100um fibres, input end circular 1mm diam, length: 12m
**Filters:**
**Mirrors:** Front silvered rotating mirror and quartz lens optic.
**Temperature control of detector:** -20 °C |
| **Instrument performance** | **Spectral range/resolution: multi band configurable; typical two bands are: alternating** 290–363 nm and 400-460; 0.6 nm
**Azimuthal scan/direct-sun capabilities:** no
**Elevation angle capability:** fully configurable; step: < 0.1°
**Field of view:** about 0.5°
**Typical integration time:** 16ms -20s
**Typical scan duration:** 60 s (but flexible) |
| **Calibration/characterization procedures** | **Elevation angles:** Bubble level on mirror and external laser level
**Field of view: m**easured using laser level
**Straylight:** estimated using Schott filters to cut light at shorter wavelengths.<1e-2 ?
**Dark signal:** night spectra or manual scan
**Line shape:** taken from Hg and other line lamp spectra
**Polarization:** 12 m fibre effectively depolarizes incoming light
**Detector nonlinearity:** quantified by comparing observations of a clear sky with and without neutral density filter.
**Pixel-to-pixel variability**: measured with white lamp. |
| **Spectral analysis software** | STRATO (Lauder, NIWA) |
| **Supporting measurements** | GPS time, Camera possible. |

<table>
<tr><td colspan="2">

**Institute:** National University of Sciences and Technology (**NUST**), Islamabad, Pakistan

**Responsible person(s):** Muhammad Fahim Khokhar and Junaid Khayyam Butt

**Contact details:** fahim.khokhar@iese.nust.edu.pk, jkb2ravian@gmail.com

</td><td rowspan="2">
[Figure]
</td></tr>
<tr><td>

**Instrument type:** Mini MAX-DOAS

</td><td>

**Nr:**
CINDI-
2.33

</td></tr>
</table>

| | |
|---|---|
| **Overall design of the instrument** | **Optical head including telescope:** integrated
**Spectrometer type:** Czerny-Turner spectrometer
**Detector type:** 1 dimensional CCD (Sony ILX511, 2048 individual pixels)
**Optical fibers:** n/a
**Filters:** n/a
**Mirrors:** n/a
**Temperature control of spectrometer and detector:** n/a |
| **Instrument performance** | **Spectral range/resolution:** 320–465/0.7 nm
**Azimuthal scan/direct-sun capabilities:** no/no
**Elevation angle capability:** fully configurable; 1 degree resolution
**Field of view:** ~1.2°
**Typical integration time:** 10-60s
**Typical scan duration:** 20 minutes |
| **Calibration/characterization procedures** | **Elevation angles:** water/sprit level
**Field of view:** n/a
**Straylight:** n/a
**Dark signal:** manual procedure
**Line shape:** n/a
**Polarization:** n/a
**Detector nonlinearity:** n/a
**Pixel-to-pixel variability:** n/a |
| **Spectral analysis software** | QDOAS (version:2.111) / WinDOAS |
| **Supporting measurements** | GPS but not integrated |

[Figure]

[Figure]

| | |
|---|---|
| **Institute:** Department of Physics, University of Toronto (**UTO**), Toronto, Canada
**Responsible person(s):** Kristof Bognar, Xiaoyi Zhao, Kimberly Strong
**Contact details:** kbognar@physics.utoronto.ca, xizhao@atmosp.physics.utoronto.ca, strong@atmosp.physics.utoronto.ca | |

| | | |
|---|---|---|
| **Instrument type:** PEARL-GBS instrument (MAX-DOAS, ZSL-DOAS, and DS) | **Nr:**
CINDI-2.36 | |

| | |
|---|---|
| **Overall design of the instrument** | **Optical head including telescope:** separated; elevation and azimuth angles fully configurable
**Spectrometer type:** Jobin Yvon Triax-180 triple-grating spectrometer
**Detector type:** back-illuminated cooled CCD with 2048 x 512 pixels
**Optical fibers:** fiber bundle (37 HOH mapped fibres, spot-to-slit), spot end diameter: ~0.8 mm, length: 6 m
**Filters:** Filter wheel containing one empty slot, four metallic neutral density filters (31.6%, 1%, 0.1%, 0.01% transmittance) and a UV diffuser
**Mirrors:** UV-enhanced aluminum (suntracker)
**Temperature control of spectrometer and detector:** 25°C/-70°C |
| **Instrument performance** | **Spectral range/resolution:** 340–560/0.75 nm
**Azimuthal scan/direct-sun capabilities:** yes/yes
**Elevation angle capability:** fully configurable
**Field of view:** 0.6°
**Typical integration time:** 50-140 s
**Typical scan duration:** 12-23 minutes for 9 elevation angles |
| **Calibration/characterization procedures** | **Elevation angles:** calibrated by levelling the suntracker
**Field of view:** calculated analytically
**Straylight:** determined using a red filter and a halogen lamp
**Dark signal:** determined from a series of closed shutter measurements
**Line shape:** assumed to be Gaussian
**Polarization:** determined using a polarizer and a halogen lamp; fiber bundle mostly depolarizes incoming light
**Detector nonlinearity:** <0.4% as given by the CCD manufacturer
**Pixel-to-pixel variability**: not characterized |
| **Spectral analysis software** | Raw data is processed using in-house MATLAB code and analysis is performed using the QDOAS software |
| **Supporting measurements** | Webcam |
| **Reference** | A. Fraser, C. Adams, J.R. Drummond, F. Goutail, G. Manney, and K. Strong. The Polar Environment Atmospheric Research Laboratory UV-Visible Ground-Based Spectrometer: First Measurements of $O_3$, $NO_2$, BrO, and OClO Columns. *J. Quant. Spectrosc. Radiat. Transfer*, **110 (12)**, 986-1004, 2009. |